# Horizontal Diffusion Models: Riemannian Score-based Generative Modeling via Frame-Connection Geometry

## Abstract

Modeling data supported on curved manifolds poses significant challenges due to the need for geometric operations such as geodesic computations, parallel transport, and geodesic distance, which are often intractable or ill-defined on general Riemannian manifolds. To address this, we propose a novel framework for generative modeling on manifolds that bypasses these limitations by operating directly on the orthonormal frame bundle, a geometric space that retains manifold structure while offering computational compatibility with Euclidean learning. Our method introduces horizontal diffusion processes whose dynamics and score fields respect the underlying geometry without requiring manifold-specific neural architectures. A key insight is that standard Euclidean score networks can be lifted into the frame bundle to yield geometry-consistent vector fields, enabling seamless integration of manifold constraints with modern generative modeling techniques. Through theoretical analysis and experiments on complex curved domains, including the parametric surfaces and celestial bodies, we demonstrate that our approach achieves high-quality generation while preserving geometric fidelity. This work provides a general and scalable pathway for bridging differential geometry and score-based generative models.

## 1 Introduction

Geometric modeling has become an essential paradigm in AI for Science, where data often inhabit intrinsically curved or structured spaces such as molecules, astrophysical objects, quantum systems, and materials with nontrivial topology or symmetry (Bronstein et al., 2021; Mathieu et al., 2019). In these scientific domains, capturing and generating data requires models that faithfully respect underlying geometric structures, whether they arise from molecular conformation spaces, orientation groups, biological manifolds, or celestial mechanics. However, the range of geometric structures that can be explored or modeled in practice has been limited by the structural assumptions and tractability of existing generative approaches. Many traditional and modern methods remain fundamentally constrained to spaces that admit global coordinates, explicit embeddings, or closed-form geometric primitives, which excludes a wide variety of naturally occurring and scientifically relevant manifolds. As a result, scientific generative modeling often falls short of harnessing the full richness of geometry observed in real-world datasets, especially as complexity, dimensionality, or curvature increases.

A broad array of geometric generative methods including Riemannian flows, score-based diffusion models, and manifold-aware normalizing flows have been proposed to address these challenges (Marin et al., 2021; Brehmer & Cranmer, 2020; Poli et al., 2020; De Bortoli et al., 2022a; Huang et al., 2022; Thornton et al., 2022; Jo et al., 2023; Bertolini et al., 2025; Courts & Kvinge, 2022). These approaches typically rely on explicit Riemannian geometric constructions such as exponential and logarithmic maps, geodesic distances, or volume forms, and often require closed-form geodesic computation, parallel transport, or spectral analysis of the Laplace-Beltrami operator. While powerful on highly symmetric or flat (*e.g.*, spheres, flat-torus), such methods encounter fundamental obstacles on general smooth manifolds: explicit geodesics or spectral decompositions are unavailable and parallel transport is computationally intensive and not intrinsic to learned geometry. Consequently, the scientific impact of geometric generative models is limited not only by technical but also by foundational geometric barriers, restricting their reach in scientific applications.

To move beyond these fundamental geometric barriers, we seek a truly intrinsic and scalable approach that does not rely on explicit geodesic computations. An elegant solution is found in the geometric and probabilistic machinery of the *orthonormal frame bundle* $\mathcal{O}(\mathcal{M})$ over a Riemannian manifold $(\mathcal{M}, g)$, where Euclidean stochastic processes are lifted into tangent spaces via moving frames. This perspective has deep roots in stochastic differential geometry, originating from the theory of stochastic development and Brownian motion on manifolds (Eells & Sampson, 1964; Elworthy, 1982; Hsu, 2002). In this framework, the Levi-Civita connection induces a horizontal distribution on $\mathcal{O}(\mathcal{M})$, enabling coordinate-free stochastic flows that preserve intrinsic geometry without relying on geodesics or spectral decompositions (Kobayashi & Nomizu, 1963a; Elworthy, 1988; Malliavin, 1978). Our work builds on this classical foundation and extends it to the generative modeling setting, providing a scalable and intrinsically consistent framework that bridges differential geometry with modern learning paradigms. The main contribution of this work is two-folds:

- **Connection-aware Riemannian Score-based Generative Modeling:** We introduce the first score-based generative modeling framework that operates on the orthonormal frame bundle, thereby enabling efficient and geometrically consistent diffusion modeling on general Riemannian manifolds. By leveraging the intrinsic geometry of the frame bundle, our method is applicable even to manifolds lacking closed-form analytic structures, and thus overcomes fundamental scalability limitations present in previous manifold diffusion models.

- **Horizontal Lift of Euclidean Process for Training and Inference:** We introduce a principled parameterization that lifts standard Euclidean score networks into the frame bundle as gauge-equivariant horizontal vector fields, ensuring intrinsic compatibility with manifold geometry and fiber symmetries. This construction allows seamless adaptation of existing neural architectures for manifold data, and supports fast, parallelizable sampling and training without requiring model-specific architectural changes or geometric post-processing.

## 2 HORIZONTAL DIFFUSION MODELS

This section introduces the basic structure of horizontal diffusion model, together with its associated training objective and geometric consistency induced by the frame bundle formulation. For notations not explicitly defined in this section, we refer the reader to Section A.

**Motivation.** To motivate our construction, we begin by recalling the conventional formulation of score-based generative modeling, which serves as the computational backbone for many diffusion models. Given arbitrary potential $\phi_\bullet : [0, T] \times \mathbb{R}^d \to \mathbb{R}$, consider a following pair of linear stochastic differential equations (SDEs) which defines a time-symmetric diffusion system in $\mathbb{R}^d$:

$$d\mathbf{E}_t = -\frac{1}{2}\nabla\phi_t(\mathbf{E}_t)dt + dW_t, \quad d\mathbf{E}_s = \left[-\frac{1}{2}\nabla\phi_s(\mathbf{E}_s) + \nabla\log\varrho_s(\mathbf{E}_s)\right]ds + dW_s. \quad (1)$$

These Euclidean forward–reverse SDEs have been widely employed in modern diffusion-based generative models, where the reverse drift involves the score function $\nabla\log\rho_t$ and is approximated by a neural network $s_\theta$. In recent works Bortoli et al. (2022), this framework has been generalized to *Riemannian manifolds* by replacing the Euclidean gradient $\nabla$ with the Riemannian gradient $\nabla^\mathcal{M}$, yielding the Riemannian forward-reverse dynamics with Riemannian potential $\phi_\bullet : [0, T] \times \mathcal{M} \to \mathbb{R}$:

$$d\mathbf{X}_t = -\frac{1}{2}\nabla^\mathcal{M}\phi_t(\mathbf{X}_t)dt + dW_t^\mathcal{M}, \, d\mathbf{X}_s = \left[-\frac{1}{2}\nabla^\mathcal{M}\phi_s(\mathbf{X}_s) + \nabla^\mathcal{M}\log\rho_s^\mathcal{M}(\mathbf{X}_s)\right]ds + dW_s^\mathcal{M},$$

$$(2)$$

where the diffusion $\mathbf{X}_t \in \mathcal{M}$ is governed by Brownian motion on the manifold, and all vector fields respect the intrinsic geometry. Despite providing a principled extension of diffusion modeling to curved spaces, these models depend critically on explicit geometric operations such as geodesic distance computations ($d_\mathcal{M}$), exponential and logarithmic maps, manifold-specific distribution estimations ($\rho_t^\mathcal{M}$), and tailored Riemannian score networks ($s_\theta^\mathcal{M}$) as highlighted in Table 1. These requirements introduce significant computational and analytical challenges, particularly for general curved manifolds where closed-form geodesic and exponential solutions are unavailable (*i.e.*, first row). Moreover, the absence of Christoffel symbols ($\Gamma_{ij}^k$) and gauge symmetry ($\mathcal{G}$) further restricts their scalability and poses substantial obstacles to exploring more general manifold geometries, where connection structures and fiber-wise symmetries become indispensable for accurate modeling.

| Method | $d_{\mathcal{M}}$ | exp | log | $\rho_t^{\mathcal{M}}$ | $s^{\mathcal{M}}$ | $g$ | $\Gamma_{ij}^k$ | $\mathcal{G}$ | $s^{\text{Euc}}$ |
|---|---|---|---|---|---|---|---|---|---|
| *Closed-form?* | ✗ | ✗ | ✗ | ✗ | ✗ | ✓ | ✓ | ✓ | ✓ |
| De Bortoli et al. (2022b) | ✓ | ✓ | ✓ | ✓ | ✓ | ✗ | ✗ | ✗ | ✗ |
| Thornton et al. (2022) | ✓ | ✓ | ✓ | ✓ | ✓ | ✗ | ✗ | ✗ | ✗ |
| Chen & Lipman (2024) | ✓ | ✓ | ✓ | ✓ | △ | ✗ | ✗ | ✗ | ✗ |
| Diepeveen et al. (2025b) | △ | △ | △ | ✓ | △ | ✗ | ✗ | ✗ | ✗ |
| Ours | ✗ | ✗ | △ | ✗ | ✗ | ✓ | ✓ | ✓ | ✓ |

**Table 1:** Comparison of geometric operators used in existing Riemannian generative models.

Our proposed framework uniquely addresses these limitations by avoiding explicit manifold-specific computations while instead leveraging Euclidean score networks ($s_\theta^{\text{Euc}}$). In particular, by operating directly on the orthonormal frame bundle, we make use of horizontal lifts defined through connection induced vector fields to ensure that the manifold's intrinsic structures such as metric ($g$), Christoffel symbols ($\Gamma_{ij}^k$), and gauge consistency ($\mathcal{G}$) are naturally incorporated. As a result, the method not only circumvents the need for costly manifold-dependent operations but also significantly enhances scalability and computational efficiency, enabling high-quality generative modeling across diverse and complex manifolds. Please refer to Section B for a detailed comparison.

**Frame-Connection Geometry**. For rigor, we review the geometric structure of the orthonormal frame bundle, which underlies our construction of horizontal score flows. Additional notations not included in the main paper are provided in the appendix. Let $(\mathcal{M}, g)$ be a connected, oriented, $d$-dimensional Riemannian manifold. The *orthonormal frame bundle* $O(\mathcal{M})$ is a principal $O(d)$-bundle over $\mathcal{M}$, with projection map $\pi : O(\mathcal{M}) \to \mathcal{M}$. Each point $\mathbf{U} = (x, e) \in O(\mathcal{M})$ consists of a base point $x \in \mathcal{M}$ and a linear isometry $e : \mathbb{R}^d \to T_x\mathcal{M}$, called a *frame*, which maps the standard Euclidean basis to a basis of the tangent space at $x$ satisfying $e^\top g(x)e = I$. The bundle admits a natural right action by $O(d)$: for $h \in O(d)$, the action is given by $\mathbf{U} \cdot h = (x, eh)$. This group action preserves the orthonormality of frames and defines the structure of the principal bundle.

The Levi–Civita connection $\nabla_{\mathcal{M}}$ on a Riemannian manifold $(\mathcal{M}, g)$ induces a *principal connection* on the orthonormal frame bundle $O(\mathcal{M})$ via a $\mathfrak{so}(d)$-valued 1-form $\omega_{\mathbf{U}}(V) := e^{-1}\left(\nabla_{\pi_*V}e\right)$ for any $V \in T_{\mathbf{U}}O(\mathcal{M})$. A tangent vector $V$ is said to be *horizontal* if $\omega_{\mathbf{U}}(V) = 0$, and the corresponding horizontal distribution is given by $\text{Hor}_{\mathbf{U}} := \ker \omega_{\mathbf{U}} \subset T_{\mathbf{U}}O(\mathcal{M})$. The canonical basis for this distribution consists of horizontal vector fields $\{H_a := H_e[\varepsilon^a]\}_{a=1}^d$, where each $H_a$ satisfies $\omega(H_a) = 0$ and $\pi_*H_a = e_a := e(\varepsilon_a)$, with $\{\varepsilon_a\}_{a=1}^d$ the standard basis of $\mathbb{R}^d$. To lift Euclidean forward–reverse SDEs to the frame bundle in a geometry-consistent way, we define a family of horizontal vector fields $H_e(\mathbf{U})[w]$ that map directions $w \in \mathbb{R}^d$ to elements of the horizontal distribution $H_{\mathbf{U}}$ at $\mathbf{U} = (x, e)$. Each $H[w]$ is characterized by $\pi_*H_e(\mathbf{U})[w] = e(w) \in T_x\mathcal{M}$ and $\omega_{\mathbf{U}}(H_e(\mathbf{U})) = 0$. Hence, it represents the horizontal lift of the tangent vector $e(w) \in T_x\mathcal{M}$. In local coordinates, using Christoffel symbols $\Gamma_{ij}^k(x)$, the horizontal lift conditioned on Euclidean vector $w$ can be written in coordinate-free form as a differential operator acting on smooth functions $f \in C^\infty(O(\mathcal{M}))$:

$$T_{\mathbf{U}}O(\mathcal{M}) \ni H_e(\mathbf{U})[w]f = w^a e_a^i \frac{\partial f}{\partial x^i} - \left(w^a \Gamma_{ij}^k(x)e_a^j \frac{\partial f}{\partial e_i^k}\right) \xleftarrow{\texttt{Horizontal Lift}} w \in T_w\mathbb{R}^d. \quad (3)$$

The operator $H$ is central to our construction, as it describes the *horizontal lift* of a Euclidean vector field to the frame-valued vector fields, preserving the intrinsic geometry defined by the Levi–Civita connection. This lift ensures that stochastic dynamics remain consistent with the manifold geometry without the need for global geodesic or exponential maps. This construction will serve as a computationally feasible mechanism for importing the Euclidean score-based framework into manifold settings, without requiring well-known special geometric structure.

**Horizontal Diffusion Models.** To extend score-based generative modeling beyond Riemannian settings, we formulate lifted score dynamics on the orthonormal frame bundle, where geometric constraints are naturally encoded through horizontal flows. In this setting, the Riemannian gradient $\nabla^{\mathcal{M}}$ on the base manifold is replaced by the horizontal gradient $\nabla^{\text{Hor}}$ on the frame bundle, yielding an intrinsic representation of the diffusion process that respects the manifold's geometry through horizontal flows.

**Definition 2.1** (Horizontal Diffusion Models). *Let $\boldsymbol{\varepsilon} = \{\varepsilon_1, \ldots, \varepsilon_d\}$ denote the standard basis of $\mathbb{R}^d$. The horizontal diffusion models are the system of forward-reverse Stratonovich SDEs on the orthonormal frame bundle $\mathcal{O}(\mathcal{M})$, with local coordinate $\mathbf{U}_t = (x_t, e_t) \in \mathcal{O}(\mathcal{M})$, evolves according to*

$$
\begin{bmatrix} d\mathbf{U}_t \\ d\mathbf{U}_s \end{bmatrix} = \begin{bmatrix} -\frac{1}{2}\nabla^{\mathrm{Hor}}\Phi_t(\mathbf{U}_t) \\ \frac{1}{2}\nabla^{\mathrm{Hor}}\Phi_s(\mathbf{U}_s) + s^{\mathrm{Hor}}(s, \mathbf{U}_s) \end{bmatrix} \begin{bmatrix} dt \\ ds \end{bmatrix}^{\mathsf{T}} + \begin{bmatrix} H_a(\mathbf{U}_t) \\ H_a(\mathbf{U}_s) \end{bmatrix} \begin{bmatrix} \circ \, dW_t^a \\ \circ \, dW_s^a \end{bmatrix}^{\mathsf{T}}, \tag{4}
$$

*Then, the **horizontal score function** $s^{\mathrm{Hor}}$ is defined as time-indexed horizontal vector fields*

$$
s^{\mathrm{Hor}}(t, \mathbf{U}_t) := \nabla^{\mathrm{Hor}} \log \rho_t^{\mathrm{Hor}}(\mathbf{U}_t), \tag{5}
$$

*which lies in the horizontal distribution $\mathrm{Hor}_{\mathbf{U}_t} \subset T_{\mathbf{U}_t}\mathcal{O}(\mathcal{M})$ at each point $\mathbf{U}_t \in \mathcal{O}(\mathcal{M})$. Here, $\rho_t^{\mathrm{Hor}}$ denotes a Markovian probability distribution with respect to forward frame dynamics, i.e., $\mathbf{U}_t \sim \rho_t^{\mathrm{Hor}}$.*

In this paper, we define *horizontal diffusion models* as pairs of noise and data generative processes $(\mathbf{U}_t, \mathbf{U}_s)$ that extend the Riemannian formulation Bortoli et al. (2022) while preserving coordinate invariance without requiring global charts or extrinsic parameterizations. The overall mechanism follows the general principle of score-based diffusion models. The forward process first injects noise until reaching a stationary distribution $\mathbf{U}_\infty \sim \rho_\infty^{\mathrm{Hor}}$, whereas the reverse process progressively removes noise to recover the original initial distribution $\mathbf{U}_0 \sim \rho_0^{\mathrm{Hor}}$. The canonical projection $\pi : \mathcal{O}(\mathcal{M}) \to \mathcal{M}$, defined by $\pi(x, e) = x$ for $(x, e) \in \mathcal{O}(\mathcal{M})$, ensures that the data distribution is obtained by projecting the reversed horizontal diffusion trajectory, $\rho_0^{\mathcal{M}} = \pi_\# \rho_{s \to 0}^{\mathrm{Hor}}$. In this context, the ultimate goal of score-based generative modeling is to approximate the true score function with neural networks, *i.e.*, $s_\theta^{\mathrm{Hor}} \approx s^{\mathrm{Hor}}$ to reconstruct data distribution $\rho_0^{\mathcal{M}}$. Before proceeding, one may naturally ask how the probability distribution $\rho_t^{\mathrm{Hor}}$ arises in this setting and how it relates to the data distribution $\rho^{\mathcal{M}}$ on the manifold. Proposition D.4 and Corollary D.5 establish the existence of the marginal distribution $\rho_t^{\mathrm{Hor}} : \mathcal{O}(\mathcal{M}) \to \mathbb{R}_+$ on the frame bundle. It is defined via the joint evolution of the initial frame $\mathbf{U}_0$ and the lifted process $\mathbf{U}_t \in \mathcal{O}(\mathcal{M})$, as follows:

$$
\rho_t^{\mathrm{Hor}}(\mathbf{U}) = \int_{\mathcal{O}(\mathcal{M})} p_t^{\mathrm{Hor}}(\mathbf{U} \mid \mathbf{U}_0) \rho_0(\mathbf{U}_0) d\lambda_0(\mathbf{U}_0) \xrightarrow{t \to \infty} \exp\left(-\Phi(\mathbf{U})\right), \tag{6}
$$

where $\lambda_0$ is the volume measure on $\mathcal{O}(\mathcal{M})$, $\rho_0$ is the initial distribution, and $p_t^{\mathrm{Hor}}(\mathbf{U}|\mathbf{U}_0)$ is the conditional transition kernel of the lifted process. While our horizontal diffusion models are defined on the orthonormal frame bundle $\mathcal{O}(\mathcal{M})$, the ultimate objective is to model distributions over the base manifold $\mathcal{M}$. The lifted process allows us to define the marginal manifold distribution by pushing forward the frame bundle distribution: $\mathbf{X}_t \sim \rho_t^{\mathcal{M}}(x) := \int_{\pi^{-1}(x)} \rho_t^{\mathrm{Hor}}(U) d\mathrm{Haar}(\mathbf{U})$, where $\mathrm{Haar}(\mathbf{U})$ is the Haar measure on the fiber $\pi^{-1}(x) \simeq O(d)$.

**Horizontal Score-Matching.** With the horizontal diffusion processes in place, we next introduce the *score-matching principle* as the foundation of our learning objective. We aim to develop a score-matching framework formulated directly on the frame bundle as an extended counterpart of conventional approaches. In this setting, the learnable score field $s^{\mathrm{Hor}}$ is constrained to lie in the horizontal distribution, and the discrepancy between forward and reverse drifts is evaluated within this structure. To formalize, we introduce a geometry-aware divergence functional called a *horizontal Kullback–Leibler divergence* as follows:

**Proposition 2.2** (Horizontal KL Divergence). *Let $b_t(\mathbf{U}_t)$ and $\tilde{b}_t(\mathbf{U}_t^\theta)$ be horizontal drift vector fields on the frame bundle $\mathcal{O}(M)$, with path laws $\mathbf{U}_t \sim \boldsymbol{\nu}_b$ and $\mathbf{U}_t^\theta \sim \boldsymbol{\nu}_{\tilde{b}}$ representing distinct horizontal diffusions. Let $\| \cdot \|_H$ denote the Sasaki metric norm on the horizontal subbundle $H_U$, and $\Gamma^{\mathcal{O}(M)}$ the parallel transport operator. Given the projection $\mathbf{P}_{\mathrm{Hor}}(Z) := Z - \omega(Z)^\sharp \cdot U,^a$ which maps tangent vectors to the horizontal subspace $\mathrm{Hor}_U \subset T_U \mathcal{O}(M)$, the horizontal KL divergence is given by*

$$
\mathbf{KL}_{\mathrm{Hor}}[\boldsymbol{\nu}_{\tilde{b}} \| \boldsymbol{\nu}_b] = \frac{1}{2} \mathbb{E}_{\boldsymbol{\nu}_{\tilde{b}}} \left[ \int_0^T \|\mathbf{P}_{\mathrm{Hor}}(\Delta_t)\|_H^2 \, dt \right], \quad \Delta_t := \Gamma^{\mathcal{O}(M)}_{\mathbf{U}_t^\theta \to \mathbf{U}_t} \left( \tilde{b}_t(\mathbf{U}_t^\theta) \right) - b_t(\mathbf{U}_t). \tag{7}
$$

---

[a]Here, $\omega(Z)^\sharp$ denotes the fundamental vertical vector field associated with the Lie algebra element $\omega(Z) \in \mathfrak{o}(d)$. We follow the same notation suggested in Kobayashi & Nomizu (1963b).

The projection operator $\mathbf{P}_{\mathrm{Hor}}$ serves to constrain all vector fields within the horizontal distribution during divergence computation, thereby guaranteeing both geometric consistency and invariance under orthogonal transformations. However, despite the rigorous theoretical basis provided by Proposition 2.2 for defining the horizontal KL divergence, practical numerical implementation remains challenging, primarily due to the necessity of path-wise parallel transport computations. To address this issue, we introduce a computationally feasible alternative leveraging gauge-equivariance, thereby enabling efficient adaptation of standard Euclidean neural network architectures.

**Horizontal Parameterization Trick.** To address the computational issue in evaluating Horizontal KL divergence, we propose a simple yet effective detour. Rather than directly comparing vector fields on the frame bundle, we lift all geometric quantities via a horizontal lifting map $\Psi^1$, which maps Euclidean latent path $E_\bullet \in C([0,T], \mathbb{R}^d)$ to *gauge-equivariant quotient path space*:

$$\Psi : C([0,T], \mathbb{R}^d) \xrightarrow{H_e} C([0,T], \mathcal{O}(M))/\mathcal{G}, \quad \Psi(E_\bullet) := [U_\bullet] = \{U_\bullet \cdot g_\bullet \mid g_\bullet \in \mathcal{G}\}, \quad (8)$$

where $\mathcal{G} := C([0,T], O(d))$ denotes the group of time-dependent gauge transformations. This embedding ensures that geometric evaluations are performed at canonical representatives in the quotient space and eliminates the need for explicit parallel transport. We refer to this approach as a *parameterization trick*, where the gradient of potential and the horizontal score function can be parameterized as their euclidean counterparts:

**Proposition 2.3** (Horizontal Lift of Gradients and Scores). *Let* $\mathbf{E}_t \in \mathbb{R}^d$ *be the solution to Euclidean Stratonovich SDE, and let* $\mathbf{U}_t = (x_t, e_t) \in \mathcal{O}(\mathcal{M})$ *be its horizontal lift defined by* $d\mathbf{U}_t = H_{e_t}(\mathbf{U}_t)\big[\circ \, d\mathbf{E}_t\big]$. *Then, for every* $t \geq 0$,

$$\nabla^{\mathrm{Hor}}\Phi(\mathbf{U}_t) = H_{e_t}(\mathbf{U}_t)\big[\nabla\phi(\mathbf{E}_t)\big], \quad \nabla^{\mathrm{Hor}}\log\rho_t^{\mathrm{Hor}}(\mathbf{U}_t) = H_{e_t}(\mathbf{U}_t)\big[\nabla\log\rho_t(\mathbf{E}_t)\big], \quad (9)$$

$$s^{\mathrm{Hor}}(t,\mathbf{U}_t) := \nabla^{\mathrm{Hor}}\log\rho_t^{\mathrm{Hor}}(\mathbf{U}_t) = H_{e_t}(\mathbf{U}_t)\big[s(t,\mathbf{E}_t)\big] = \sum_{a=1}^{d} s^a(t,\mathbf{E}_t)H_a(\mathbf{U}_t). \quad (10)$$

The result follows directly from the definition of the quotient mapping $\Psi$. Since horizontal vector fields such as gradients and score functions are invariant under gauge transformations, their evaluation can be consistently performed on any element within the equivalence class. Proposition 2.3 formalizes this principle by demonstrating that the horizontal gradient and score at a lifted point $\mathbf{U}_t$ can be obtained by applying the horizontal lift operator $H_{e_t}(\mathbf{U}_t)[\cdot]$ to Euclidean counterparts at $\mathbf{E}_t$.

This construction enables a practical simplification of the geometric framework. By expressing all horizontal quantities in terms of the latent Euclidean process, the formulation avoids the need for explicit parallel transport or projection operations, which are otherwise computationally expensive and numerically sensitive. Consequently, the horizontal KL divergence introduced in equation 7 admits the following Euclidean-form objective, which remains faithful to the geometry $\mathcal{O}(\mathcal{M})$:

$$\mathbf{KL}_{\mathrm{Hor}}(\theta) := \mathbf{KL}_{\mathrm{Hor}}[\boldsymbol{\nu}_b \parallel \boldsymbol{\nu}_{\tilde{b}^\theta}] = \frac{1}{2}\int_0^T \sigma_t^2 \mathbb{E}_{\mathbf{E}_t \sim \rho_t}\big[\|s(t,\mathbf{E}_t) - s_\theta(t,\mathbf{E}_t)\|_E^2\big]\,dt, \quad (11)$$

Here, $b = \frac{1}{2}\nabla^{\mathrm{Hor}}\Phi + s^{\mathrm{Hor}}$ and $\tilde{b}^\theta = \frac{1}{2}\nabla^{\mathrm{Hor}}\Phi + s_\theta^{\mathrm{Hor}}$ denote the true (*i.e.*, $s^{\mathrm{Hor}}$) and model-induced (*i.e.*, $s_\theta^{\mathrm{Hor}}$) reverse vector fields along horizontal trajectories, respectively. This result marks a key advancement in manifold-based score modeling. It enables direct computation of the distribution in equation 6 using the lifted Euclidean score, bypassing the need for explicit transition distribution $\rho_t^{\mathrm{Hor}}$ based on spectral decomposition. Most importantly, the path discrepancy in equation 11 simplifies to a Euclidean score-matching loss, allowing the use of *standard Euclidean score networks* $s_\theta$ *without architectural changes*, while maintaining geometric consistency.

**Horizontal Ornstein-Uhlenbeck Bridge Process.** Extending the previously developed horizontal diffusion framework and associated score-matching methodology, we now provide a rigorous construction of the horizontal diffusion model by explicitly specifying both the potential function and the parameterization of the lifted dynamics. We construct the Euclidean potential function $\phi$ in equation 1, which characterizes latent Euclidean Ornstein-Uhlenbeck (OU) processes as follows:

$$\phi(t,E) = -\frac{\gamma_t}{\sigma_t^2}\|E\|^2 + \frac{2\lambda_t}{\sigma_t^2}\langle v_t, E\rangle, \quad d\mathbf{E}_t = [\lambda_t e_t^{-1}v_t - \gamma_t\mathbf{E}_t]\,dt + \sigma_t \circ dW_t, \quad \mathbf{E}_0 = 0, \quad (12)$$

---
[1]For a detailed proof and definition, please refer to Proposition D.1 in Appendix.

where the coefficients $(\gamma_t, \lambda_t, \sigma_t) > 0$ and the time-dependent vector $v_t \in \mathbb{R}^d$ are judiciously chosen to satisfy model requirements. Notably, this OU process admits a Gaussian transition density for the forward process, i.e., $\mathbf{E}_t \sim \mathcal{N}(0, \Sigma_t)$. Next, we lift this construction by harnessing the result in Proposition 2.3 to define the *horizontal Ornstein-Uhlenbeck bridge process* on the frame bundle:

$$d\mathbf{U}_t = \sum_a \left(2\gamma_t \varepsilon^a + \sigma_t^2 v_t^a\right) H_a(\mathbf{U}_t)dt + \sum_a \sigma_t H_a(\mathbf{U}_t) \circ dW_t^a, \quad \Sigma_t = \sigma_t^\top \sigma_t, \qquad (13)$$

where $H_a(\mathbf{U}_t) := H_{e_t}(\mathbf{U}_t)[\varepsilon^a]$ denote fundamental horizontal vector fields. We specifically design the coefficients such that $\gamma_t, \lambda_t \to 0$ with $\gamma_t \ll \lambda_t$. In this configuration, the drift components vanish, and the horizontal diffusion asymptotically reduces to pure horizontal Brownian motion. Under this design choice, the stationary distribution asymptotically converges to the uniform measure with respect to the Sasaki volume (*i.e.*, product measure of $\mathrm{Vol}_g$ and $\mathrm{Haar}_{O(d)}$):

$$d\rho_\infty^{\mathrm{Hor}}(x, e) = d\mathrm{Vol}_{\mathrm{Sas}}(x, e) = d\mathrm{Vol}_g(x) \otimes d\mathrm{Haar}_{O(d)}(e). \qquad (14)$$

The samples from stationary distribution can be efficiently generated by independently drawing base points from $d\mathrm{Vol}_g(x)$ and frames from $d\mathrm{Haar}_{O(d)}(e)$, as summarized in Alg. 1. To ensure that the reverse process drives samples toward the data distribution, we introduce the bridge vectors $v_t = e_t^{-1} \log_{x_t}(x_0)$, which act as time-dependent attraction terms directing the dynamics back to the initial state $x_0$. These vectors satisfy the zero-mean property $\mathbb{E}_{x_\bullet \sim \rho_\bullet^{\mathrm{Hor}}}[v_\bullet] = 0$, as established in Lemma C.3. As a result, the Euclidean score simplifies to $s(t, E) = -\Sigma_t E$ for some time-variant covariance matrix $\Sigma_t$, and the corresponding reverse horizontal dynamics are given by

$$d\mathbf{U}_s = -\sum_a \left(2\gamma_s \varepsilon^a + \sigma_s^2 v_s^a\right) H_a(\mathbf{U}_s)ds + s^{\mathrm{Hor}}(s, \mathbf{U}_s)ds + \sum_a \sigma_s H_a(\mathbf{U}_s) \circ dW_s^a, \qquad (15)$$

where the horizontal score function takes the form $s^{\mathrm{Hor}}(s, \mathbf{U}_s) := -\sum_{a,b}[\Sigma_s]_{ab}\varepsilon_s^b H_a(\mathbf{U}_s)$. In practice, the logarithmic map $\log_{x_t}(x_0)$ is approximated by its second-order expansion involving as

$$\log_{x_t}(x_0) \approx -\Delta_x + \frac{1}{2}\Gamma(x_t)[\Delta_x, \Delta_x], \quad \left[\Gamma(x_t)[\Delta_x, \Delta_x]\right]^i = \sum_{j,k} \Gamma_{jk}^i(x_t)(\Delta_x)^j(\Delta_x)^k. \qquad (16)$$

where $\Delta_x := x_t - x_0$ denotes the discrepancy between coordinates. In sampling the reverse horizontal dynamics, we proceeds as follows: At each reverse step, we compute the bridge vector $b_s$ using the above second-order approximation of $\Gamma$ to evaluate the score function $s^{\mathrm{Hor}}$, and apply the update rule specified in the reverse SDE. Proposition D.2 shows that the reverse dynamics in equation 15 reconstruct the original data distribution $\rho_0^{\mathcal{M}}$. Please refer to Alg. 5 for a detailed algorithm.

**Training Neural Networks.** To instantiate the proposed horizontal diffusion framework, we parameterize the learnable score field via neural networks lifted to the frame bundle. Specifically, the horizontal score is defined as $\mathrm{NN}^{\mathrm{Hor}}(t, \mathbf{U}; \theta) = \sum_a \mathrm{NN}(t, \mathbf{E}^a; \theta)H_a$, where $\mathrm{NN}(t, E)$ is a standard Euclidean neural network and $H_a$ are canonical horizontal vector fields. The network output decomposes as $\mathrm{NN}(t, E) = s^\theta(t, E) + v(t, \theta)$, with $v(t, \theta)$ representing an auxiliary drift aligned with the data-driven bridge vector. The training objective is

$$\mathcal{J}(\theta) = \mathbf{KL}_{\mathrm{Hor}}(\theta) + \mathbb{E}_{x_0, x_t} \|v(t, \theta) - v_t(x_t, x_0)\|^2, \qquad (17)$$

where $\mathbf{KL}_{\mathrm{Hor}}(\theta)$ is the horizontal KL divergence from Proposition 2.2, and the second term regularizes the auxiliary drift toward the analytic bridge vector. This composite objective enforces geometric consistency while enabling efficient training. Importantly, as formalized below, the built-in gauge equivariance of the horizontal score field induces an intrinsic symmetry that streamlines optimization.

**Gauge Equivariance $\mathcal{G}$.** In the frame-connection geometry, *gauge equivariance* refers to the invariance of geometric quantities under fiber-wise orthogonal transformations. Let us define the map $\mathcal{R}_h : \mathcal{O}(M) \to \mathcal{O}(M)$ as $\mathcal{R}_h(U) := U \cdot h$. A horizontal vector field $V : \mathcal{O}(M) \to T\mathcal{O}(M)$ is said to be *gauge-equivariant* if it satisfies $V(U \cdot h) = (\mathcal{R}_h)_* V(U)$ for all $h \in O(d)$, where $(\mathcal{R}_h)_* : T_U \mathcal{O}(M) \to T_{U \cdot h}\mathcal{O}(M)$ is the pushforward of the right action $\mathcal{R}_h$. Importantly, this geometric property is not only intrinsic to the frame bundle structure, but also plays a pivotal role in the design of our learning framework. Gauge equivariance guarantees geometric consistency, enables modeling on the quotient space $\mathcal{O}(M)/O(d)$, and enhances generalization while maintaining full compatibility with standard Euclidean parameterizations. Corollary 2.4 shows that the gauge

equivariance is naturally incorporated into our model via equation 8 by construction, where the horizontal score function is defined as the contraction of a Euclidean score with canonical horizontal directions.

**Proposition 2.4** (Gauge Equivariance of horizontal score function). *The proposed horizontal score function is gauge-equivariant in the sense that $s^{\mathrm{Hor}}(t, U \cdot h) = (\mathcal{R}_h)_* s^{\mathrm{Hor}}(t, U)$ for all $h \in O(d)$.*

**Corollary 2.5** (Isometry–equivariance of the base score). *Let* $\mathrm{Isom} : \mathcal{M} \to \mathcal{M}$ *be any isometry on* $\mathcal{M}$. *Then, the manifold score function satisfies the equivariance:* $s^{\mathcal{M}}\big(t, \mathrm{Isom}(x)\big) = \mathrm{Isom}_* s^{\mathcal{M}}(t, x)$, *where* $\mathrm{Isom}_{*x} : T_x \mathcal{M} \to T_{\mathrm{Isom}(x)}\mathcal{M}$ *denotes the differential (push-forward) of* $\mathrm{Isom}$ *at the point* $x$.

Equivariance of score-based diffusion models in the Euclidean setting has been extensively studied to ensure model robustness and efficiency under symmetry transformations Köhler et al. (2020); Hoogeboom et al. (2022). Proposition 2.4 and Corollary 2.5 extend this essential notion of equivariance to geometric manifolds, thus addressing a critical limitation in existing manifold-based diffusion frameworks. Unlike previous methods De Bortoli et al. (2022a); Thornton et al. (2022); Diepeveen et al. (2025a), which typically lack intrinsic compatibility with fiber symmetries (*i.e.*, gauge-inequivariant) and depend heavily on global geometric constructions, our proposed formulation inherently respects fiber symmetries, resulting in a more principled and geometrically consistent approach.

**The Price of Horizontal Lift.** Despite its effectiveness in facilitating generative modeling on complex manifolds via the lifting of Euclidean scores to the frame bundle, our methodology is subject to inherent geometric distortions. The subsequent theorem rigorously characterizes the *worst-case horizontal KL divergence* through a uniform generalization bound and elucidates how both the expressivity of the neural parameterization and the intrinsic curvature of the manifold fundamentally constrain the statistical performance of our approach.

**Theorem 2.6** (informal). *Assume the neural score networks $\theta \in \Theta$ are parameterized as $L$-layer, width $W$ feedforward ReLU networks with spectral norm bounds. Assume the underlying manifold $\mathcal{M}$ has bounded sectional curvature $\kappa_{\max}$ and diameter $D_{\mathcal{M}} := \mathrm{Diam}(\mathcal{M})$. Let $n$ be the number of samples per time in evaluating objective function, $M$ the number of time steps. Then for any $\delta \in (0, 1)$, with probability at least $1 - \delta$, the following holds:*

$$\sup_{\theta \in \Theta} \mathbf{KL}_{\mathrm{Hor}}(\theta) \leq \frac{6 C_{\mathcal{M}}(\kappa_{\max}, D_{\mathcal{M}}) \cdot C_{\mathrm{NN}}(L, W)}{n^{1/4}} + \sqrt{\frac{C_{\mathrm{SubG}}^2(\sigma_{t_m}, \Sigma) \log(1/\delta)}{4nM}}. \quad (18)$$

*where $C_{\mathcal{M}}$ encodes the geometric dependency on the manifold, $C_{\mathrm{NN}}$ depends on the neural score network architecture and norm constraints, and $C_{\mathrm{SubG}}$ is the sub-Gaussian complexity constant.*

Here, $C_{\mathcal{M}}$ is proportional to the maximal sectional curvature $\kappa_{\max}$, indicating that the generalization error bound increases as the manifold becomes more curved or non-flat. Consequently, to achieve accurate generalization on highly curved manifolds, a sufficiently large number of samples $n$ is required to guarantee statistical convergence of the KL bound. This highlights the increased sample complexity and learning difficulty associated with nontrivial geometric structure. Please refer to equation 89 in the appendix for the explicit expressions of all constants (*i.e.*, $C_{\mathcal{M}}, C_{\mathrm{NN}}, C_{\mathrm{SubG}}$).

## 3 RELATED WORK

A broad range of methods has emerged to address the challenge of generative modeling on curved manifolds, notably through the use of Riemannian flows, score-based diffusion models, and manifold-aware normalizing flows. Brehmer & Cranmer (2020) developed flows designed for simultaneous manifold learning and density estimation, significantly advancing manifold-aware generative modeling. Further contributions Poli et al. (2020) introduced Riemannian continuous normalizing flows. De Bortoli et al. (2022b) expanded this direction with Riemannian score-based generative modeling, combining diffusion models and Riemannian geometry. Chen & Lipman (2024) generalized flow-matching on Riemannian manifolds. Huang et al. (2022) proposed Riemannian diffusion models tailored for geometric data, and Thornton et al. (2022) explored Riemannian diffusion Schrödinger bridges. Jo et al. (2023) developed generative models via mixtures of Riemannian bridge processes, introducing a flexible approach to manifold-constrained generative modeling. Recently, Lou et al. (2023) expanded these concepts to Riemannian manifolds, demonstrating how scalable diffusion

**Figure 1: Density Estimation on the torus and catenoid surfaces**. For each surface, the ground truth distribution (left) and the distribution generated by ours (right) are shown. Both surfaces are equipped with a Gaussian mixture, and the generated samples closely match the true multi-modal density.

| Model | Density | Qubit ($\mathbb{CP}^1$) | Sphere ($\mathbb{S}^d$) | Torus ($\mathbb{T}^d$) | Catenoid$^\dagger$ | Enner$^\dagger$ | Dupin$^\dagger$ |
|---|---|---|---|---|---|---|---|
| Oracle | $\bar{\rho}^{\mathcal{M}}$ | 0.032 / 0.136 | 0.045 / 0.158 | 0.097 / 0.345 | 0.055 / 0.385 | 0.101 / 0.323 | 0.112 / 0.710 |
| Riem SGM | $\rho^{\mathcal{M}}$ | 0.101 / 0.355 | 0.159 / 0.569 | 0.213 / 0.711 | 0.421 / 1.552 | 0.534 / 1.764 | 0.811 / 2.022 |
| Riem CFM | $\rho^{\mathcal{M}}$ | 0.083 / 0.303 | 0.141 / 0.488 | 0.205 / 0.667 | 0.379 / 1.426 | 0.497 / 1.621 | 0.768 / 1.884 |
| Bundle NF | $\rho^{P(\mathcal{M})}$ | – | – | 0.196 / 0.635 | – | – | 0.746 / 1.842 |
| Scaling | $\rho^{\mathcal{M}}$ | 0.085 / 0.244 | 0.125 / 0.525 | 0.211 / 0.606 | 0.317 / 1.356 | 0.530 / 1.466 | 0.803 / 1.762 |
| Pull NF | $\rho^{\mathcal{M}}$ | 0.138 / 0.431 | 0.192 / 0.714 | 0.284 / 0.918 | 0.531 / 2.051 | 0.677 / 2.264 | 1.082 / 2.699 |
| HDM (Ours) | $\rho^{\text{Hor}}$ | **0.075 / 0.235** | **0.079 / 0.268** | **0.102 / 0.351** | **0.161 / 0.721** | **0.211 / 0.654** | **0.327 / 0.959** |

**Table 2: Comparison of Density Estimation on Parametric Surfaces and Quantum State Manifold** with sliced 2-Wasserstein distance. Lower values indicate better performance. Prior results on Catenoid, Enner, and Dupin marked with a dagger symbol$^\dagger$ are highlighted in light red. Further details are provided in Section E.

models can facilitate generative modeling on manifolds by leveraging Riemannian score matching. Courts & Kvinge (2022) presented generative models on bundle networks leveraging fiber bundles.

## 4 EXPERIMENTS

In this section, we empirically evaluate our framework on a diverse collection of manifolds, encompassing both analytically defined surfaces and real-world scientific domains. Specifically, our experiments span (i) family of parametric surfaces and (ii) complex real-world geometric structures parameterized by spherical harmonics. In all experiments, the quality of the generated samples is evaluated using the (maximal) sliced 2-Wasserstein distance (*i.e.*, $\mathcal{SW}_2, \mathcal{MSW}_2$), which measures the discrepancy between the empirical and generated distributions. Section E contains the implementation details on the experimental setup and **additional experiments**, **ablation study**.

**Synthetic Dataset: Parametric Surfaces.** As our first benchmark for generative modeling on curved geometries, we construct synthetic datasets based on five analytically defined parametric surfaces: *Sphere*, *Torus*, *Catenoid*, *Enner*, and *Dupin* where each surface is described by an explicit embedded parametric equation. For the more intricate cases, the embeddings admit compact forms such as Catenoid $(u,v) \mapsto (\cosh v \cos u, \cosh v \sin u, v)$, Enner $(u,v) \mapsto (\cos u \cos v, \cos u \sin v, \sin u)$, and Dupin $(u,v) \mapsto \left(\frac{\cos u}{1+\sin v}, \frac{\sin u}{1+\sin v}, \frac{\cos v}{1+\sin v}\right)$. Notably, with the exception of the sphere or flat torus, explicit analytic forms for the probability density and geodesic equations are generally infeasible for these parametric surfaces. For each surface, samples are drawn from a Riemannian GMM $p_{\mathcal{M}}(x) = \sum_{k=1}^{K} \mathfrak{a}_k \mathcal{N}_{\mathcal{M}}(x; \mu_k, \Sigma_k)$ where $\sum_k \mathfrak{a}_k = 1$ and $\mathcal{N}_{\mathcal{M}}$ denotes the Gaussian distribution defined on the manifold with Gaussian parameters $(\mu_k, \Sigma_k)$.

**Real-world Dataset: Quantum Qubit.** Utilizing the QDataSet Perrier et al. (2022), which provides experimentally realistic quantum-control simulations, we construct datasets explicitly supported on the qubit manifold $\mathbb{CP}^1$. Importantly, the raw QDataSet does not directly supply Bloch vectors, but rather 18-dimensional expectation values of an informationally complete operator set. To recover the underlying physical states, we perform a linear inversion tomography step: given measurement operators $\{V_k\}_{k=1}^{18}$ and observed expectations $E_k = \text{Tr}(\rho V_k)$, we solve an overdetermined least-squares system to estimate the Bloch vector $r = (r_x, r_y, r_z)$, enforcing $\|r\| \leq 1$. This procedure yields the corresponding density matrices $\rho = \frac{1}{2}(I + r_x \sigma_x + r_y \sigma_y + r_z \sigma_z)$, thus projecting the QDataSet outputs onto $\mathbb{CP}^1 \cong \mathbb{S}^2$.

Fig. 1 and Table 2 summarize both the qualitative and quantitative generative modeling results on these benchmark surfaces. $\bar{\rho}^{\mathcal{M}}$ is the oracle reference, which measures the sliced Wasserstein distance between the ground-truth and reconstructed results using the theoretically exact reverse dynamics. Existing geometric methods lack native support for complex surfaces, requiring additional geometric

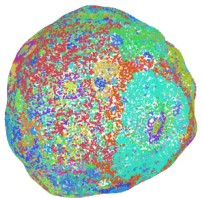 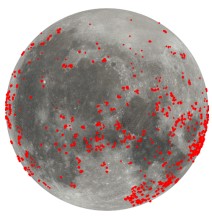 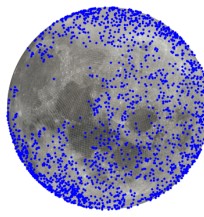

**Figure 3: Density Estimation of Craters on Asteroid and Moon**. *(Left)* Reconstructed crater density on the Eros asteroid. *(Second)* True geological label-specific crater densities on the lunar surface. *(Right two)* Density estimation results for IM and NC regions on the Moon, demonstrating label-conditional modeling performance.

| Dataset | Riem SGM | Riem CFM | Scaling NF | Pull NF | HDM (Ours) |
|---|---|---|---|---|---|
| Density | $\rho^{\mathbb{S}^2 \times \mathbb{R}}$ | $\rho^{\mathbb{S}^2 \times \mathbb{R}}$ | $\rho^{\mathbb{S}^2 \times \mathbb{R}}$ | $\rho^{\mathbb{S}^2 \times \mathbb{R}}$ | $\rho^{\text{Hor}}$ |
| Eros | 0.691 / 1.104 | 0.707 / 1.198 | 0.672 / 1.078 | 0.689 / 1.170 | **0.312 / 0.470** |
| Moon (IM) | 1.110 / 1.754 | 1.200 / 1.896 | 1.258 / 2.015 | - | **0.512 / 0.754** |
| Moon (NC) | 1.201 / 1.798 | 1.116 / 1.675 | 1.285 / 2.197 | - | **0.489 / 0.726** |
| Anatomy | 0.866 / 1.389 | 0.920 / 1.472 | 0.825 / 1.405 | - | **0.414 / 0.606** |

**Table 3: Density Estimation on Astronomy and Anatomy Datasets.** Each entry reports sliced 2-Wasserstein distance (left) and maximal sliced 2-Wasserstein distance (right). Lower values indicate better performance.

projections to canonical manifolds (Sec E). While such projections inherently break isometry and introduce additional distortion, it limits their performance. By contrast, our framework operates on the frame bundle, eliminating the need for projection, achieving consistently superior results.

**Real-world Dataset: Spherical Harmonics.** A widely used technique for representing the geometry of closed surfaces is the expansion of the radial function in spherical harmonics. In this approach, parametrized surfaces are given as $r(\theta, \varphi) = r_0(\theta, \varphi) + \sum_{l,m} c_{l,m} Y_{l,m}(\theta, \varphi)$, where $(\theta, \varphi)$ are spherical angular coordinates, $Y_{l,m}$ denote spherical harmonics of degree $l$ and order $m$, and $c_{l,m}$ are scalar coefficients. Upon this structure, the first experiment considers modeling density on the surfaces of various celestial bodies, exemplified by the *Moon* and *asteroids* (*i.e.*, Eros).

Figure 3 illustrates the modeling of crater densities on curved celestial surfaces. The first image shows the density approximation of craters on the asteroid Eros, generated by our model. The second image visualizes the entire set of labeled craters on the Moon dataset, where each crater is colored according to its class and plotted with height information. The last two images present generated samples for the IM and NC crater labels on the Moon surface. In the second application, we consider the task of modeling tumor distributions on human anatomical surfaces (*i.e.*, *knee*). As summarized in Table 3 and Figures 3 and 2, our experimental results demonstrate that the

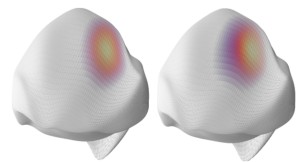

**Figure 2:** True (left) and generated (right) tumor probability on human anatomy (*i.e.*, Knee).

proposed method consistently achieves high-fidelity density estimation on a diverse set of surfaces, ranging from celestial bodies (*i.e.*, asteroid Eros and lunar craters) to human anatomical surfaces (*i.e.*, knee tumors). While existing methods typically rely on embedding data into the ambient product space $\mathbb{S}^2 \times \mathbb{R}$ and subsequently projecting it back onto the target manifold, induced projection distortions limit reconstruction quality. In contrast, our framework operates natively on the frame bundle, preserving geometric integrity without projection-induced distortions. The consistently lower sliced Wasserstein distances achieved by our approach underscore its superior capability to model intricate data distributions across general geometry.

## 5 CONCLUSION

In this work, we introduced Horizontal Diffusion Models, a novel framework for generative modeling on general Riemannian manifolds leveraging orthonormal frame bundles. By horizontally lifting standard Euclidean diffusion processes to the frame bundle, our approach maintains geometric consistency without explicit manifold-specific computations. Experiments across synthetic and real-world manifold datasets demonstrate superior performance, validating our model's scalability and geometric fidelity. This establishes a robust, geometry-aware pathway for future generative modeling research on complex, curved manifolds.

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

# A   NOTATION TABLE

For the reader's convenience, we provide a summary of the primary symbols and notation employed in this work. The table below serves as a quick reference for all key terms and variables.

| Symbol | Meaning |
|---|---|
| $M, g$ | $d$-dimensional Riemannian manifold and its metric |
| $\pi : \mathcal{O}(M) \to M$ | Orthonormal frame bundle (principal O$(d)$-bundle) |
| $(x^i, e^i_a)$ | Local coordinates on $\mathcal{O}(M)$; $e_a$ is the $a$-th frame vector at $x$ |
| $\omega$ | Levi–Civita connection 1-form on $\mathcal{O}(M)$ |
| $\Omega = d\omega + \frac{1}{2}[\omega, \omega]$ | Curvature 2-form of $\omega$ |
| $H_e[w]$ | Horizontal lift of $w \in \mathbb{R}^d$ at $e$, giving a vector in $T_U\mathcal{O}(M)$ |
| $\mathbf{U}_t, \mathbf{E}_t$ | Horizontal diffusion on $\mathcal{O}(M)$ and latent Euclidean process in $\mathbb{R}^d$ |
| $\nu_t = \mathcal{L}(\mathbf{U}_t, \mathbf{E}_t)$ | Joint law of $(\mathbf{U}_t, \mathbf{E}_t)$ on $\mathcal{O}(M) \times \mathbb{R}^d$ |
| $\rho_t$ | Density of $\mathbf{E}_t$ on $\mathbb{R}^d$ |
| $\rho_t^{\mathrm{Hor}}(U)$ | Density of $\mathbf{U}_t$ on $\mathcal{O}(M)$ (Sasaki volume) |
| $\rho_t^M(x) = \pi_\# \rho_t^{\mathrm{Hor}}$ | Induced marginal on $M$ |
| $s_\theta^{\mathrm{Hor}}(t, U)$ | Learnable horizontal score field on $\mathcal{O}(M)$ |
| $s_*, s$ | True Euclidean score functions $\nabla \log \rho_t$ |
| $s_*^{\mathcal{M}}, s^{\mathcal{M}}$ | True manifold score functions $\nabla^{\mathcal{M}} \log \rho_t^{\mathcal{M}}$ |
| $s_*^{\mathrm{Hor}}, s^{\mathrm{Hor}}$ | True horizontal score functions $\nabla^{\mathrm{Hor}} \log \rho_t^{\mathrm{Hor}}$ |
| $\nu_{[0,T]}, \tilde{\nu}_{[0,T]}$ | Path laws of forward / reverse horizontal SDEs on $[0, T]$ |
| $\mathrm{Hor}_U, \mathrm{Ver}_U$ | Horizontal and vertical subspaces of $T_U\mathcal{O}(M)$ |
| $L_t^{\mathbf{E}}$ | Generator of horizontal diffusion conditioned on latent path $\mathbf{E}_{[0,t]}$ |
| $p_t^{\mathbf{E}}(U \mid U_0)$ | Conditional transition density on $\mathcal{O}(M)$ given $\mathbf{E}_{[0,t]}$ |
| $\nabla^{\mathrm{Hor}}$ | Horizontal gradient; $\nabla^{\mathrm{Hor}} f = P_{\mathrm{Hor}}(\nabla f)$ |
| $P_{\mathrm{Hor}}$ | Projection $T\mathcal{O}(M) \to H$ onto horizontal subbundle |
| $\mathrm{KL}^{\mathrm{Hor}}$ | Geometry-aware horizontal Kullback–Leibler divergence |
| $W_t, \tilde{W}_s := W_s$ | Forward and reverse $\mathbb{R}^d$ Brownian motions |
| $\sigma_t^{(a)} H_a$ | Horizontal noise field in lifted SDE |
| $\Gamma_{ij}^k, R_{irs}^k$ | Christoffel symbols and Riemann curvature components of $(M, g)$ |
| $\|\cdot\|_g, \langle \cdot, \cdot \rangle_g$ | Norm / inner product induced by $g$ (Sasaki metric on $H$) |
| $\delta_U$ | Dirac measure at $U \in \mathcal{O}(M)$ |
| $\mathrm{Law}_{\mathbb{P}}(X)$ | Distribution of $X$ under probability measure $\mathbb{P}$ |
| $T_U\mathcal{O}(M)$ | Tangent space of $\mathcal{O}(M)$ at $U$ |

**Table 4:** Summary of notation used throughout the paper.

# B  TECHNICAL COMPARISON WITH EXISTING WORKS

**Numerical Computation of $\rho_t^{\mathcal{M}}$.** A principal technique in classical score-based generative modeling on manifolds is the spectral decomposition of the Laplace–Beltrami operator. Given a compact Riemannian manifold $\mathcal{M}$, the Laplace–Beltrami operator $\Delta_{\mathcal{M}}$ admits a discrete set of eigenpairs $\{(\lambda_k, \rho_k)\}_{k=0}^{\infty}$ such that

$$\Delta_{\mathcal{M}}\rho_k = \lambda_k \rho_k, \qquad \rho^{\mathcal{M}}(x) = \sum_{k=0}^{\infty} c_k \rho_k(x), \qquad k = 0, 1, 2, \ldots,$$

with the eigenfunctions $\{\rho_k\}$ forming an orthonormal basis of $L^2(\mathcal{M})$. Existing methods approximate probability densities or construct kernels on the manifold by expanding density $\rho^{\mathcal{M}}$ in spectral basis.

This facilitates analytic calculation of heat kernels and allows the construction of diffusion models via explicit spectral representations. However, the effectiveness of this approach is fundamentally limited to manifolds where the spectral data can be computed in closed form. For general, non-symmetric, or high-curvature manifolds, such explicit decomposition is unavailable or infeasible. Therefore, these methods struggle to scale to arbitrary geometric domains. In contrast, our framework obviates the need for direct spectral decomposition by employing horizontal diffusion processes on the frame bundle. In particular, as established in Proposition 2.3, the horizontal lifting of the Euclidean score function enables an indirect yet scalable mechanism for approximating the data score function on the manifold, irrespective of the analytic tractability or the spectral properties of the underlying space.

**Neural Network Architecture.** Conventional manifold-based diffusion models often require geometry-specific neural network architectures, with model design and parameterization tailored individually to the underlying manifold for spheres, tori, or more general curved spaces. In contrast, our framework employs a single, unified neural network architecture across all experiments, irrespective of the specific manifold geometry. This universality is made possible by the use of the horizontal lift, which intrinsically adapts the representation of the score function to the manifold structure at hand. As a result, our approach achieves geometric adaptability without necessitating bespoke neural network designs for each manifold, enabling a consistent and scalable modeling paradigm.

**Geometric Operations.** A central feature of our approach is the definition of data dynamics through frame rotations on the orthonormal frame bundle, thereby intrinsically modeling geodesic flows of the underlying manifold. This construction is rooted in Elworthy's stochastic development Elworthy (1988), wherein the evolution of a point on the manifold is governed jointly with its moving frame.Mathematically, the evolution of the frame $e_t$ along a trajectory $x_t$ is described by the stochastic differential equation:

$$de_t = -\sum_{i,j,k} \Gamma_{ij}^k(x_t)\, e_t^j \circ dx_t^i\, E_k,$$

where $\Gamma_{ij}^k(x_t)$ are the Christoffel symbols of the Levi-Civita connection, and $E_k$ denotes the standard basis in $\mathbb{R}^d$. This formulation ensures that the frame is parallel transported along the curve $x_t$, faithfully encoding the connection-induced geometry.

Through this coupled evolution of position and frame, our method is able to realize geodesic and stochastic flows on the manifold in an intrinsic, coordinate-free manner. As a consequence, we are able to model data dynamics without requiring access to global distance functions, explicit geodesic computations. Instead, all essential geometric operations are performed intrinsically via the connection and horizontal lift, emphasizing the mathematical and computational scalability of our framework for generative modeling on general manifolds. In contrast, the heavy reliance of most existing methods on global geometric operations critically limits their practical scalability and application to broader classes of manifolds.

## C   LEMMAS

**Lemma C.1.** *Let $H_e$ be the horizontal lift operator defined for $w \in \mathbb{R}^d$, $e \in \mathcal{O}_x(\mathcal{M})$. Then, for the differentiable test function $f \in C^\infty(\mathcal{O}(\mathcal{M}))$, this operator can be written in local coordinate as follows:*

$$H_e(U)[w]f = w^a e_a^i \frac{\partial f}{\partial x^i} - w^a \Gamma_{ij}^k(x) e_a^j \frac{\partial f}{\partial e_i^k}, \tag{19}$$

*where $\mathbf{U} = (x, e) \in \mathcal{O}(\mathcal{M})$, and $\Gamma_{ij}^k(x)$ are the Christoffel symbols of the Levi-Civita connection at $x \in \mathcal{M}$. Then, the map $H_e : T_w\mathbb{R}^d \to T_U\mathcal{O}(\mathcal{M})$ is linear for all $w \in \mathbb{R}^d$.*

*Proof.* To verify linearity, it suffices to evaluate the action of the horizontal lift $H_e(U)[w]$ on an arbitrary smooth test function $f \in C^\infty(\mathcal{O}(\mathcal{M}))$ and confirm that it behaves linearly in $w \in \mathbb{R}^d$. Let $w_1, w_2 \in \mathbb{R}^d$ be arbitrary vectors, and $\lambda \in \mathbb{R}$ be a scalar. We consider the expression for $H_e[U][w_1 + \lambda w_2]f$, and expand it using the definition in equation 19:

$$H_e(U)[w_1 + \lambda w_2]f = (w_1^a + \lambda w_2^a)e_a^i \frac{\partial f}{\partial x^i} - (w_1^a + \lambda w_2^a)\Gamma_{ij}^k(x)e_a^j \frac{\partial f}{\partial e_i^k}$$

$$= w_1^a e_a^i \frac{\partial f}{\partial x^i} - w_1^a \Gamma_{ij}^k(x)e_a^j \frac{\partial f}{\partial e_i^k}$$

$$+ \lambda \left( w_2^a e_a^i \frac{\partial f}{\partial x^i} - w_2^a \Gamma_{ij}^k(x)e_a^j \frac{\partial f}{\partial e_i^k} \right)$$

$$= H_e[U][w_1]f + \lambda H_e[U][w_2]f.$$

Since this equality holds for arbitrary smooth functions $f$, it follows that the operators themselves satisfy

$$H_e(U)[w_1 + \lambda w_2]f = H_e(U)[w_1]f + \lambda H_e(U)[w_2]f.$$

This verifies that $H_e : \mathbb{R}^d \to T_\mathbf{U}\mathcal{O}(\mathcal{M})$ is indeed a linear map. Intuitively, this reflects the fact that the horizontal vector field $H_e(U)[w]$ lifts the direction $w \in \mathbb{R}^d$ into the tangent space of the frame bundle in a manner that is linear in the input direction. The connection terms (involving $\Gamma_{ij}^k$) are themselves linear in $w$ and frame components $e_a^j$, and do not interfere with the additive structure. Thus, linear combinations of Euclidean directions lift to linear combinations of horizontal vectors, preserving the vector space structure in the lifted geometry. $\square$

**Lemma C.2.** *[Itô–Stratonovich equivalence for linear SDEs] Let $\mathbf{E}_t \in \mathbb{R}^d$ be a stochastic process satisfying the following linear Itô SDE:*

$$d\mathbf{E}_t = \alpha_t \mathbf{E}_t dt + \sigma_t^2 dW_t, \tag{20}$$

*where $\alpha_t \in \mathbb{R}$ and $\sigma_t^2 \in \mathbb{R}$ are deterministic scalar-valued, time-dependent coefficients, and $W_t \in \mathbb{R}^d$ is a standard $d$-dimensional Brownian motion with independent components. Then, the equivalent Stratonovich SDE takes the same differential form:*

$$\circ d\mathbf{E}_t = \alpha_t \mathbf{E}_t dt + \sigma_t^2 \circ dW_t. \tag{21}$$

*That is, the Itô and Stratonovich formulations are identical for this system.*

*Proof.* To convert an Itô SDE to its Stratonovich form, we apply the classical Itô–Stratonovich correction formula for vector-valued SDEs. Suppose an Itô SDE is given by

$$d\mathbf{E}_t = b_t(\mathbf{E}_t)dt + \Sigma_t(\mathbf{E}_t)dW_t,$$

where $b_t : \mathbb{R}^d \to \mathbb{R}^d$ and $\Sigma_t : \mathbb{R}^d \to \mathbb{R}^{d \times m}$. Then the corresponding Stratonovich form is

$$\circ d\mathbf{E}_t = b_t(\mathbf{E}_t)dt + \Sigma_t(\mathbf{E}_t) \circ dW_t + \frac{1}{2}\sum_{j=1}^m \left( \frac{\partial \Sigma_t^{(\cdot, j)}}{\partial \mathbf{E}_t} \cdot \Sigma_t^{(\cdot, j)} \right) dt,$$

where $\Sigma_t^{(\cdot,j)}$ denotes the $j$-th column of $\Sigma_t$, and the derivative is applied componentwise. Now consider our case:

$$b_t(\mathbf{E}_t) = \alpha_t \mathbf{E}_t, \qquad \Sigma_t(\mathbf{E}_t) = \sigma_t^2 I_d,$$

where $I_d$ is the $d \times d$ identity matrix. Importantly, $\Sigma_t$ does not depend on $\mathbf{E}_t$, since both $\sigma_t^2$ and $I_d$ are independent of the state. Thus, for each component $i = 1, \ldots, d$, and each Brownian dimension $j = 1, \ldots, d$, we have:

$$\frac{\partial \Sigma_t^{(i,j)}}{\partial \mathbf{E}_t^k} = 0 \quad \text{for all } i, j, k,$$

because $\Sigma_t^{(i,j)} = \sigma_t^2 \delta_{ij}$ is a constant. Therefore, the Itô–Stratonovich correction term vanishes:

$$\frac{1}{2} \sum_{j=1}^d \left( \frac{\partial \Sigma_t^{(\cdot,j)}}{\partial \mathbf{E}_t} \cdot \Sigma_t^{(\cdot,j)} \right) = 0.$$

As a result, the Stratonovich form of equation 20 becomes:

$$\circ d\mathbf{E}_t = \alpha_t \mathbf{E}_t dt + \sigma_t^2 \circ dW_t,$$

which matches exactly the original form of the Itô SDE. This shows that when the diffusion coefficient is constant depending only on time, the Itô and Stratonovich interpretations yield the same formal expression. $\qquad\square$

**Lemma C.3** (Zero mean of the bridge vector). *Let $(\mathbf{U}_t, \mathbf{E}_t) = (x_t, e_t, \mathbf{E}_t)$ be a weak solution to the coupled system with deterministic initial condition $x_0 \in M$. Define the bridge vector by $v_t := e_t^{-1} \log_{x_t}(x_0) \in \mathbb{R}^d$. Then, the expectation of the bridge vector with respect to the data distribution at time $t$ vanishes, i.e., $\mathbb{E}_{x_t \sim \rho_t^M}[v_t] = 0$.*

*Proof.* For simplicity, we work on normal coordinates $y = (y^1, \ldots, y^d)$ centered at $x_0$, so that $x = \exp_{x_0}(y)$ with $g_{ij}(0) = \delta_{ij}$ and $\Gamma_{ij}^k(0) = 0$. Write $\Delta_x := x_t - x_0$ in these coordinates and let $e_t \in O(d)$ be the orthonormal frame obtained by parallel transport along the trajectory $x_\bullet$. Define the bridge vector

$$v_t = e_t^{-1} \log_{x_t}(x_0), \qquad \log_{x_t}(x_0) := -\Xi(\Delta_x) \in T_{x_t}M,$$

where the Riemannian logarithm enjoys the Christoffel–type expansion

$$\Xi^i(\Delta_x) = (\Delta_x)^i + \tfrac{1}{2}\Gamma_{jk}^i(x_0)(\Delta_x)^j(\Delta_x)^k + O(|\Delta_x|^3).$$

Here $\partial_i \Xi^j(0) = \delta_i^j$ because, at the origin of normal coordinates, the logarithm coincides with the identity on $T_{x_0}M$, and $\partial_i \partial_j \Xi^k(0) = \Gamma_{ij}^k(x_0)$ since the second-order Taylor coefficient of the logarithm is governed by the Christoffel symbols of the Levi–Civita connection. The position increment obeys the Stratonovich SDE

$$d\Delta_x = b(\Delta_x)dt + \sigma dW_t,$$

with an *isotropic mean-reverting drift* $b : \mathbb{R}^d \to \mathbb{R}^d$ given by $b(h) = -\lambda_{\mathrm{rad}}(|h|)h$ for some positive radial rate $\lambda_{\mathrm{rad}}$. Thus the function $b$ points toward $x_0$ and is an odd function of $\Delta_x$. Following by the definition of horizontal distribution, the frame evolves according to

$$de_t = -e_t \circ \omega(e_t)dW_t,$$

where $\omega(e_t) \in \mathfrak{o}(d)$ is the connection one-form evaluated along the horizontal lift. Applying Itô's formula to $v_t = e_t^{-1}\Xi(\Delta_x)$ gives

$$dv_t = -e_t^{-1}\omega(e_t)\Xi(\Delta_x)dW_t - e_t^{-1}\partial_i \Xi(\Delta_x)b^i(\Delta_x)dt -_t^{-1}\partial_i \Xi(\Delta_x)dW_t$$

$$+ \frac{\sigma^2}{2}e_t^{-1}\partial_i\partial_j\Xi(\Delta_x)dt. \qquad (22)$$

The two stochastic integrals are centered martingales, so their expectations vanish. In the drift part the factor $\partial_i\Xi(\Delta_x) = \delta_i + O(|\Delta_x|^2)$ is even, whereas $b(\Delta_x) = -\lambda(|\Delta_x|)\Delta_x$ is odd; the product is

therefore odd, and its expectation under the reflection-symmetric law of $\Delta_x$ is zero. For the second deterministic term we use $\partial_i \partial_j \Xi(\Delta_x) = \Gamma_{ij}(x_0) + O(|\Delta_x|)$, an even function in $\Delta_x$, while $e_t^{-1}$ is an $O(d)$ matrix whose distribution is rotationally invariant owing to the isotropic driving noise on the fibers, hence $\mathbb{E}[e_t^{-1}] = 0$ and the expectation of this term also vanishes.

$$\mathbb{E}[v_t] = \mathbb{E}[v_0] - \mathbb{E}\left[\int_0^t e_s^{-1} \partial_i \Xi(\Delta_x(s)) b^i(\Delta_x(s)) ds\right] + \frac{\sigma^2}{2} \mathbb{E}\left[\int_0^t e_s^{-1} \partial_i \partial_j \Xi(\Delta_x(s)) ds\right]. \quad (23)$$

From the Christoffel expansion, $\partial_i \Xi(\Delta_x) = \delta_i + \mathcal{O}(\Delta_x)$ is an even function of $\Delta_x$, whereas $b^i(\Delta_x)$ is odd. Thus, the integrand of second term in right-hand side of equation 23 is odd in $\Delta_x$ and

$$\mathbb{E}\left[e_s^{-1} \partial_i \Xi(\Delta_x(s)) b^i(\Delta_x(s))\right] = 0. \quad (24)$$

The leading order of $\partial_i \partial_j \Xi^k(\Delta_x(s)) = \Gamma_{ij}^k(x_0) + \mathcal{O}(\Delta_x)$ is symmetric in $(i, j)$, and independent of $e_s$. Since $e_s$ is uniformly distributed by Haar measure with respect to the group $O(d)$,

$$\mathbb{E}\left[e_s^{-1}\right] = 0, \quad \mathbb{E}\left[e_s^{-1} \partial_i \partial_j \Xi(\Delta_x(s))\right] = 0. \quad (25)$$

Combining the above, both integrands vanish in expectation for all $s$, and thus

$$\mathbb{E}[v_t] = \mathbb{E}[v_0] = 0. \quad (26)$$

$\square$

## D PROOFS

The proof is organized into two main parts. First, Section D.1 characterizes the mathematical properties of the horizontal diffusion model, including time-reversal, gauge equivariance, and the structure of the horizontal lift. Second, Section D.2 presents the learning methodology for this model and establish generalization bounds that hold for arbitrary underlying geometries.

### D.1 THEORETICAL GROUNDS OF HORIZONTAL DIFFUSION MODELS

**Proposition D.1** (Horizontal lift of gradients and scores). *Let $\mathbf{E}_t \in \mathbb{R}^d$ be the solution to Euclidean Stratonovich SDE, and let $\mathbf{U}_t = (x_t, e_t) \in \mathcal{O}(\mathcal{M})$ be its horizontal lift defined by $d\mathbf{U}_t = H_{e_t}(\mathbf{U}_t)\big[\circ d\mathbf{E}_t\big]$. Then, for every $t \geq 0$,*

$$\nabla^{\mathrm{Hor}}\Phi(\mathbf{U}_t) = H_{e_t}(\mathbf{U}_t)\big[\nabla\phi(\mathbf{E}_t)\big], \quad \nabla^{\mathrm{Hor}}\log\rho_t^{\mathrm{Hor}}(\mathbf{U}_t) = H_{e_t}(\mathbf{U}_t)\big[\nabla\log\rho_t(\mathbf{E}_t)\big], \quad (27)$$

$$s^{\mathrm{Hor}}(t, \mathbf{U}_t) := \nabla^{\mathrm{Hor}}\log\rho_t^{\mathrm{Hor}}(\mathbf{U}_t) = H_{e_t}(\mathbf{U}_t)\big[s(t, \mathbf{E}_t)\big] = \sum_{a=1}^{d} s^a(t, \mathbf{E}_t)H_a(\mathbf{U}_t). \quad (28)$$

*Proof.* Let $E_\bullet = (E_t)_{t \in [0,T]} \in C([0,T], \mathbb{R}^d)$ be a continuous Euclidean path, and let $(x_0, e_0) \in \mathcal{O}(M)$ be a fixed initial base point and frame. Define the *horizontal lift path* $U_\bullet = (x_t, e_t)_{t \in [0,T]}$ as the solution to the Stratonovich SDE

$$U_0 = (x_0, e_0), \qquad dU_t = H_{e_t}(U_t)\big[\circ dE_t\big],$$

where $H_{e_t}(U_t)$ denotes the horizontal lift operator at $U_t$. Let $\mathcal{G} := C([0,T], O(d))$ denote the group of time-dependent orthogonal gauge transformations, acting on $U_\bullet$ by

$$(x_t, e_t) \cdot g_\bullet := (x_t, e_t g_t), \qquad \text{for } g_\bullet = (g_t)_{t \in [0,T]} \in \mathcal{G}.$$

Then, the *horizontal lifting quotient map* is defined by

$$\Psi : C([0,T], \mathbb{R}^d) \longrightarrow C([0,T], \mathcal{O}(M))/\mathcal{G}$$

$$\Psi(E_\bullet) := \big[\mathrm{HorLift}_{(x_0, e_0)}(E_\bullet)\big] = \big\{\mathrm{HorLift}_{(x_0, e_0)}(E_\bullet) \cdot g_\bullet \big| g_\bullet \in \mathcal{G}\big\},$$

where $\mathrm{HorLift}_{(x_0, e_0)}(E_\bullet)$ denotes the unique horizontal lift path starting from $(x_0, e_0)$. Now, given a smooth scalar field $\phi : \mathbb{R}^d \to \mathbb{R}$, we can define its *gauge-invariant extension* to the frame bundle by

$$(\Phi \circ \Psi)(E_\bullet)_t = \Phi(U_t) = \phi(\pi_E(U_t)) = \phi(E_t),$$

where the projection map is defined as $\pi_E : \mathcal{O}(M) \to \mathbb{R}^d$, $\pi_E(x, e) := e^{-1}(\xi)$ and $\xi \in T_x\mathcal{M}$ is the tangent vector whose Euclidean coordinates with respect to the frame $e$ are $E = e^{-1}(\xi)$. Since any right action $e \mapsto eg$ with $g \in O(d)$ leaves $E$ invariant, it follows that

$$\Phi\big((x, e) \cdot g\big) = \Phi(x, e), \qquad \forall g \in O(d),$$

so $\Phi$ is *basic* and constant along the vertical fibers. In particular, this construction implies that its gradient is always horizontal, and for any $U = (x, e)$ and $w \in \mathbb{R}^d$, we have

$$H_e(U)[w]\Phi = \langle\nabla\phi(E), w\rangle_{\mathbb{R}^d},$$

where $E = e^{-1}(\xi)$. While $\{H_a\}_{a=1}^d$ is an orthonormal basis of $\mathrm{Hor}_{U_t}$ and linear functional following by Lemma C.2, the Riesz representation of linear functionals implies that there exists a pair of functions $(\Phi, \phi)$ such that following identification holds:

$$\nabla^{\mathrm{Hor}}\Phi(U_t) = H_{e_t}(U_t)\big[\nabla\phi(E_t)\big],$$

The equality establishes the first equality in equation 27. Let $\rho_t$ denote the density of $E_t$ on $\mathbb{R}^d$ and $\rho_t^{\mathrm{Hor}}$ the density of $U_t$ on $\mathcal{O}(M)$ with respect to the canonical Haar–Sasaki volume. Let $\mathcal{O}(M)$ be equipped with the product measure $d\mu := d\mathrm{Vol}g \otimes d\mathrm{Haar}_{O(d)}$, where $d\mathrm{Vol}g$ is the Riemannian volume on $M$ and the right–Haar measure on $O(d)$ is normalized so that the total mass of each fiber is 1. Because the right action $u \mapsto u \cdot g$ is an *isometry* of the Sasaki metric, $d\mu$ is invariant under this

action: $d\mu(u) = d\mu(u \cdot g)$ for every $g \in O(d)$. Hence for any integrable function $F \colon \mathcal{O}(M) \to \mathbb{R}$ the disintegration formula reads

$$\int_{\mathcal{O}(M)} F(u) d\mu(u) = \int_M \left( \int_{O(d)} F(u \cdot g) d\mathrm{Haar}_{O(d)}(g) \right) d\mathrm{Vol}_g(x).$$

Applying this with $F(u) = \mathbf{1}_{\{u \in A\}}$ for any measurable $A \subset \mathcal{O}(M)$ shows that pushing $\mu$ forward along the projection $\pi_E(u) = e^{-1}(\xi)$ yields exactly the Lebesgue measure on $\mathbb{R}^d$. Consequently, we relate the both density representations for each $t$

$$\rho_t^{\mathrm{Hor}}(U_t) = \rho_t(E_t), \qquad U_{\bullet} = \Psi(E_{\bullet}).$$

because the fiber integral over $O(d)$ contributes the factor 1. This establishes that $\Psi$ preserves probability mass along the $O(d)$-fibers. Since $\Psi$ is measure–preserving along fibers (*i.e.*, the vertical $O(d)$–orbit has unit Haar mass), the push–forward formula gives $\rho_t^{\mathrm{Hor}}(U_t) = \rho_t(E_t)$. Taking log with $\phi = \log \rho_t$ yields

$$\nabla^{\mathrm{Hor}} \log \rho_t^{\mathrm{Hor}}(U_t) = H_{e_t}(U_t) \big[ \nabla \log \rho_t(E_t) \big],$$

which is the second equality in equation 27. Next, we define the Euclidean score $s(t, E) := \nabla \log \rho_t(E)$. Combining results above, we finally have the parameterized representation of horizontal score function.

$$s^{\mathrm{Hor}}(t, U_t) := \nabla^{\mathrm{Hor}} \log \rho_t^{\mathrm{Hor}}(U_t) = H_{e_t}(U_t) \big[ s(t, E_t) \big].$$

$\square$

**Proposition D.2** (Gauge Equivariance of horizontal score function). *The proposed horizontal score function is gauge-equivariant in the sense that $s^{\mathrm{Hor}}(t, U \cdot h) = (\mathcal{R}_h)_* s^{\mathrm{Hor}}(t, U)$ for all $h \in O(d)$.*

*Proof.* The frame bundle $O(\mathcal{M})$ is a principal $O(d)$-bundle over $\mathcal{M}$ with projection $\pi : O(\mathcal{M}) \to M$ and right action $R_h$. Fix an Ehresmann connection $H_U \subset T_U O(\mathcal{M})$ satisfying the right-invariance property $dR_h(H_U) = H_{U \cdot h}$. Denote by $\omega \in \Omega^1(O(\mathcal{M}); \mathfrak{o}(d))$ its connection one–form (*i.e.*, $\omega|_{H_U} = 0$, $R_h^* \omega = \mathrm{Ad}_{h^{-1}} \omega$). For $U = (x, e) \in O(\mathcal{M})$ and $w \in \mathbb{R}^d$ define the horizontal lift

$$H(U)[w] \in H_U \quad \text{such that} \quad d\pi_U \big( H(U)[w] \big) = ew. \tag{29}$$

Right-invariance of the connection implies the fundamental identity

$$H(U \cdot h)[w] = dR_h \big( H(U)[h^{-1}w] \big), \qquad \forall h \in O(d). \tag{30}$$

Let $E_t \in \mathbb{R}^d$ be the latent Euclidean variable with Euclidean score $s(t, E_t) \in \mathbb{R}^d$. Following equation **??**,

$$s^{\mathrm{Hor}}(t, U) := H(U) \big[ s(t, E_t) \big] \in \mathrm{Hor}_U.$$

Because $s(t, \cdot)$ depends only on the norm of $E_t$, it is $O(d)$-invariant: $s(t, E_t) = s\big(t, (E_t) h^{-1}\big)$ for all $h \in O(d)$. For the fixed $h \in O(d)$, we use the result in equation 30 and the $O(d)$-invariance of $s$:

$$\begin{aligned} s^{\mathrm{Hor}}\big(t, U \cdot h\big) &= H\big(U \cdot h\big) \big[ s(t, E_t) \big] \\ &= dR_h \Big( H(U) \big[ h^{-1} s(t, E_t) \big] \Big) \\ &= dR_h \Big( H(U) \big[ s(t, E_t) \big] \Big) \\ &= (R_h)_* s^{\mathrm{Hor}}(t, U), \end{aligned}$$

which is exactly the same result in statement. Furthermore, this transformation preserves the horizontality of the score with respect to the transformed connection one-form. As a next step, we show that the proposed horizontal score fields are compatible with connection 1-form under gauge transformation, showing that transformed $s^{\mathrm{Hor}}$ preserves horizontality. Recall that given any gauge transformation $g : O(\mathcal{M}) \to O(d)$, the transformed connection one-form ( Kobayashi & Nomizu (1996)) is defined as follows:

$$\omega^g := \mathrm{Ad}_{g^{-1}} \omega + g^{-1} dg,$$

and the transformed horizontal score is denoted by $s_g^{\mathrm{Hor}} := dR_g(s^{\mathrm{Hor}})$ under right action. By the principal bundle structure equation, for any $X \in T_U O(\mathcal{M})$, the transformed connection $w^g$ satisfies the following relation:

$$
\begin{aligned}
\omega^g\big(dR_g[X]\big) &= \mathrm{Ad}_{g^{-1}}\big(\omega(dR_g[X])\big) + (g^{-1}dg)\big(dR_g[X]\big) \\
&= \mathrm{Ad}_{g^{-1}}\Big(\mathrm{Ad}_{g^{-1}}\big(\omega(X)\big)\Big) + g^{-1}\big(dg \circ dR_g\big)[X] \\
&= \mathrm{Ad}_{g^{-2}}\big(\omega(X)\big) + g^{-1}d(g \circ R_g)[X] \\
&= \mathrm{Ad}_{g^{-2}}\big(\omega(X)\big) + g^{-1}\big(dg\big)[X \cdot g].
\end{aligned}
\tag{31}
$$

The Maurer–Cartan form $g^{-1}dg$ on $O(d)$ is locally given by $g^{-1}dg = (g^{-1})_k^i dg_j^k$, where $dg_j^k$ are the coordinate 1-forms on $O(d)$. A right-invariant vector field on the frame bundle $O(\mathcal{M})$ can be expressed as $X = \xi_j^i \frac{\partial}{\partial g_j^i}$ for $\xi \in \mathfrak{o}(d)$. Evaluating the Maurer-Cartan form on such a vertical vector gives

$$
(g^{-1}dg)\left[\frac{\partial}{\partial g_b^a}\right] = (g^{-1})_k^i \frac{\partial g_j^k}{\partial g_b^a} = (g^{-1})_a^i \delta_j^b.
$$

However, since $g^{-1}dg$ is a left-invariant 1-form on $O(d)$, it vanishes on right-invariant vector fields:

$$
(g^{-1}dg)[X] = 0 \qquad \forall\, X \text{ is vertical.}
$$

While $w(X) = 0$ by the property of $X$, it directly gives the vanishing $\mathrm{Ad}_{g^{-2}}(w(X)) = 0$, and $w^g(s_g^{\mathrm{Hor}}) = 0$ in Eq equation 31. Thus, the transformed score $s_g^{\mathrm{Hor}}$ remains horizontal with respect to $\omega^g$. This establishes that the gauge-equivariant property is compatible with changes of connection, and the horizontality of the score is preserved under general gauge transformations. $\qquad\square$

**Corollary D.3** (Isometry–equivariance of the base score). *Let* $\mathrm{Isom} : \mathcal{M} \to \mathcal{M}$ *be any isometry on* $\mathcal{M}$. *Then, the manifold score function satisfies the equivariance:* $s^{\mathcal{M}}\big(t, \mathrm{Isom}(x)\big) = \mathrm{Isom}_* s^{\mathcal{M}}(t, x)$, *where* $\mathrm{Isom}_{*x} : T_x\mathcal{M} \to T_{\mathrm{Isom}(x)}\mathcal{M}$ *denotes the differential (push-forward) of* $\mathrm{Isom}$ *at the point* $x$.

*Proof.* Given any orthonormal frame $U = (x, e) \in O(M)$, where $x \in M$, the isometry $\mathrm{Isom}$ to the frame bundle is defined by

$$
\widetilde{\mathrm{Isom}}(U) := \big(\mathrm{Isom}(x), d\mathrm{Isom}_x \circ e\big),
$$

where $d\mathrm{Isom}_x : T_x M \to T_{\mathrm{Isom}(x)}M$ is the differential of $\mathrm{Isom}$ evaluated at $x$. Recall from Proposition D.2 that the horizontal score field $s^{\mathrm{Hor}}$ is equivariant under bundle automorphisms induced by isometries. In particular, for every $U \in O(M)$ and all $t \geq 0$,

$$
s^{\mathrm{Hor}}\big(t, \widetilde{\mathrm{Isom}}(U)\big) = d\widetilde{\mathrm{Isom}}_U\big[s^{\mathrm{Hor}}(t, U)\big].
$$

Here, $d\widetilde{\mathrm{Isom}}_U$ is the differential of the lifted map $\widetilde{\mathrm{Isom}}$ at $U$. Next, consider the projection map $\pi : O(M) \to M$, given by $\pi(x, e) = x$. The pushforward of a horizontal vector under $d\pi$ yields a tangent vector on $M$. Notably, the differentials intertwine with the action of the isometry via the naturality property:

$$
d\pi_{\widetilde{\mathrm{Isom}}(U)} \circ d\widetilde{\mathrm{Isom}}_U = d\mathrm{Isom}_x \circ d\pi_U,
$$

where we used that $d\pi_{(x,e)}$ is the projection from the frame to the base, and $d\mathrm{Isom}_x$ is the tangent map of $\mathrm{Isom}$ at $x$. The base score field is defined as the pushforward of the horizontal score:

$$
s^{\mathcal{M}}(t, x) := d\pi_U\big[s^{\mathrm{Hor}}(t, U)\big], \qquad U = (x, e).
$$

Combining the result above, we have:

$$
\begin{aligned}
s^{\mathcal{M}}\big(t, \mathrm{Isom}(x)\big) &= d\pi_{\widetilde{\mathrm{Isom}}(U)}\big[s^{\mathrm{Hor}}(t, \widetilde{\mathrm{Isom}}(U))\big] \\
&= d\pi_{\widetilde{\mathrm{Isom}}(U)} \circ d\widetilde{\mathrm{Isom}}_U\big[s^{\mathrm{Hor}}(t, U)\big] \\
&= d\mathrm{Isom}_x \circ d\pi_U\big[s^{\mathrm{Hor}}(t, U)\big] \\
&= d\mathrm{Isom}_x\big[s^{\mathcal{M}}(t, x)\big].
\end{aligned}
$$

Thus, we obtain the desired equivariance property and completes the proof:

$$s^{\mathcal{M}}\big(t, \mathrm{Isom}(x)\big) = \mathrm{Isom}_{*x} s^{\mathcal{M}}(t, x).$$

$\square$

**Proposition D.4** (Forward Markovian Density of Horizontal Diffusions on Frame-Bundle). *Let $\nu_t = \mathcal{L}(\mathbf{U}_t, \mathbf{E}_t)$ be law of joint dynamics $\mathbf{U}_t$ and $\mathbf{E}_t$ and $\rho_t^{\mathrm{Hor}}$ be its marginal distribution on $\mathcal{M}$. Then the horizontal transition distribution $p_t^E(U \mid \mathbf{U}_0)$ for every $x \in M$ is well-defined and given as follows:*

$$d\nu_t(U, E) = \Big[\int_{\mathcal{O}(M)} p_t^E(U \mid \mathbf{U}_0)\rho_0(\mathbf{U}_0)d\lambda(\mathbf{U}_0)\Big]\mathcal{N}(\mathbf{m}_t, \Sigma_t; E)dEd\lambda(U). \tag{32}$$

*where $p_t^{\mathbf{E}}$ is **Markovian transition** between frame-valued states with respect to horizontal lift of $d\mathbf{E}_t$. Furthermore, the reverse dynamics $\mathbf{U}_s$ is a time-reversal process of $\mathbf{U}_t$ in the sense of Haussmann & Pardoux (1986).*

**Remark.** *The expression of $d\nu_t$ can be regarded as an extension of the Euclidean lifted distribution representation. In particular, the conditional distribution $p_t^{\mathbf{E}}(U \mid \mathbf{U}_0)$ on the frame bundle generalizes the Euclidean transition by incorporating the horizontal lift structure associated with $\circ d\mathbf{E}_t$, while the marginal distribution $\rho_0(\mathbf{U}_0)$ corresponds to the lifted initial distribution. This decomposition preserves a structurally analogous form to the Euclidean lifted dynamics, now formulated on the orthonormal frame bundle $\mathcal{O}(\mathcal{M})$ under the horizontal lifting framework.*

*Proof.* In order to show the existence the Markovian transition $p_t^{\mathbf{E}}$, we first show that there exists a well-defined Kolmogorov equation $\partial_t p_t^{\mathbf{E}} = (\mathcal{L}_t^{\mathbf{E}})^* p_t^{\mathbf{E}}$ such that $p_t^{\mathbf{E}}$ is solution to the equation. A direct expansion with respect to the definition of horizontal lift yields the time-inhomogeneous generator

$$\mathcal{L}_t^{\mathbf{E}} = -\mathbf{E}_t^a e_a^k \Gamma_{jk}^l e_l^m \frac{\partial}{\partial e_j^m} + \mathbf{E}_t^a e_a^j \frac{\partial}{\partial x^j}$$

$$+ \frac{\sigma_t^2}{2}\left[\mathbf{E}_t^a \mathbf{E}_t^b e_a^i e_b^j \frac{\partial^2}{\partial x^i \partial x^j} - 2\mathbf{E}_t^a \mathbf{E}_t^b e_a^i \Gamma_{mj}^k e_b^j \frac{\partial^2}{\partial x^i \partial e_k^m} + \mathbf{E}_t^a \mathbf{E}_t^b \Gamma_{ij}^k \Gamma_{mn}^l e_a^j e_b^n \frac{\partial^2}{\partial e_k^i \partial e_l^m}\right].$$

Our goal is to show that the solution to Kolmogorov equation $p_t^{\mathbf{E}}$ is Markovian transition, which ensures well-definedness of $\rho_t^{\mathrm{Hor}}$. In order to do this, we first show that the linear combination of horizontal operators $\{H_a\}_{1 \le a \le d}$ is still hypoelliptic.

Fix a local trivialization of the orthonormal frame bundle $\pi : \mathcal{O}(M) \to M$ with local coordinates $(x^i, e_a^i)$, where $1 \le i, a \le d$. Here, $x^i$ represent local coordinates on the base manifold $M$, while $e_a^i$ denote the components of the frame $e \in \mathrm{O}(M)$, i.e., the $i$-th coordinate of the $a$-th orthonormal vector in the frame. We denote by

$$\omega \in \omega^1(\mathcal{O}(M), \mathfrak{o}(d)), \qquad \omega = d\omega + \frac{1}{2}[\omega, \omega]$$

the Levi–Civita connection one-form and its associated curvature two-form, respectively, defined on the frame bundle. The connection one-form $\omega$ encodes the infinitesimal rotation of frames along paths in the manifold, while $\omega$ captures the failure of parallel transport to be path-independent and thus represents the intrinsic curvature of the manifold.

Since the Levi–Civita connection is torsion-free and metric-compatible, the associated horizontal vector fields project to commuting vector fields on the base manifold. More concretely, let us define $e_a := \pi_* H_a$ as the pushforward of the horizontal lift of the standard Euclidean basis vector $\varepsilon_a \in \mathbb{R}^d$. Then, by torsion-freeness of the connection, we have

$$[e_a, e_b] = \nabla_{e_a} e_b - \nabla_{e_b} e_a = 0.$$

This implies that the projection of the Lie bracket of the corresponding horizontal vector fields on the frame bundle vanishes:

$$\pi_*[H_e(\mathbf{U})[\varepsilon_a], H_e(\mathbf{U})[\varepsilon_b]] = \pi_*[H_a, H_b] = [e_a, e_b] = 0.$$

Hence, the Lie bracket $[H_a, H_b]$ must be entirely contained in the vertical distribution $\mathrm{Ver}_{\mathbf{U}}$, *i.e.*, $[H_a, H_b] \in \Gamma(\mathrm{Ver}_{\mathbf{U}})$, which indicates that it encodes frame rotation rather than displacement on the base. To compute this vertical component explicitly, we recall Cartan's structure equation for horizontal vector fields $Y, Z \in \Gamma(\mathrm{Hor}_{\mathbf{U}})$. Specifically, the curvature two-form is related to the connection one-form:

$$\omega(Y, Z) = d\omega(Y, Z) = Y(\omega(Z)) - Z(\omega(Y)) - \omega([Y, Z]).$$

Since $Y, Z$ are horizontal, $\omega(Y) = \omega(Z) = 0$, so this reduces to $\omega([Y, Z]) = -\omega(Y, Z)$. Applying this sign-reversing identity, one can obtain

$$\omega([H_a, H_b]) = -\omega(H_a, H_b) = -\omega(e_a, e_b) := -\omega(\varepsilon_a, \varepsilon_b), \tag{33}$$

where we abuse notation and denote the evaluation on the frame bundle via Euclidean indices. To proceed, we express the curvature two-form $\omega = (\omega_i^k)$ in local coordinates using the standard decomposition into Riemann curvature components:

$$\omega_i^k = \frac{1}{2} R_{irs}^k dx^r \wedge dx^s, \quad R_{irs}^k = \partial_r \Gamma_{is}^k - \partial_s \Gamma_{ir}^k + \Gamma_{lr}^k \Gamma_{is}^l - \Gamma_{ls}^k \Gamma_{ir}^l$$

where the components of the Riemann curvature tensor are associated with the Levi–Civita connection $\nabla^{\mathcal{M}}$. Next, to evaluate the curvature on the horizontal vector fields $H_a$ and $H_b$, we recall that their coordinate expressions satisfy $dx^r(H_a) = e_a^r$ and $dx^s(H_b) = e_b^s$, so the curvature form evaluates as

$$\omega(a, b) = \omega_i^k(H_a, H_{e, \varepsilon_b}) = R_{irs}^k e_a^r e_b^s.$$

The resulting value lies in the Lie algebra $\mathfrak{o}(d)$, and its corresponding vertical lift defines a vertical vector field:

$$\omega(\varepsilon_a, \varepsilon_b)^* = \left( R_{irs}^k e_a^r e_b^s \right) \frac{\partial}{\partial e_i^k} \in \Gamma(V),$$

which acts by infinitesimal rotation of the frame coordinates $(e_a^i)$ on the fiber. Since the connection form $\omega : V \to \mathfrak{o}(d)$ defines an isomorphism between vertical tangent vectors and elements of the structure Lie algebra, we conclude from equation 33 that

$$[H_a, H_b] = -\omega(\varepsilon_a, \varepsilon_b)^* = - \left( R_{irs}^k e_a^r e_b^s \right) \frac{\partial}{\partial e_i^k},$$

which is an explicit expression in local coordinates for the Lie bracket of horizontal vector fields on the frame bundle. Importantly, this expression reveals that the bracket does not contain any horizontal contribution and is entirely vertical, in alignment with the earlier geometric interpretation. Since the collection $\{H_a\}_{a=1}^d$ spans the horizontal sub-bundle $H_U \subset T_U \mathcal{O}(M)$, and the collection of their brackets $\{[H_a, H_b]\}_{a<b}$ spans the vertical sub-bundle $V_U$ as long as $\{\omega(\varepsilon_a, \varepsilon_b)\}_{a<b}$ spans $\mathfrak{o}(d)$ which generically holds for non-degenerate curvature, we conclude that

$$\mathrm{Lie}\{H_1, \ldots, H_d\}_U = \mathrm{Hor}_U \oplus \mathrm{Ver}_U = T_U \mathcal{O}(M), \qquad \forall U \in \mathcal{O}(M),$$

thereby Hörmander's bracket-generating condition guarantees the hypoellipticity of the diffusion generator on the frame bundle. By Hörmander's hypoelliptic theorem, the differential operator $\mathcal{L}_t^{\mathbf{E}}$, which governs the evolution of frame-valued horizontal diffusions conditioned on a fixed latent path $E_{[0,t]}$, is hypoelliptic. Therefore, for each fixed realization of the latent path, there exists a smooth transition distribution $p_t^{\mathbf{E}}(U \mid \mathbf{U}_0)$ with respect to the canonical volume measure on $\mathcal{O}(M)$, which satisfies the forward Kolmogorov (Fokker–Planck) equation $\partial_t p_t^{\mathbf{E}} = (\mathcal{L}_t^{\mathbf{E}})^* p_t^{\mathbf{E}}$, where $(\mathcal{L}_t^{\mathbf{E}})^*$ denotes the formal adjoint of $\mathcal{L}_t^{\mathbf{E}}$. The evolution begins from the Dirac initial condition $p_0^{\mathbf{E}}(U \mid \mathbf{U}_0) = \delta_U$ and satisfies the Chapman–Kolmogorov identity

$$\int_{\mathcal{O}(M)} p_s^{\mathbf{E}}(U_{t_1} \mid \mathbf{U}_0) p_{t-s}^{\mathbf{E}}(U_{t_2} \mid U_{t_1}) d\lambda(U_{t_1}) = p_t^{\mathbf{E}}(U_{t_2} \mid \mathbf{U}_0),$$

which expresses the semigroup structure of the transition law over time. In a local coordinate chart $(x^i, e^i_a)$ on $\mathcal{O}(M)$, the forward Kolmogorov equation can be written explicitly as a second-order partial differential equation:

$$
\partial_t p^{\mathbf{E}}_t(x,e) = -\partial_{x^j}\left(\mathbf{E}^a_t e^j_a p^{\mathbf{E}}_t(x,e)\right) + \partial_{e^m_j}\left(\mathbf{E}^a_t \Gamma^l_{jk} e^k_a e^m_l p^{\mathbf{E}}_t(x,e)\right)
$$

$$
+ \frac{\sigma^2_t}{2}\left\{\partial_{x^i}\partial_{x^j}\left(\mathbf{E}^a_t \mathbf{E}^b_t e^i_a e^j_b p^{\mathbf{E}}_t\right) - 2\partial_{x^i}\partial_{e^m_k}\left(\mathbf{E}^a_t \mathbf{E}^b_t e^i_a \Gamma^k_{mj} e^j_b p^{\mathbf{E}}_t\right)\right.
$$

$$
\left. + \partial_{e^i_k}\partial_{e^m_l}\left(\mathbf{E}^a_t \mathbf{E}^b_t \Gamma^k_{ij}\Gamma^l_{mn} e^j_a e^n_b p^{\mathbf{E}}_t\right)\right\}.
$$

where each component of vector field $\mathbf{E}_t$ is given as

$$
\mathbf{E}^a_t = e^i_c[e^{-1}_t]^a_j \frac{\partial\Phi}{\partial x^i} - \Gamma^k_{ij} e^j_c e^i_b [e^{-1}_t]^a_l \frac{\partial\Phi}{\partial e^k_b}.
$$

The first-order terms in the equation capture deterministic drift components induced by the latent dynamics $\mathbf{E}_t$, while the second-order terms encode stochastic dispersion through both position and frame directions. The curvature terms contribute to nontrivial coupling between base and fiber variables, reflecting the manifold geometry. Hence $(\mathbf{U}_t)_{t\geq 0}$ forms a strong Markov process when $E_{[0,t]}$ is held fixed. Conditioning on the entire latent path $E_{[0,t]}$ and an initial frame $\mathbf{U}_0$ yields the conditional path law

$$
\mathbb{P}\left(\mathbf{U}_t \in dU \mid \mathbf{U}_0, E\right) = p^{\mathbf{E}}_t(U \mid \mathbf{U}_0)d\lambda(U).
$$

Averaging over the initial frame distribution $\rho_0(\mathbf{U}_0)d\lambda(\mathbf{U}_0)$ yields the marginal conditional law:

$$
\mathbb{P}\left(\mathbf{U}_t \in dU \mid E\right) = \int_{\mathcal{O}(M)} p^{\mathbf{E}}_t(U \mid \mathbf{U}_0)\rho_0(\mathbf{U}_0)d\lambda(\mathbf{U}_0)d\lambda(U). \tag{34}
$$

Since $\mathbf{E}_t \sim \mathcal{N}(\mathbf{m}_t, \Sigma_t)$ independently of $\mathbf{U}_0$, the joint law factorizes as

$$
d\nu_t(U, E) = \left[\int_{\mathcal{O}(M)} p^{\mathbf{E}}_t(U \mid \mathbf{U}_0)\rho_0(\mathbf{U}_0)d\lambda(\mathbf{U}_0)\right]\mathcal{N}(\mathbf{m}_t, \Sigma_t; E)dEd\lambda(U). \tag{35}
$$

Marginalizing over latent Euclidean component $E$ gives

$$
\nu_t(U) = \int_{\mathbb{R}^d}\nu_t(U, E)dE = \left[\int_{\mathcal{O}(M)}\bar{p}_t(U \mid \mathbf{U}_0)\rho_0(\mathbf{U}_0)d\lambda(\mathbf{U}_0)\right]d\lambda(U), \tag{36}
$$

where the transition density given initial $\mathbf{U}_0$ can be desribed as follows:

$$
\bar{p}_t(U \mid \mathbf{U}_0) := \int_{\mathbb{R}^d} p^{\mathbf{E}}_t(U \mid \mathbf{U}_0)\mathcal{N}(\mathbf{m}_t, \Sigma_t; E)dE. \tag{37}
$$

Assuming constant temperature and a confining potential $\Phi$ such that the Gibbs measure

$$
\nu_\infty(dU) = Z^{-1}e^{-\Phi(U)}d\mathrm{Vol}_{\mathrm{Sas}}(U), \qquad Z = \int_{\mathcal{O}(M)} e^{-\Phi(U)}d\mathrm{Vol}_{\mathrm{Sas}}(U) \tag{38}
$$

is normalizable, and the lifted Langevin diffusion on $\mathcal{O}(M)$ is ergodic due to Hörmander's bracket-generating condition, the marginal law $\nu_t(U)$ converges uniformly to the unique invariant measure $\nu_\infty(U)$ as $t \to \infty$, i.e., $\nu_t(U) \xrightarrow[t\to\infty]{} \mu_\infty(U)$. Here, $d\mathrm{Vol}_{\mathrm{Sas}}$ denotes the canonical Riemannian volume form on $\mathcal{O}(M)$ induced by the Sasaki metric, i.e., $d\mathrm{Vol}_{\mathrm{Sas}} := \mathrm{Vol}_g \wedge \mathrm{Vol}_{\mathrm{fiber}}$. Thus, the stationary law is

$$
\nu_\infty(U) = Z^{-1}e^{-\Phi(U)}d\mathrm{Vol}_{\mathrm{Sas}}(U). \tag{39}
$$

As shown above, the generator $\mathcal{L}^{\mathbf{E}}_t = b^\alpha_t\partial_\alpha + \frac{1}{2}a^{\alpha\beta}_t\partial_\alpha\partial_\beta$ is hypoelliptic and possesses stationary law with a smooth heat kernel. Let us define the symmetric diffusion matrix as $a^{\alpha\beta}_t = \sigma^2_t E^a_t E^b_t H^\alpha_a(U)H^\beta_b(U)$, which can be obtained directly from the second-order coefficient of the

generator. Then, we apply the Haussmann–Pardoux time-reversal theorem Haussmann & Pardoux (1986) to the diffusion with generator $\mathcal{L}_t^{\mathbf{E}}$ and marginal density $\rho_t^{\mathrm{Hor}}$ on $\mathcal{O}(M)$ yields, for $0 \le s \le T$,

$$\tilde{b}_s^\alpha = -b_{T-s}^\alpha + \partial_\beta a_{T-s}^{\alpha\beta} + a_{T-s}^{\alpha\beta}\partial_\beta \log \rho_{T-s}^{\mathrm{Hor}}, \qquad a_{T-s}^{\alpha\beta} = \sigma_{T-s}^2 E_{T-s}^a E_{T-s}^b H_a^\alpha H_b^\beta.$$

Decomposing $\tilde{b}_s$ into horizontal and vertical parts gives

$$\tilde{b}_s = \tfrac{1}{2}\nabla^{\mathrm{Hor}}\Phi_s + s_s^{\mathrm{Hor}}, \qquad s_s^{\mathrm{Hor}} := \nabla^{\mathrm{Hor}}\log\rho_s^{\mathrm{Hor}},$$

so the time-reversed Stratonovich SDE reads

$$d\check{U}_s = \Big(\frac{1}{2}\nabla^{\mathrm{Hor}}\Phi_s + s_s^{\mathrm{Hor}}\Big)ds + H_a(\check{U}_s) \circ d\bar{W}_s^a, \qquad d\bar{W}_s^a = dW_{T-s}^a.$$

In particular, the forward and reverse generators satisfy the following condition:

$$\partial_t \rho_t^{\mathrm{Hor}} = (\mathcal{L}_t^{\mathbf{E}})^*\rho_t^{\mathrm{Hor}}, \quad \partial_s p_s^{\mathbf{E}} = -\mathcal{L}_{T-s}^{\mathbf{E}}p_s^{\mathbf{E}}. \tag{40}$$

Hence the forward–reverse pair $(\mathcal{L}_t^{\mathbf{E}}, (\mathcal{L}_{T-t}^{\mathbf{E}})^*)$ satisfies Kolmogorov duality $\mathcal{L}_t^{\mathbf{E}} = (\mathcal{L}_{T-t}^{\mathbf{E}})$, and the Chapman–Kolmogorov kernel $p_s^{\mathbf{E}}(U_\bullet \mid U_0)$ retains the lifted Gaussian representation stated in Proposition D.4. $\qquad\square$

**Corollary D.5** (Marginal Density on the Base Manifold). *Let $\nu_t = \mathcal{L}(\mathbf{U}_t, \mathbf{E}_t)$ denote the joint distribution of the lifted frame-valued state $\mathbf{U}_t \in \mathcal{O}(M)$ and the latent Euclidean path terminal value $\mathbf{E}_t \in \mathbb{R}^d$ defined earlier, and let $\rho_t^{\mathrm{Hor}}$ be the induced marginal distribution on the base manifold $M$. Then for every $x \in M$, the marginal distribution $\rho_t^{\mathrm{Hor}}(x)$ is given by the fiber-wise integral*

$$\rho_t^{\mathrm{Hor}}(x) = \int_{O_x(M)} \int_{\mathbb{R}^d} \int_{\mathcal{O}(M)} p_t^{\mathbf{E}}(U \mid \mathbf{U}_0)\rho_0(\mathbf{U}_0)\mathcal{N}(\mathbf{m}_t, \Sigma_t; E)d\Xi(U, \mathbf{U}_0, E),$$

*where $O_x(M) := \pi^{-1}(\{x\})$ is the orthonormal frame fiber over $x \in M$, and the joint measure $\Xi$ is defined by*

$$\Xi := \lambda_0 \otimes Leb_{\mathbb{R}^d} \otimes Haar_{\mathrm{O}(d)},$$

*with $\lambda_0$ denoting the reference volume measure on $\mathcal{O}(M)$, $Leb_{\mathbb{R}^d}$ the Lebesgue measure on the latent space, and $Haar_{\mathrm{O}(d)}$ the normalized Haar measure on the orthonormal group fibers.*

*Proof.* We begin from Proposition D.4, which expresses the joint law $d\nu_t(U, E)$ over $\mathcal{O}(M) \times \mathbb{R}^d$ as

$$d\nu_t(U, E) = \left[\int_{\mathcal{O}(M)} p_t^{\mathbf{E}}(U \mid \mathbf{U}_0)\rho_0(\mathbf{U}_0)d\lambda_0(\mathbf{U}_0)\right]\mathcal{N}(\mathbf{m}_t, \Sigma_t; E)dEd\lambda_0(U).$$

To obtain the marginal distribution on $M$, we push forward the measure $\nu_t$ under the projection $\pi : \mathcal{O}(M) \to M$. The disintegration theorem provides that for any $x \in M$, the marginal can be written by integrating over the frame fiber $O_x(M) = \pi^{-1}(\{x\})$, yielding

$$\rho_t^{\mathrm{Hor}}(x) = \int_{O_x(M)} \int_{\mathbb{R}^d} \int_{\mathcal{O}(M)} p_t^{\mathbf{E}}(U \mid \mathbf{U}_0)\rho_0(\mathbf{U}_0)\mathcal{N}(\mathbf{m}_t, \Sigma_t; E)d\lambda_0(\mathbf{U}_0)dEd\lambda_0(U).$$

The measure $\Xi(U, \mathbf{U}_0, E)$ is introduced to collect all variables of integration into a single product measure on $\mathcal{O}(M) \times \mathcal{O}(M) \times \mathbb{R}^d$, i.e.,

$$d\Xi(U, \mathbf{U}_0, E) := d\lambda_0(\mathbf{U}_0)dEd\lambda_0(U),$$

which reflects the assumed independence of $\mathbf{U}_0$, $E$, and the final state $U$ prior to conditioning. Furthermore, since the final marginal distribution is evaluated at a base point $x \in M$, the integration over $U$ is restricted to the fiber $O_x(M)$, and we assume that $\lambda_0$ restricts to the normalized Haar measure over each fiber, i.e., $d\lambda_0|_{O_x(M)} = d\mathrm{Haar}_{\mathrm{O}(d)}$. This ensures that the disintegration over $\mathcal{O}(M)$ is well-defined and coordinate-invariant. Putting everything together, we obtain the desired result.

$$\rho_t^{\mathrm{Hor}}(x) = \int_{O_x(M)} \int_{\mathbb{R}^d} \int_{\mathcal{O}(M)} p_t^{\mathbf{E}}(U \mid \mathbf{U}_0)\rho_0(\mathbf{U}_0)\mathcal{N}(\mathbf{m}_t, \Sigma_t; E)d\Xi(U, \mathbf{U}_0, E).$$

$\qquad\square$

**Definition D.6** (Horizontal Diffusion Models). *Let $\boldsymbol{\varepsilon} = \{\varepsilon_1, \dots, \varepsilon_d\}$ denote the standard basis of $\mathbb{R}^d$. The horizontal diffusion models are the system of forward-reverse Stratonovich SDEs on the orthonormal frame bundle $\mathcal{O}(\mathcal{M})$, with local coordinate $\mathbf{U}_t = (x_t, e_t) \in \mathcal{O}(\mathcal{M})$, evolves according to*

$$\begin{bmatrix} d\mathbf{U}_t \\ d\mathbf{U}_s \end{bmatrix} = \begin{bmatrix} -\frac{1}{2}\nabla^{\mathrm{Hor}}\Phi_t(\mathbf{U}_t) \\ \frac{1}{2}\nabla^{\mathrm{Hor}}\Phi_s(\mathbf{U}_s) + s^{\mathrm{Hor}}(s, \mathbf{U}_s) \end{bmatrix} \begin{bmatrix} dt \\ ds \end{bmatrix}^{\mathsf{T}} + \begin{bmatrix} H_a(\mathbf{U}_t) \\ H_a(\mathbf{U}_s) \end{bmatrix} \begin{bmatrix} \circ\, dW_t^a \\ \circ\, dW_s^a \end{bmatrix}^{\mathsf{T}}, \tag{41}$$

*Then, the **horizontal score function** $s^{\mathrm{Hor}}$ is defined as time-indexed horizontal vector fields*

$$s^{\mathrm{Hor}}(t, \mathbf{U}_t) := \nabla^{\mathrm{Hor}} \log \rho_t^{\mathrm{Hor}}(\mathbf{U}_t), \tag{42}$$

*which lies in the horizontal distribution $\mathrm{Hor}_{\mathbf{U}_t} \subset T_{\mathbf{U}_t}\mathcal{O}(\mathcal{M})$ at each point $\mathbf{U}_t \in \mathcal{O}(\mathcal{M})$. Here, $\rho_t^{\mathrm{Hor}}$ denotes a Markovian probability distribution with respect to forward frame dynamics, i.e., $\mathbf{U}_t \sim \rho_t^{\mathrm{Hor}}$.*

We end this section by integrating the preceding theoretical results to comprehensively elucidate the rationale of horizontal diffusion models. For a frame $e : \mathbb{R}^d \simeq T_x\mathcal{M}$ and a vector $w \in \mathbb{R}^d$, the horizontal lift $H_e \in \mathrm{Hor}_{\mathbf{U}:=(x,e)}$ is defined by

$$\omega(H_e) = 0, \qquad \pi_* H_e = e(w). \tag{}$$

Recall that $\mathbf{E}_t \in \mathbb{R}^d$ is a latent Euclidean diffusion satisfying the forward Stratonovich SDE

$$d\mathbf{E}_t = -\nabla\phi_t(\mathbf{E}_t)dt + \sigma_t \circ dW_t, \quad \mathbf{E}_0 \sim \rho_0. \tag{}$$

The stochastic development $\mathbf{U}_t = (x_t, e_t) \in \mathcal{O}(\mathcal{M})$ of $\mathbf{E}_t$ is defined as the solution of the Stratonovich SDE

$$\begin{aligned} d\mathbf{U}_t &= \nabla^{\mathrm{Hor}}\Phi_t(\mathbf{U}_t) + \sigma_t H_a(\mathbf{U}_t) \circ dW_t^{(a)} \\ &= H_{e_t}(\mathbf{U}_t)[-\nabla\phi(\mathbf{E}_t)]dt + \sigma_t H_a \circ dW_t^{(a)}, \end{aligned} \tag{43}$$

where the conversion between Euclidean and Horizontal potentials and their associated gradient is detailed in Proposition D.1. Specifically, the horizontal lift of $\nabla\phi_s$ evaluated at $U_s$ becomes given that $w^a = \partial_{E^a}\phi_s(E_s)$:

$$\nabla^{\mathrm{Hor}}\Phi_s(U_s) = \sum_{a=1}^d (\partial_{E^a}\phi_s(E_s)) \left( e_a^i \frac{\partial}{\partial x^i} - \Gamma_{ij}^k(x_s)e_a^j \frac{\partial}{\partial e_i^k} \right). \tag{44}$$

Alternatively, one may use the fact that the set $\{H_a\}_{a=1}^d$ forms an orthonormal basis under the Sasaki metric. Then, for any scalar function $\Phi$ on $\mathcal{O}(M)$, the horizontal gradient is expressed as:

$$\nabla^{\mathrm{Hor}}\Phi_s(U_s) = \sum_{a=1}^d (H_a\Phi_s)H_a \tag{45}$$

$$= \sum_{a=1}^d \left( e_a^i \frac{\partial\Phi_s}{\partial x^i} - \Gamma_{ij}^k(x_s)e_a^j \frac{\partial\Phi_s}{\partial e_i^k} \right) \left( e_a^m \frac{\partial}{\partial x^m} - \Gamma_{pq}^r(x_s)e_a^q \frac{\partial}{\partial e_p^r} \right). \tag{46}$$

Now, suppose the scalar function on the frame bundle is given by $\Phi_s(U_s) = \phi_s(E_s)$, where $E_s := e_s^{-1}(\dot{x}_s)$ is the Euclidean representation of the velocity. Then, by chain rule,

$$\frac{\partial\Phi_s}{\partial x^i} = \sum_{b=1}^d \frac{\partial\phi_s}{\partial E^b} \cdot \frac{\partial}{\partial x^i}(e_b^j \dot{x}^j) = \sum_b \frac{\partial\phi_s}{\partial E^b} \cdot e_{bi}, \tag{47}$$

$$\frac{\partial\Phi_s}{\partial e_i^k} = \sum_{b=1}^d \frac{\partial\phi_s}{\partial E^b} \cdot \frac{\partial}{\partial e_i^k}(e_b^j \dot{x}^j) = \sum_b \frac{\partial\phi_s}{\partial E^b} \cdot \delta_k^b \dot{x}^i = \frac{\partial\phi_s}{\partial E^k} \cdot \dot{x}^i. \tag{48}$$

Substituting into the expression for the horizontal gradient yields the final form:

$$\nabla^{\mathrm{Hor}}\Phi_s(U_s) = \sum_{a=1}^d \left( \frac{\partial\phi_s}{\partial E^a} \right) \left[ e_a^i e_a^m \frac{\partial}{\partial x^m} - \Gamma_{ij}^r(x_s)e_a^j e_a^m \frac{\partial}{\partial e_m^r} \right]. \tag{49}$$

This gives the coordinate representation of the lifted gradient field $\nabla^{\mathrm{Hor}}\Phi$ on the frame bundle, derived from the base manifold function $\phi$ through its composition with the Euclideanized velocity field $E_s$. This shows that the lifted forward process on frame-bundle $\mathbf{U}_t$ can be written in local coordinate as follows:

$$
\begin{aligned}
d\mathbf{U}_t = &\sum_{a=1}^{d}\left(\frac{\partial\phi_t}{\partial E^a}(\mathbf{E}_t)\cdot e_a^m\right)\left(e_a^i\frac{\partial}{\partial x^i} - \Gamma_{ij}^k(x_t)e_a^j\frac{\partial}{\partial e_i^k}\right)dt \\
&+ \sigma_t\sum_{a=1}^{d}\left(e_a^i\frac{\partial}{\partial x^i} - \Gamma_{ij}^k(x_t)e_a^j\frac{\partial}{\partial e_i^k}\right)\circ dW_t^{(a)}.
\end{aligned}
\tag{50}
$$

Let $\rho_t$ be the marginal distribution of $\mathbf{E}_t$ at time $t$. Then, the horizontal lift of Euclidean dynamics $\circ d\mathbf{E}_t$ can be further summarized as

$$
H_{e_t}(\mathbf{U}_t)[-\nabla\phi_s + \sigma_s^2\nabla\log\rho_s] = -\nabla^{\mathrm{Hor}}\Phi_s + \sigma_s^2\nabla^{\mathrm{Hor}}\log\rho_s^{\mathrm{Hor}}
$$

with $\rho_s^{\mathrm{Hor}}$ denoting the marginal distribution of $\mathbf{U}_s$ under the forward dynamics. As similar manner with forward dynamics, the lifted reverse Stratonovich SDE on the frame bundle becomes

$$
d\mathbf{U}_s = \left[-\nabla^{\mathrm{Hor}}\Phi_s(\mathbf{U}_s) + \sigma_s^2\nabla^{\mathrm{Hor}}\log\rho_s^{\mathrm{Hor}}(\mathbf{U}_s)\right]ds + \sigma_s H_{e_s,\varepsilon_a}(\mathbf{U}_s)\circ dW_s^{(a)},
\tag{51}
$$

where $H_e$ denotes the horizontal lift of a vector $w \in \mathbb{R}^d$ with respect to the moving frame $e_s$ at $x_s = \pi(\mathbf{U}_s)$. Proposition D.4 establishes that $\mathbf{U}_s$ constitutes the genuine time-reversal process of $\mathbf{U}_t$, and further demonstrates that the associated probability density admits a joint representation in terms of both the latent process $\mathbf{E}$ and the data process $\mathbf{U}$. Corollary D.5 subsequently characterizes the resulting data density on the manifold, showing how it emerges under canonical projection from the frame bundle. Finally, we observe that the horizontal score field $s^{\mathrm{Hor}}$ is gauge-equivariant in the sense that for any smooth gauge transformation $g : \mathcal{O}(\mathcal{M}) \to O(d)$ following by Proposition D.2,

$$
s^{\mathrm{Hor}}(t, R_g(\mathbf{U})) = dR_g\big(s^{\mathrm{Hor}}(t, \mathbf{U})\big),
\tag{52}
$$

where $R_g$ denotes the right action of $g$ on the frame bundle. This ensures that our construction is consistent with the natural $O(d)$-equivariance structure of the frame bundle geometry.

### D.2 TRAINING HORIZONTAL DIFFUSION MODELS

**Proposition D.7** (Horizontal KL Divergence). *Let $b_t(\mathbf{U}_t)$ and $\tilde{b}_t(\mathbf{U}_t^\theta)$ be horizontal drifts on the frame bundle $\mathcal{O}(M)$ with path laws $\boldsymbol{\nu}_b$ and $\boldsymbol{\nu}_{\tilde{b}}$, and let $\mathbf{P}_{\mathrm{Hor}}(Z) := Z - \omega(Z)^\sharp \cdot U$ project[a] to the horizontal subspace $\mathrm{Hor}_U \subset T_U\mathcal{O}(M)$ via the Levi–Civita connection $\omega$, and $\|\cdot\|_H$ be the Sasaki metric norm on $H_U$, $\Gamma^{\mathcal{O}(\mathcal{M})}$ be parallel transport. Then, the horizontal KL divergence is given by*

$$\mathbf{KL}_{\mathrm{Hor}}[\boldsymbol{\nu}_{\tilde{b}}\|\boldsymbol{\nu}_b] = \frac{1}{2}\mathbb{E}_{\boldsymbol{\nu}_{\tilde{b}}}\left[\int_0^T \|\mathbf{P}_{\mathrm{Hor}}(\Delta_t)\|_H^2\, dt\right], \quad \Delta_t := \Gamma^{\mathcal{O}(M)}_{\mathbf{U}_t^\theta \to \mathbf{U}_t}\left(\tilde{b}_t(\mathbf{U}_t^\theta)\right) - b_t(\mathbf{U}_t),$$

---

[a]The term $\omega(Z)^\sharp$ refers to the fundamental vertical vector field on the frame bundle associated with the Lie algebra element $\omega(Z) \in \mathfrak{o}(d)$ Kobayashi & Nomizu (1963b).

*Proof.* Fix a latent Euclidean path $E_\bullet \in C([0,T], \mathbb{R}^d)$ satisfying the solution to SDEs in equation 1, and $\mathbf{E}_t$ its canonical frame representation. The forward horizontal Stratonovich SDE on $\mathcal{O}(M)$ is

$$d\mathbf{U}_t = -\nabla^{\mathrm{Hor}}\Phi_t(\mathbf{U}_t)dt + \sigma_t H_{\mathbf{E}_t, \varepsilon_a}(\mathbf{U}_t) \circ dW_t^{(a)}, \qquad \mathbf{U}_0 \sim \mu_0. \tag{53}$$

whose path law $\nu_{[0,T]} := \mathbf{Law}_{\mathbb{P}}(U_\bullet)$ lives on $C([0,T], \mathcal{O})$. Introducing a learnable horizontal score function $s_\theta^{\mathrm{Hor}} : [0,T] \times \mathcal{O} \to H$ we define the reverse Stratonovich SDE

$$dU_s = \left[-\nabla^{\mathrm{Hor}}\Phi_s(U_s) + \sigma_s s^{\mathrm{Hor}}(s, U_s)\right]ds + \sqrt{\sigma_s} H_{e_s, \varepsilon_a}(U_s) \circ d\widetilde{W}_s^{(a)}, \qquad U_T \sim \nu_T. \tag{54}$$

yielding a new path law $\tilde{\nu}_{[0,T]} := \mathbf{Law}_{\tilde{\mathbb{P}}}(U_\bullet)$. To compare reverse and forward probabilistic representations, *i.e.*, $\tilde{\nu}_{[0,T]}$ and $\nu_{[0,T]}$, we first pass from Stratonovich SDEs to Itô form:

$$d\mathbf{U}_t = \begin{cases} \left(-\nabla^{\mathrm{Hor}}\Phi_t(\mathbf{U}_t) + \sigma_t^2 \mathcal{V}_t(\mathbf{U}_t)\right)dt + \sigma_t H_a(\mathbf{U}_t)dW_t^{(a)}, & \text{(forward)}, \\[2mm] \left(-\nabla^{\mathrm{Hor}}\Phi_t(\mathbf{U}_t) - \sigma_t^2 s^{\mathrm{Hor}}(t, \mathbf{U}_t) + \sigma_t^2 \mathcal{V}_t(\mathbf{U}_t)\right)dt + \sigma_t H_a(\mathbf{U}_t)d\widetilde{W}_t^{(a)}, & \text{(reverse)}, \\[2mm] \left(-\nabla^{\mathrm{Hor}}\Phi_t(\mathbf{U}_t) - \sigma_t^2 s_\theta^{\mathrm{Hor}}(t, \mathbf{U}_t) + \sigma_t^2 \mathcal{V}_t(\mathbf{U}_t)\right)dt + \sigma_t H_a(\mathbf{U}_t)d\widehat{W}_t^{(a)}, & \text{(denoiser)} \end{cases}$$

where the horizontal gradient is given by

$$\mathcal{V}_t(\mathbf{U}_t) := \nabla_{H_a}^{O(\mathcal{M})} H_a(\mathbf{U}_t). \tag{55}$$

This covariant derivative on the frame-bundle (*e.g.*, $\nabla^{O(\mathcal{M})}$) can be written in local coordinate with the following form:

$$\nabla_{H_a}^{O(\mathcal{M})} H_a = \left[(\partial_{x^r} e_a^i)e_a^r - e_a^i \partial_{x^i}(\Gamma_{mj}^k e_a^j)\right]\partial_{x^i} - (\partial_{x^i}\Gamma_{mj}^k)e_a^i e_a^j \partial_{e_m^k}. \tag{56}$$

Subsequently, a true score function of forward dynamics is given by

$$s^{\mathrm{Hor}}(t, \mathbf{U}_t) = \sum_{a=1}^d \left(H_a \log \rho_t^{\mathrm{Hor}}(\mathbf{U}_t)\right) \cdot H_a(\mathbf{U}_t), \quad \mathbf{E}_t := e_t^{-1} \circ \pi_* \nabla^{\mathrm{Hor}}\Phi(\mathbf{U}_t). \tag{57}$$

For simplicity, let us denote $\tilde{b}_t := -\sigma_t^2 s^{\mathrm{Hor}}(t, \mathbf{U}_t) + \sigma_t^2 \mathcal{V}_t(\mathbf{U}_t)$ and $\tilde{b}_t^\theta := -\sigma_t^2 s_\theta^{\mathrm{Hor}}(t, \mathbf{U}_t) + \sigma_t^2 \mathcal{V}_t(\mathbf{U}_t)$, respectively. Then, our aim is to derive the KL divergence between path-measures $\tilde{\nu}_\bullet$ and $\tilde{\nu}_\bullet^\theta$ For this, let us consider the change of measure between the dynamics defined on the orthonormal frame bundle $\mathcal{O}(M)$. The true reverse SDE is driven by the horizontal score function $s^{\mathrm{Hor}}(t, \mathbf{U}_t)$, while the learned reverse SDE is parameterized by $s_\theta^{\mathrm{Hor}}(t, \mathbf{U}_t^\theta)$. These induce drift vector fields $\tilde{b}_t(\mathbf{U}_t)$ and $\tilde{b}_t^\theta(\mathbf{U}_t^\theta)$ respectively. Since the two drifts are evaluated at different frame points $\mathbf{U}_t, \mathbf{U}_t^\theta \in \mathcal{O}(M)$, we align them via parallel transport. Specifically, we define the drift discrepancy

$$\Delta_t := \Gamma^{\mathcal{O}(M)}_{\mathbf{U}_t^\theta \to \mathbf{U}_t}\left(\tilde{b}_t^\theta(\mathbf{U}_t^\theta)\right) - \tilde{b}_t(\mathbf{U}_t),$$

where $\Gamma^{\mathcal{O}(M)}_{\mathbf{U}_t^\theta \to \mathbf{U}_t}$ denotes the parallel transport operator on $\mathcal{O}(M)$ along a horizontal curve connecting $\mathbf{U}_t^\theta$ to $\mathbf{U}_t$. This ensures that $\Delta_t \in T_{\mathbf{U}_t}\mathcal{O}(M)$ is a well-defined comparison of vector fields at the

same point. To preserve horizontal gauge-invariance, we apply the horizontal projection operator $\mathbf{P}_{\text{Hor}} : T_{\mathbf{U}_t}\mathcal{O}(M) \to H_{\mathbf{U}_t}$ and define the filtered discrepancy

$$\widehat{\Delta}_t := \mathbf{P}_{\text{Hor}}(\Delta_t) = \Delta_t - \omega(\Delta_t)^\sharp \cdot \mathbf{U}_t,$$

where $\omega$ is the connection 1-form and $\omega(\Delta_t)^\sharp$ denotes the fundamental vertical lift in the sense of Kobayashi-Nomizu. Let $g_H$ denote the pull-back of the Riemannian metric $g$ on the base manifold $M$ to the horizontal bundle $H\mathcal{O}(M)$ via the canonical projection $\pi : \mathcal{O}(M) \to M$. Explicitly, for any $U \in \mathcal{O}(M)$ and any horizontal vectors $V, W \in H_U \subset T_U\mathcal{O}(M)$, we define

$$g_H(V, W) := g[\pi(U)]\left(\pi_* V, \pi_* W\right), \quad \text{so that} \quad \|V\|_H^2 := g_H(V, V).$$

Then, all terms in the Girsanov distribution can now be expressed using $\widehat{\Delta}_t$ alone. Define the horizontal Girsanov exponential by

$$\widehat{Z}_T^H = \exp\left(-\int_0^T \frac{g_H\left(\widehat{\Delta}_t, \sum_{a=1}^d H_a(\mathbf{U}_t) \circ dW_t^{(a)}\right)}{\sigma_t} - \frac{1}{4}\int_0^T \frac{\|\widehat{\Delta}_t\|_H^2}{\sigma_t^2} dt\right),$$

where $g_H$ is the Sasaki metric restricted to the horizontal bundle and we assumed that the condition $\mathbb{E}\exp\left(\frac{1}{2}\int_0^T \frac{\|\widehat{\Delta}_t\|_H^2}{2\sigma_t^2} dt\right) < \infty$ holds. This distribution satisfies the Radon–Nikodym identity

$$\widehat{Z}_T^H(U_\bullet) = \frac{d\tilde{\nu}_{[0,T]}^\theta}{d\tilde{\nu}_{[0,T]}}(U_\bullet),$$

and therefore the horizontal Kullback-Leibler divergence is given by

$$\mathbf{KL}_{\text{Hor}}\left(\tilde{\nu}_{[0,T]}^\theta \| \tilde{\nu}_{[0,T]}\right) = \mathbb{E}_{\tilde{\nu}_{[0,T]}}\left[\log \widehat{Z}_T^H\right] = \frac{1}{2}\int_0^T \mathbb{E}_{\tilde{\nu}_{[0,T]}}\left[\left\|\widehat{\Delta}_t\right\|_H^2\right] dt.$$

In this form, the KL divergence compares the true and learned reverse processes intrinsically within the geometry of the frame bundle $\mathcal{O}(M)$, with all drift differences consistently aligned via parallel transport and projected horizontally.

To explicitly connect the geometric Girsanov distribution to its Euclidean counterpart, we recall that the horizontal Stratonovich SDE on $\mathcal{O}(M)$ admits a unique strong solution for each realization of the latent path $E_\bullet \in C([0,T], \mathbb{R}^d)$ due to the global Lipschitz continuity of $H_{\mathbf{E}_t, \varepsilon_a}$. The horizontal lifting operator $\Psi : C([0,T], \mathbb{R}^d) \to C([0,T], \mathcal{O}(M))$ then defines a measurable transformation that preserves filtrations:

$$\mathcal{F}_t^E = \sigma(E_s : 0 \le s \le t) \quad \to \quad \mathcal{F}_t^U = \sigma(\mathbf{U}_s : 0 \le s \le t) = \sigma(\Psi(E_s) : 0 \le s \le t).$$

Specifically, by the assumption that this mapping is measurable and almost surely injective for the fixed deterministic initial frame $\mathbf{U}_0 \in \mathcal{O}(M)$, with a measurable left-inverse $\Psi^{-1}(\mathbf{U}_\bullet) = \left(e_t^{-1}\pi_*\dot{\mathbf{U}}_t\right)_{t\in[0,T]}$, we have $\mathcal{F}_t^E = \mathcal{F}_t^U$. We define the Euclidean path measures as push-forwards through $\Psi^{-1}$:

$$\mu_{[0,T]} := \mathbb{P} \circ \Psi^{-1}, \quad \tilde{\mu}_{[0,T]} := \tilde{\mathbb{P}} \circ \Psi^{-1}, \quad \tilde{\mu}_{[0,T]}^\theta := \tilde{\mathbb{P}}^\theta \circ \Psi^{-1}.$$

Because the Girsanov distribution $\hat{Z}_T^H(U_\bullet)$ is $\mathcal{F}_T^U$-measurable and $U_\bullet = \Psi(E_\bullet)$, it lifts through the change of variables:

$$\hat{Z}_T^H(U_\bullet) = \frac{d\tilde{\nu}_{[0,T]}^\theta}{d\tilde{\nu}_{[0,T]}}(U_\bullet) = \frac{d\tilde{\mu}_{[0,T]}^\theta}{d\tilde{\mu}_{[0,T]}}(E_\bullet).$$

This identity shows that the horizontal Girsanov distribution on the frame bundle $\mathcal{O}(M)$ pulls back to the standard Girsanov distribution on $\mathbb{R}^d$, evaluated at the latent Euclidean path $E_\bullet$. In particular, if the learned reverse path law $\tilde{\nu}_{[0,T]}^\theta$ is generated by lifting $\tilde{\mu}_{[0,T]}^\theta$ through $\Psi$, we have

$$\hat{Z}_T^H(\Psi(E_\bullet)) = \frac{d\tilde{\mu}_{[0,T]}^\theta}{d\tilde{\mu}_{[0,T]}}(E_\bullet),$$

thus justifying the interpretation of $Z_T^H$ as a pathwise distribution simultaneously defined on $C([0,T], \mathbb{R}^d)$ and $C([0,T], \mathcal{O}(M))$ via the lifting map $\Psi$. We now compute the horizontal KL divergence in terms of the coefficient expansion of the parameterized score.

The true and learnable horizontal score functions $s_\theta^{\mathrm{Hor}}(t, U)$ admit the basis expansion in terms of the moving frame $\{H_{\mathbf{E}_t, \varepsilon_a}\}_{a=1}^d$ as

$$s^{\mathrm{Hor}}(t, U) = H_{\mathbf{E}_t, \mathbf{1}^\intercal s}(U) = \nabla \log \varrho_t \cdot H_{\mathbf{E}_t, \varepsilon_a}(U) = -\Sigma_t \nabla \phi(\mathbf{E}_t) \cdot H_{\mathbf{E}_t, \varepsilon_a}(U), \tag{58}$$

$$s_\theta^{\mathrm{Hor}}(t, U) = H_{\mathbf{E}_t, \mathbf{1}^\intercal s_\theta}(U) = s_\theta^a(t, \mathbf{E}_t) H_{\mathbf{E}_t, \varepsilon_a}(U), \tag{59}$$

where the Einstein summation convention is assumed over repeated indices. Then the drift discrepancy between the learned and true reverse processes becomes

$$\Delta_t := \Gamma_{\mathbf{U}_t^\theta \to \mathbf{U}_t}^{\mathcal{O}(M)} \left( \tilde{b}_t^\theta(\mathbf{U}_t^\theta) \right) - \tilde{b}_t(\mathbf{U}_t) \tag{60}$$

$$= -2\sigma_t^2 \left( \Gamma_{\mathbf{U}_t^\theta \to \mathbf{U}_t}^{\mathcal{O}(M)}(s^{\mathrm{Hor}}) - s_\theta^{\mathrm{Hor}} \right)(t, \mathbf{U}_t) \tag{61}$$

$$= -2\sigma_t^2 \sum_a (s^a(t, \mathbf{U}_t) - s_\theta^a(t, \mathbf{U}_t)) H_{\mathbf{E}_t, \varepsilon_a}(\mathbf{U}_t). \tag{62}$$

Since the drift discrepancy $\Delta_t := \tilde{b}_t - \tilde{b}_t^\theta$ is defined as the difference between two horizontally lifted vector fields, it naturally lies in the horizontal bundle $H_{\mathbf{U}_t} \subset T_{\mathbf{U}_t} \mathcal{O}(M)$. Therefore, the application of the horizontal projection operator $\mathbf{P}_{\mathrm{Hor}}$ is algebraically trivial $\widehat{\Delta}_t := \mathbf{P}_{\mathrm{Hor}}(\Delta_t) = \Delta_t$.

Nevertheless, we retain the notation explicitly to emphasize that only horizontal components are ever relevant in the Girsanov distribution, and that our construction is intrinsically gauge-invariant by design. Next, to evaluate the Radon–Nikodym distribution $Z_T^H$, we first compute the squared norm of the drift difference $\Delta_t$ under the Sasaki metric $g_H$. By Using linearity of $g_H$ and the orthonormality above, we compute

$$\|\widehat{\Delta}_t\|_H^2 = g_H(\Delta_t, \Delta_t)$$

$$= 4\sigma_t^2 \sum_{a,b=1}^d \left( s^a(t, \mathbf{U}_t) - s_\theta^a(t, \mathbf{U}_t) \right) \left( s^b(t, \mathbf{U}_t) - s_\theta^b(t, \mathbf{U}_t) \right) g_H \left( H_{\mathbf{E}_t, \varepsilon_a}, H_{\mathbf{E}_t, \varepsilon_b} \right)$$

$$= 4\sigma_t^2 \sum_{a=1}^d |s^a(t, \mathbf{U}_t) - s_\theta^a(t, \mathbf{U}_t)|^2.$$

This quadratic form serves as the regularization term in the log-distribution and will directly determine the scale of the Kullback–Leibler divergence. Since $\pi_* H_a(\mathbf{U}_t) = e_t(\varepsilon_a) \in T_{\pi(\mathbf{U}_t)} M$, the third line follows by the fact:

$$g_H\big(H_a, H_b\big) = g\big(\pi_* H_a, \pi_* H_b\big) = g\big(e_t(\varepsilon_a), e_t(\varepsilon_b)\big) = \langle \varepsilon_a, \varepsilon_b \rangle_{\mathbb{R}^d} = \delta_{ab},$$

where the last equality holds as $e_t$ is an orthonormal frame. We then compute the stochastic inner product appearing in the Girsanov exponential. The horizontal noise process in the lifted SDE is expressed via the Stratonovich increment $\circ dB_t := \sigma_t^2 \sum_{b=1}^d H_b(\mathbf{U}_t) \circ dW_t^{(b)}$, which is a linear combination of orthonormal horizontal frame fields modulated by Euclidean Brownian noise. Taking the inner product of $\Delta_t$ with this increment using the Sasaki metric yields:

$$g_H(\widehat{\Delta}_t, \circ dB_t) = -(2\beta)^{3/2} \sum_{a=1}^d (s^a(t, \mathbf{U}_t) - s_\theta^a(t, \mathbf{U}_t)) \circ dW_t^{(a)}.$$

This term reflects the interaction between the residual score and the random fluctuations of the process; however, due to martingale properties, it will integrate out under expectation when computing the KL divergence. Combining both the quadratic and stochastic terms, the Girsanov–Cameron–Martin–type exponential distribution becomes

$$Z_T^H = \exp \Bigg( -\sigma_t \int_0^T \sum_{a=1}^d (s^a(t, \mathbf{U}_t) - s_\theta^a(t, \mathbf{U}_t)) \circ dW_t^{(a)}$$

$$-\frac{1}{2} \int_0^T \sigma_t^2 \sum_{a=1}^d |s^a(t, \mathbf{U}_t) - s_\theta^a(t, \mathbf{U}_t)|^2 dt \Bigg). \tag{63}$$

This expression encapsulates the full pathwise deviation between the learned and true horizontal dynamics, measured along the noise directions defined by the frame bundle. Since the stochastic integral has mean zero under the reference measure $\tilde{\nu}_{[0,T]}$ (as it represents a martingale term), the expected log-distribution simplifies cleanly to the deterministic $L^2$ error between the true and parameterized scores:

$$\mathbf{KL}_{\mathrm{Hor}}(\tilde{\nu}^{\theta}_{[0,T]} \| \tilde{\nu}_{[0,T]}) = \frac{1}{2} \int_0^T \sigma_t^2 \mathbb{E}_{\tilde{\nu}_{[0,T]}} \left[ \sum_{a=1}^{d} |s_\theta^a(t, \mathbf{U}_t) - s^a(t, \mathbf{U}_t)|^2 \right] dt$$

This final form reveals the essential structure of score-matching in the geometric setting: the KL divergence is minimized precisely when the learned score $s_\theta^a(t, U)$ matches the true score $s^a(t, U)$ in the $L^2$ sense along each horizontal direction $H_{e_t, \varepsilon_a}$. The weighting by $\sigma_t^2$ encodes the diffusion strength at each time $t$, and the geometric structure of the orthonormal frame ensures that the entire loss remains invariant to the choice of local coordinates. This result faithfully generalizes the Euclidean score-matching loss to the orthonormal frame bundle of a Riemannian manifold. $\square$

**Theorem D.8** (Uniform Generalization Bound for Worst-Case Horizontal KL). *Assume the neural score networks are parameterized as $L$-layer, width $W$ feedforward ReLU networks with spectral norm bounds $s_\ell$, $(2,1)$-norm bounds $b_\ell$, activation Lipschitz constants $\rho_\ell$. Assume the manifold $\mathcal{M}$ has sectional curvature $|K(\pi)| \leq \kappa_{\max}$ and diameter $\mathrm{Diam}(\mathcal{M})$; noise scales $\sigma_{t_m}$ are bounded below by $\sigma_{\min}$. Let $n$ be the number of samples per time, $M$ the number of time steps, $D$ the maximal function class diameter. Then for any $\delta \in (0,1)$, with probability at least $1 - \delta$, the following holds:*

$$\sup_{\theta \in \Theta} \mathrm{KL}_{\mathrm{Hor}}(\tilde{\nu}^{\theta}_{[0,T]} \| \tilde{\nu}_{[0,T]}) \leq \frac{6 C_{\mathcal{M}}(\sigma_{\min}, \kappa_{\max}, \mathrm{Diam}(\mathcal{M}), D) \cdot C_{\mathrm{NN}}(L, W, s_\ell, b_\ell, \rho_\ell, D)}{n^{1/4}}$$

$$+ \sqrt{\frac{C_{\mathrm{SubG}}^2(\sigma_{t_m}, L_{t_m}^{\mathrm{tot}}(\theta), \lambda_{\max}(\Sigma_{t_m})) \log(1/\delta)}{4nM}}, \quad (64)$$

*where Eq equation 89 provides full description of constants $C_{\mathcal{M}}$, $C_{\mathrm{NN}}$ and $C_{\mathrm{SubG}}$.*

**Remark.** *The result states for the case where the neural network score models are drawn from the classical family of $L$-layer, width-$W$ feedforward ReLU networks with uniformly bounded spectral norms and $(2,1)$-norms, for which explicit covering number bounds are available via classical results Bartlett et al. (2017). However, the generalization argument and the overall structure of the proof are not tied to these specific architectural assumptions. In particular, the identical analysis applies for any hypothesis class $\mathfrak{F}$ whose empirical covering numbers can be suitably bounded. The key requirement is the ability to upper-bound the entropy number of the hypothesis class with respect to the empirical $L_2$ metric induced by the data distribution.*

*Proof.* As a first step, we introduce the main function classes that play a central role in our analysis. Let $\mathfrak{F}$ denote the class of parameterized functions given by the neural score models at each time step, measuring the (scaled) difference between the learned score and the reference (true) score:

$$\mathfrak{F} := \left\{ f_{\theta,m}(E) = \sqrt{\sigma_{t_m}} \left[ s_\theta(t_m, E) - s(t_m, E) \right] : \theta \in \Theta, \ m = 1, \ldots, M \right\} \subset \mathbb{R}^d, \quad (65)$$

where $\theta$ indexes the neural network weights, $s_\theta$ is the learned score network, and $s$ is the true oracle score. Next, we define the class $\mathfrak{G}$ as the set of squared horizontal-norm evaluations, obtained by applying the horizontal lift operator $H_{e_{t_m}}$ to elements of $\mathfrak{F}$ and taking the squared norm in the Sasaki metric:

$$\mathfrak{G} := \left\{ g_{\theta,m}(E) = \left\| H_{e_{t_m}} [f_{\theta,m}(E)] \right\|_H^2 : f_{\theta,m} \in \mathfrak{F} \right\}. \quad (66)$$

To quantify the generalization properties of the learned model, we introduce the population and empirical risks associated with these lifted functions:

$$\mathcal{R}^{\mathrm{Hor}}(\theta) = \frac{1}{T} \sum_{m=1}^{M} \mathbb{E}_{E_{t_m} \sim \rho_{t_m}} \left[ g_{\theta,m}(E_{t_m})^2 \right], \quad \widehat{\mathcal{R}}_n^{\mathrm{Hor}}(\theta) = \frac{1}{nM} \sum_{m=1}^{M} \sum_{i=1}^{n} \left[ g_{\theta,m}(E_{t_m}^{(i)})^2 \right]. \quad (67)$$

Our main focus is to control the worst-case deviation of the horizontal KL divergence, which, in the context of lifted score matching, is bounded above by the maximal population risk over all networks and time indices:

$$\sup_{\theta \in \Theta} \mathbf{KL}_{\mathrm{Hor}}\big(\tilde{\nu}_{[0,T]}^{\theta} \| \tilde{\nu}_{[0,T]}\big) \leq \frac{1}{2} \sup_{g_{\theta,m} \in \mathcal{G}} \mathcal{R}^{\mathrm{Hor}}(\theta). \tag{68}$$

In what follows, we provide a non-asymptotic generalization bound for this quantity, making explicit the dependence on both the network parameterization and the geometry of the underlying manifold. To control the generalization gap, we invoke the symmetrization argument. For any $\theta$, by introducing an independent ghost sample $(E_{t_m}^{(i)\prime})$, we can write

$$\mathbb{E}\left|\widehat{\mathcal{R}}_n(\theta) - \mathcal{R}(\theta)\right| = \mathbb{E}_E \left|\frac{1}{nM} \sum_{m=1}^{M} \sum_{i=1}^{n} \Big(g_\theta(t_m, E_{t_m}^{(i)}) - \mathbb{E}_{E_{t_m}}[g_\theta(t_m, E_{t_m})]\Big)\right| \tag{69}$$

$$\leq 2\mathbb{E}_{E,\varepsilon} \left|\frac{1}{nM} \sum_{m=1}^{M} \sum_{i=1}^{n} \varepsilon_{mi} g_\theta(t_m, E_{t_m}^{(i)})\right|, \tag{70}$$

where $(\varepsilon_{mi})$ are i.i.d. Rademacher random variables. Taking the supremum over $\theta$ and applying linearity of expectation, we obtain

$$\mathbb{E}\sup_\theta \left|\widehat{\mathcal{R}}_n(\theta) - \mathcal{R}(\theta)\right| \leq 2\mathbb{E}_{E,\varepsilon} \sup_\theta \left|\frac{1}{nM} \sum_{m=1}^{M} \sum_{i=1}^{n} \varepsilon_{mi} g_\theta(t_m, E_{t_m}^{(i)})\right|. \tag{71}$$

We define the empirical Rademacher complexity

$$\mathfrak{R}_{n,M} := \mathbb{E}_\varepsilon \sup_\theta \left|\frac{1}{nM} \sum_{m=1}^{M} \sum_{i=1}^{n} \varepsilon_{mi} g_\theta(t_m, E_{t_m}^{(i)})\right|, \quad \sup_\theta \left|\widehat{\mathcal{R}}_n(\theta) - \mathcal{R}(\theta)\right| \leq 2\mathfrak{R}_{n,M}, \tag{72}$$

where the expectation is over both the data samples and the Rademacher variables. Let us assume that $s_\theta(t_m, \cdot) : \mathbb{R}^d \to \mathbb{R}^d$ is $L_{t_m}(\theta)$–Lipschitz such that $\|s_\theta(t_m, x) - s_\theta(t_m, y)\| \leq L_{t_m}(\theta)\|x - y\|$. Following by the fact that the density for the forward dynamics is truly a mean-zero Gaussian, $\rho_t = \mathcal{N}(0, \Sigma)$, for the auxiliary function $h_{\theta,m}(x) := s_\theta(t_m, x) - s_\star(t_m, x)$, we have

$$\|h_{\theta,m}(x) - h_{\theta,m}(y)\| \leq L_{t_m}(\theta)\|x - y\| + \|\Sigma_{t_m}^{-1}\|\|x - y\| =: L_{t_m}^{\mathrm{tot}}(\theta)\|x - y\|. \tag{73}$$

While the horizontal lift $H_{e_{t_m}} : \mathbb{R}^d \to \mathrm{Hor}_{U_{t_m}}$ is an isometry under the Sasaki metric, we have $\|H_{e_{t_m}}[w]\|_H = \|w\|$ for any $w \in \mathbb{R}^d$. This shows that, for any $x$ and $y$, we have

$$\left|g_{\theta,m}(x) - g_{\theta,m}(y)\right| = \sqrt{\sigma_{t_m}} \Big|\|H_{e_{t_m}} h_{\theta,m}(x)\|_H - \|H_{e_{t_m}} h_{\theta,m}(y)\|_H\Big| \leq \sqrt{\sigma_{t_m}} L_{t_m}^{\mathrm{tot}}(\theta)\|x - y\|.$$

The inequality shows that $g_{\theta,m}$ is $\sqrt{\sigma_{t_m}} L_{t_m}^{\mathrm{tot}}(\theta)$-Lipschitz with respect to the Euclidean norm. Next, consider the Gaussian isoperimetric inequality for a Lipschitz function $f : \mathbb{R}^d \to \mathbb{R}$ and a Gaussian vector $Z_t \sim \mathcal{N}(0, \Sigma_t)$ defined as follows:

$$\mathbb{P}\big(|f(Z_T) - \mathbb{E}f(Z_T)| \geq \mathrm{r}\big) \leq 2\exp\left(\frac{-\mathrm{r}^2}{2L_f^2 \|\Sigma_t^{1/2}\|^2}\right), \qquad t > 0.$$

Applying this with $f = g_{\theta,m}$ and $\|\Sigma_{t_m}^{1/2}\| = \sqrt{\lambda_{\max}(\Sigma_{t_m})}$ gives

$$\mathbb{P}\big(|g_{\theta,m} - \mathbb{E}g_{\theta,m}| \geq \mathrm{r}\big) \leq 2\exp\left(\frac{-\mathrm{r}^2}{2\sigma_{t_m}\big(L_{t_m}^{\mathrm{tot}}(\theta)\big)^2 \lambda_{\max}(\Sigma_{t_m})}\right).$$

The right-hand side is the tail of a centered sub-Gaussian variable (Vershynin (2018), Proposition. 2.6.3). Since each $g_{\theta,m}(E)$ is sub-Gaussian in its input by the Hanson–Wright inequality and the Gaussian structure of $E$, the function class $\mathcal{G}$ is uniformly sub-Gaussian. By Bousquet's version of Talagrand's concentration inequality (Vershynin, 2018, Theorem. 4.8.1), we have

$$\mathbb{P}\left[\sup_\theta \left|\widehat{\mathcal{R}}_n(\theta) - \mathcal{R}(\theta)\right| > \mathfrak{R}_{n,M} + c\sqrt{\sigma_{t_m}\big(L_{t_m}^{\mathrm{tot}}(\theta)\big)^2 \lambda_{\max}(\Sigma_{t_m})} \cdot \sqrt{\frac{\log(1/\delta)}{nM}}\right] \leq \delta, \tag{74}$$

where $\mathfrak{R}_{n,M}$ is the empirical Rademacher complexity, $\sigma$ is the (sub-)Gaussian parameter, and $c > 0$ is some absolute constant. Next, we want to estimate the geometric impact of manifolds to function class $\mathcal{G}$. Let $U = (x, e) \in O(\mathcal{M})$ with $e = (e_1, \dots, e_d)$ an orthonormal frame at $x \in \mathcal{M}$. For the Euclidean vector $w = (w^1, \dots, w^d) \in \mathbb{R}^d$, recall that the horizontal lift is defined as

$$H_e[w] = \left( e_j w^j, -\Gamma_{ij}^k(x) e_k e_\ell^i w^\ell \right) \in T_x\mathcal{M} \oplus \mathfrak{so}(T_x\mathcal{M}),$$

where $\Gamma_{ij}^k$ are the Christoffel symbols and $e_\ell^i$ the chart components of $e_\ell$. For a tangent pair $(X, A)$ with $X \in T_x\mathcal{M}$ and $A \in \mathfrak{so}(T_x\mathcal{M})$ the Sasaki metric is defined as

$$\left\| (X, A) \right\|_H^2 := g_x(X, X) + \sum_{a=1}^d g_x\left( Ae_a, Ae_a \right). \tag{75}$$

Because $(e_1, \dots, e_d)$ is orthonormal frame, we know that $g_x(e_a, e_b) = \delta_{ab}$. With vector field $X = e_j v^j$ and component of Riemannian metric $g_{pq} = g(\partial_{x^p}, \partial_{x^q})$, the following can be represented:

$$g_x(X, X) = g_{pq} e_j^p e_\ell^q w^j w^\ell.$$

From above result and the definition of $A$, we have $A_v : T_x\mathcal{M} \to T_x\mathcal{M}, A_v(e_a) = -\Gamma_{ij}^k e_k e_a^i w^j$. This gives the full local expression of second term in Eq equation 75

$$\sum_{a=1}^d g_x\left( A_w e_a, A_w e_a \right) = g_{pq}\Gamma_{ij}^p\Gamma_{mn}^q e_a^i e_a^m w^j w^n = g_{pq}\Gamma_{ij}^p\Gamma_{mn}^q e_j^i e_\ell^m w^j w^\ell, \tag{76}$$

where we used $e_a^i e_a^m = \delta^{im}$ and re-labeled indices. Combining results, the squared norm of horizontal lift under the Sasaki metric can be rewritten in local coordinate:

$$\|H_e[w]\|_H^2 = g_{pq} e_j^p e_\ell^q w^j w^\ell + g_{pq}\Gamma_{ij}^p\Gamma_{mn}^q e_j^i e_\ell^m w^j w^\ell. \tag{77}$$

Because $\{e_a\}$ is orthonormal, we have $g_{pq} e_j^p e_\ell^q = \delta_{j\ell}$, so the first term is $\|v\|_g^2 = \|ew\|_g^2$. For the ease of computational complexity, we fix a reference point $x_0 \in \mathcal{M}$ and introduce *normal coordinates* $(x^1, \dots, x^d)$ centered at $x_0$. At this point the metric and its first derivatives take the canonical values $g_{ij}(x_0) = \delta_{ij}, \partial_k g_{ij}(x_0) = 0$, and $\Gamma_{ij}^k(x_0) = 0$. Then, one has the Taylor expansions for $x$ with geodesic distance $r = d(x, x_0) \le R$ (Riemann normal coordinate formula) as follows:

$$\Gamma_{ij}^k(x) = -\frac{1}{3}\left( R_{ijl}^k + R_{jil}^k \right) x^l + \mathcal{O}(|x|^2), \quad g_{ij}(x) = \delta_{ij} - \frac{1}{3} R_{ikjl} x^k x^l + \mathcal{O}(|x|^3), \tag{78}$$

where $R_{ijl}^k(x_0)$ denotes the Riemann curvature tensor evaluated at fixed point $x_0$. Assume that we have the upper-bounded sectional curvatures $\le \kappa_{\max}$ on some geodesic ball $B_R(x_0) \subset \mathcal{M}$. Equivalently, this shows that $|R_{abcd}(x_0)| \le \kappa_{\max}$. Inserting this constraints into the first expansion gives the uniform bound $\left| \Gamma_{ij}^k(x) \right| \le \frac{2}{3}\kappa_{\max} r$ for all $r \le R$. Then the direct computation follows:

$$g_{pq}(x)\Gamma_{ij}^p(x)\Gamma_{mn}^q(x) e_j^i e_\ell^m w^j w^\ell$$

$$= g_{pq}\left[ -\frac{1}{3}(R_{ijl}^p + R_{jil}^p) x^l \right]\left[ -\frac{1}{3}(R_{mnr}^q + R_{nmr}^q) x^r \right] e_j^i e_\ell^m w^j w^\ell$$

$$= \frac{1}{9} g_{pq}(R_{ijl}^p + R_{jil}^p)(R_{mnr}^q + R_{nmr}^q) x^l x^r e_j^i e_\ell^m w^j w^\ell \tag{79}$$

$$\le \frac{(2\kappa_{\max})^2}{9} g_{pq} r^2 e_j^i e_\ell^m w^j w^\ell = \frac{4\kappa_{\max}^2}{9} r^2 \|v\|_g^2, \quad v = ew.$$

Combining results, we have

$$\|H_e[w]\|_H^2 \le \left( 1 + \frac{4\kappa_{\max}^2}{9} r^2 \right) \|v\|_g^2, \quad v = ew. \tag{80}$$

This geometric inequality

$$g_{\theta,m}(\theta) = \|H_{e_{t_m}}[f_{\theta,m}(E) \cdot \sigma_{t_m}^{-1/2}]\|_H^2 \le \left( 1 + \frac{4\kappa_{\max}^2}{9} r^2 \right) \left\| e_{t_m} \cdot \left( f_{\theta,m}(E) \cdot \sigma_{t_m}^{-1/2} \right) \right\|_g^2$$

$$\le \left( 1 + \frac{4\kappa_{\max}^2}{9}\mathrm{Diam}^2(\mathcal{M}) \right) \left\| e_{t_m} \cdot \left( f_{\theta,m}(E) \cdot \sigma_{t_m}^{-1/2} \right) \right\|_g^2 \tag{81}$$

where we used the fact that the radius of geodesic ball $B_R(x_0)$ is always less than $\mathrm{Diam}(\mathcal{M})$ in last inequality.

$$\mathfrak{G} \subset \left(1 + \frac{4\kappa_{\max}^2}{9}r^2\right) \cdot \left\{\frac{\|e_{t_m}f\|_g^2}{\sigma_{t_m}} : f \in \mathfrak{F}\right\}. \tag{82}$$

Let $d_{P_{nM}}$ denote the empirical $L_2$-metric on $\mathfrak{F}$ induced by the sample $\{E_{t_m}^{(i)}\}_{m,i}$ over the joint sample, i.e.,

$$d_{P_{nM}}(f, \tilde{f}) := \left(\frac{1}{nM}\sum_{m=1}^{M}\sum_{i=1}^{n}\|f(t_m, E_{t_m}^{(i)}) - \tilde{f}(t_m, E_{t_m}^{(i)})\|^2\right)^{1/2}, \tag{83}$$

Now, we define another function class, which has members of scaled representations of $f \in \mathfrak{F}$:

$$\mathfrak{H} := \left\{h(f)(E) = \frac{\|e_{t_m}f(E)\|_g^2}{\sigma_{t_m}} : f \in \mathfrak{F}\right\}.$$

Naturally, this function class is equipped with a embedded $L_2$-metric $d'$ defined as

$$d'_{P_{nM}}(h, \tilde{h}) := \left(\frac{1}{nM}\sum_{m=1}^{M}\sum_{i=1}^{n}\left|h(t_m, E_{t_m}^{(i)}) - \tilde{h}(t_m, E_{t_m}^{(i)})\right|^2\right)^{1/2}, \quad h := \frac{\|e_{t_m}f\|_g^2}{\sigma_{t_m}}. \tag{84}$$

As the map $f \mapsto \|f\|^2$ is $2\bar{C}_f$-Lipschitz on the ball $\|f\| \leq \bar{C}_f$, and the scaling by $\sigma_{t_m}$ preserves the Lipschitz property up to $1/\sigma_{\min}$, where $\sigma_{\min} = \inf_m \sigma_{t_m} > 0$. Therefore,

$$\left|\frac{\|e_{t_m}f\|_g^2}{\sigma_{t_m}} - \frac{\|e_{t_m}\tilde{f}(E)\|_g^2}{\sigma_{t_m}}\right| \leq \frac{2\bar{C}_f}{\sigma_{\min}}\|f(E) - \tilde{f}(E)\|, \quad \bar{C}_f = \sup_{f \in \mathfrak{F}, E}\|f(E)\|.$$

The inequality comes from the fact that any orthonormal frames are isometry. Consequently, the covering number for $\mathfrak{H}$ can be bounded as

$$\mathcal{N}\left(\mathfrak{H}, d'_{P_{nM}}, \eta\right) \leq \mathcal{N}\left(\mathcal{F}, d_{P_{nM}}, \frac{\eta\sigma_{\min}}{2\bar{C}_f}\right).$$

Finally, applying the geometrically scaling terms, we obtain

$$\mathcal{N}\left(\mathfrak{G}, d'_{P_{nM}}, \eta\right) \leq \mathcal{N}\left(\mathfrak{H}, d'_{P_{nM}}, \eta\left(1 + \frac{4\kappa_{\max}^2}{9}\mathrm{Diam}^2(\mathcal{M})\right)^{-1}\right)$$

$$\leq \mathcal{N}\left(\mathfrak{F}, d_{P_{nM}}, \eta\sigma_{\min}\left(\bar{C}_f + \frac{8R\kappa_{\max}^2}{9}\mathrm{Diam}^2(\mathcal{M})\right)^{-1}\right),$$

Recall that standard Dudley's chaining inequality directly gives

$$\mathfrak{R}_{n,M} \leq \frac{12}{\sqrt{nM}}\int_0^{\mathrm{diam}(\mathcal{G})}\sqrt{\log\mathcal{N}\left(\mathcal{G}, d'_{P_{nM}}, \eta\right)}d\eta$$

$$\leq \frac{12}{\sqrt{nM}}\int_0^{\mathrm{diam}(\mathcal{G})}\sqrt{\log\mathcal{N}\left(\mathcal{G}, d_{P_{nM}}, \eta'(\eta)\right)}d\eta \tag{85}$$

where $\eta'(\eta) := \eta\sigma_{\min}/(2\bar{C}_f + (8/9)\bar{C}_f\kappa_{\max}^2\mathrm{Diam}^2(\mathcal{M}))$.

In order to obtain feasible entropy numbers, we construct the score network class $\mathcal{F}$ as follows. For each time index $m$, we consider a score model $s_\theta(t_m, \cdot) : \mathbb{R}^d \to \mathbb{R}^d$, implemented as a feed-forward RELU neural network of depth $L$ and hidden width at most $W$ at each layer. Specifically, the network takes the form

$$x \mapsto F_\theta(x) := W_L\sigma(W_{L-1}\sigma(\cdots\sigma(W_1 x)\cdots)),$$

where each weight matrix $W_\ell$ has shape $w_{\ell+1} \times w_\ell$ with $w_0 = d$, $w_L = d$, and $w_\ell \leq W$ for all $1 \leq \ell \leq L$. The activation function $\sigma(u) = \max\{0, u\}$ is applied coordinate-wise. To control the

function class complexity, we impose a path-norm constraint following Bartlett et al. (2017). The scaled path norm of the parameter collection $\theta$ is defined as

$$\|\theta\|_{\text{path}} := \left( \sum_{j_L \cdots j_0} \prod_{\ell=1}^{L} |W_\ell[j_\ell, j_{\ell-1}]|^2 \right)^{1/2},$$

and we assume this is uniformly bounded above by $B$ for all $\theta$ under consideration. The function class $\mathcal{F}$ itself is defined by embedding the neural network output into the target score difference, normalized by the time-dependent noise scale:

$$f_{\theta,m}(E) := \sqrt{\sigma_{t_m}}[s_\theta(t_m, E) - s(t_m, E)] = \sqrt{\sigma_{t_m}}[F_\theta(E) - \Sigma_{t_m}^{-1}E],$$

where $s_\star(t_m, E)$ is a reference score (for example, the Gaussian score function $\Sigma_{t_m}^{-1}E$). For each time index $m$, we define the associated function class $\mathfrak{F}_m$ as

$$\mathfrak{F}_m := \left\{ f_{\theta,m}(E) = \sqrt{\sigma_{t_m}}\big[F_\theta(E) - \Sigma_{t_m}^{-1}E\big] \big| \theta \in \Theta, \ \|\theta\|_{\text{path}} \leq B \right\},$$

The global function class $\mathfrak{F}$ over the time grid $\{t_1, \ldots, t_M\}$ is defined as the product class

$$\mathfrak{F} := \prod_{m=1}^{M} \mathfrak{F}_m = \{(f_1, \ldots, f_M) | f_m \in \mathfrak{F}_m\},$$

so that each element of $\mathfrak{F}$ consists of $M$-tuples of functions, one for each time index $m$. If all models share a common parameterization $\theta \in \Theta$ across time, one can equivalently write

$$\mathfrak{F} := \{(f_{\theta,1}, \ldots, f_{\theta,M}) | \theta \in \Theta, \|\theta\|_{\text{path}} \leq B\},$$

where $f_{\theta,m}(E)$ is defined as above for each $m$. By using the result of Bartlett et al. (2017), it directly gives

$$\log \mathcal{N}\left(\mathcal{F}_m, d_{P_n}, \eta\right) \leq \frac{\bar{D} \ln(2W^2)}{\eta'(\eta)^2} \left( \prod_{\ell=1}^{L} s_\ell^2 \rho_\ell^2 \right) \left( \sum_{\ell=1}^{L} \left( \frac{b_\ell}{s_\ell} \right)^{2/3} \right)^3 = \frac{C_{\text{NN}}}{\eta'(\eta)^2}, \qquad (86)$$

where $\bar{D}$ is the uniform bound on the data norm, $W$ is the maximum width of the hidden layers, $L$ is the network depth, $s_\ell$ is the spectral norm bound of the $\ell$-th layer, $\rho_\ell$ is the Lipschitz constant of the activation at the $\ell$-th layer, $b_\ell$ is the (2,1)-norm bound for the $\ell$-th layer, $D$ is the diameter of $\mathfrak{F}_m$, and $\eta_0$ is the minimal covering scale. Now, we improve the Dudley entropy number with

$$\frac{12}{\sqrt{nM}} \int_0^{\text{diam}(\mathcal{G})} \sqrt{\log \mathcal{N}\big(\mathcal{G}, d_{P_{nM}}, \eta'(\eta)\big)} d\eta \leq \frac{12}{\sqrt{n}} \int_0^{\text{diam}(\mathcal{G})} \max_m \sqrt{\log \mathcal{N}\big(\mathcal{G}, d_{P_m}, \eta'(\eta)\big)} d\eta \tag{87}$$

The Dudley entropy integral for a function class $\mathfrak{F}_m$ with respect to the empirical $L_2$ metric $d_{P_m}$ can be computed as

$$\mathfrak{R}_{n,M} \leq \int_0^D \sqrt{\frac{C_{\text{NN}}}{(\eta'(\eta))^2}} d\eta = \sqrt{C_{\text{NN}}} \cdot \frac{2\bar{C}_f + (8/9)\bar{C}_f \kappa_{\max}^2 \text{Diam}^2(\mathcal{M})}{\sigma_{\min}} \cdot \log \frac{D}{\eta_0} \tag{88}$$

where $D = \text{diam}(\mathfrak{F}_m)$ and $\eta_0$ is the minimal covering scale (typically, $\eta_0 = 1/n$). Expanding $C_{\text{NN}}$ with explicit network parameters, we have

$$\mathfrak{R}_{n,M} \leq \frac{12}{n^{1/4}} \left[ \bar{D}^2 \ln(2W^2) \left( \prod_{\ell=1}^{L} s_\ell^2 \rho_\ell^2 \right) \left( \sum_{\ell=1}^{L} \left( \frac{b_\ell}{s_\ell} \right)^{2/3} \right)^3 \right]^{1/2}$$

$$\cdot \frac{2\bar{C}_f + \frac{8}{9}\bar{C}_f \kappa_{\max}^2 \text{Diam}^2(\mathcal{M})}{\sigma_{\min}} \cdot \log \frac{D}{\eta_0}.$$

Collecting all the results together, we obtain the following high-probability uniform generalization bound. For all $\delta \in (0, 1)$, with probability at least $1 - \delta$, the following holds:

$$\sup_{\theta \in \Theta} \left| \widehat{\mathcal{R}}_n(\theta) - \mathcal{R}(\theta) \right|$$

$$\leq \frac{12}{n^{1/4}} \left[ \bar{D}^2 \ln(2W^2) \left( \prod_{\ell=1}^L s_\ell^2 \rho_\ell^2 \right) \left( \sum_{\ell=1}^L \left( \frac{b_\ell}{s_\ell} \right)^{2/3} \right)^3 \right]^{1/2} \frac{2\bar{C}_f + \frac{8}{9} \bar{C}_f \kappa_{\max}^2 \mathrm{Diam}^2(\mathcal{M})}{\sigma_{\min}} \log \frac{D}{\eta_0}$$

$$+ c \sqrt{\sigma_{t_m} \left( L_{t_m}^{\mathrm{tot}}(\theta) \right)^2 \lambda_{\max}(\Sigma_{t_m})} \sqrt{\frac{\log(1/\delta)}{nM}}.$$

Consequently, the upper bound for the worst-case horizontal KL divergence takes the following explicit form:

$$\sup_{\theta \in \Theta} \mathrm{KL}_{\mathrm{Hor}}(\tilde{\nu}_{[0,T]}^\theta | \tilde{\nu}_{[0,T]}) \leq \frac{1}{2} \left\{ \frac{12 C_{\mathrm{NN}} C_{\mathcal{M}}}{n^{1/4}} + \sqrt{\frac{C_{\mathrm{SubG}}^2 \log(1/\delta)}{nM}}. \right\}$$

with probability at least $1 - \delta$. Here, the three main terms $C_{\mathrm{NN}}, C_{\mathcal{M}}, C_{\mathrm{SubG}}$ reflect the contribution of network expressivity, manifold geometry, and stochastic noise, respectively. Their precise definitions are as follows:

$$C_{\mathrm{NN}}(L, W) := \left[ \bar{D}^2 \ln(2W^2) \left( \prod_{\ell=1}^L s_\ell^2 \rho_\ell^2 \right) \left( \sum_{\ell=1}^L \left( \frac{b_\ell}{s_\ell} \right)^{2/3} \right)^3 \right]^{1/2}, \tag{89}$$

$$C_{\mathcal{M}}(\sigma, \kappa, \mathrm{Diam}(\mathcal{M})) := \frac{2\bar{C}_f + \frac{8}{9} \bar{C}_f \kappa_{\max}^2 \mathrm{Diam}^2(\mathcal{M})}{\sigma_{\min}} \log \frac{D}{\eta_0}, \tag{90}$$

$$C_{\mathrm{SubG}}(\sigma_t, \Sigma) := c \sqrt{\sigma_{t_m} \left( L_{t_m}^{\mathrm{tot}}(\theta) \right)^2 \lambda_{\max}(\Sigma_{t_m})}. \tag{91}$$

$\square$

## E  Implementation Details

In this section, we provide comprehensive details on the implementation of our proposed horizontal diffusion model. We first specify the choice of hyperparameters used throughout the experiments. Subsequently, we formally define the construction of synthetic benchmark datasets, encompassing a suite of analytically parameterized surfaces that serve as controlled environments for evaluating geometric generative modeling. Finally, we describe the real-world datasets, including celestial surfaces and protein structures, modeled via spherical harmonics parameterizations and sourced from publicly available scientific repositories. All implementation specifics, including the design of neural architectures, training protocols, and evaluation metrics, are reported to ensure reproducibility and facilitate future extensions of our approach.

### E.1  Hyperparameters

Recall that the proposed (reverse) horizontal diffusion model for the Ornstein-Uhlenbeck (OU) bridge process on the orthonormal frame bundle is given by

$$dU_s = -\sum_a \left(2\gamma_s \epsilon^a + \sigma_s^2 v_s^a\right) H_a(U_s)ds + s^{\mathrm{Hor}}(s, U_s)ds + \sum_a \sigma_s H_a(U_s) \circ dW_s^a,$$

It depends on several time-dependent hyperparameters that critically determine both the geometry-adaptivity and empirical behavior of the model. Here, we detail the practical design and rationale for each hyperparameter as implemented in our framework.

**Damping Coefficient** $\gamma_s$**.**  The role of this coefficient is to govern the mean-reverting drift of the latent Euclidean process, corresponding to the dissipative force in the OU process. We use a constant value, $\gamma_s \equiv \gamma$, for all $s$, as this ensures numerical stability and avoids extreme drift behaviors.

**Diffusion Scale** $\Sigma_s$**,** $\sigma_s$**.**  The diffusion parameter $\sigma_s$ scales noise intensity of both forward and reverse OU process in the latent space. It is designed as $\sigma_s^2 = \gamma \Sigma_s$ so that we have $\gamma I = \sigma_s^2 \Sigma_s^{-1}$. The other paramter is exponentially controlled by using the formula $\Sigma_s = \Sigma_0 \exp(-\gamma s)$, with $\sigma_s$ computed at each time step.

**Bridge Intensity** $\lambda_s$**.**  The role of this coefficient is to control the strength of the attraction term $v_s$, driving the process toward the data manifold during reverse diffusion. In this paper, several scheduling options are considered as listed in following: (1) Exponential decay: $\lambda_s = \lambda_{\max} e^{-\alpha s}$, (2) Polynomial decay: $\lambda_s = \lambda_{\max}(1 - s/T)^k$, (3) Sigmoid decay: $\lambda_s = \lambda_{\max}/(1 + e^{\alpha(s-T/2)})$. In the experiement on sphere and torus, we used polynomial decay intensity function, and used exponential decay for other setups.

### E.2  Synthetic Dataset

**Enner Surface.** This first model space is a complete, orientable *minimal surface* in $\mathbb{R}^3$. It is most conveniently introduced through the polynomial immersion defined as follows:

$$f(u, v) = \left(u - \tfrac{u^3}{3} + uv^2, v - \tfrac{v^3}{3} + vu^2, u^2 - v^2\right), \qquad (u, v) \in \mathbb{R}^2.$$

In order to induce gradient fields, we apply differentiation $\mathbf{X}$ with respect to its parameters and yields the followings:

$$\partial_u f = \left(1 - u^2 + v^2, 2uv, 2u\right), \quad \partial_v f = \left(2uv, 1 - v^2 + u^2, -2v\right).$$

Taking inner products produces the first fundamental form

$$g_{11} = g_{22} = (1 + r^2)^2, \quad g_{12} = 0, \qquad r^2 = u^2 + v^2.$$

In other means, the induced metric $g = (1 + r^2)^2(du^2 + dv^2)$ is *conformal* to the Euclidean metric and we have $\det g = (1 + r^2)^4$. Taking derivatives on both component of Riemannian metric, we have

$$\partial_u g_{11} = 4u(1 + r^2), \quad \partial_v g_{11} = 4v(1 + r^2), \qquad \partial_u g_{22} = 4u(1 + r^2), \quad \partial_v g_{22} = 4v(1 + r^2).$$

Substituting these results into the Levi–Civita formula

$$\Gamma^i_{jk} = \frac{1}{2}g^{i\ell}\big(\partial_j g_{k\ell} + \partial_k g_{j\ell} - \partial_\ell g_{jk}\big)$$

gives the non–zero components of Christoffel symbols:

$$\Gamma^u{}_{uu} = \Gamma^v{}_{uv} = \Gamma^u{}_{uv} = \frac{2u}{1+r^2},$$

$$\Gamma^v{}_{vv} = \Gamma^u{}_{uv} = \Gamma^v{}_{vu} = \frac{2v}{1+r^2},$$

$$\Gamma^u{}_{vv} = -\frac{2u}{1+r^2}, \qquad \Gamma^v{}_{uu} = -\frac{2v}{1+r^2}.$$

**Catenoid Surface.** This second model space is another complete, orientable *minimal surface* embedded in $\mathbb{R}^3$. A convenient immersion of the catenoid can be explicitly given by the parameterization

$$f(u,v) = (\cosh v \cos u, , \cosh v \sin u, , v), \quad \text{for} \quad (u,v) \in [0, 2\pi) \times \mathbb{R}. \tag{92}$$

To analyze gradient fields and geometric properties, we compute the partial derivatives of the immersion $f$ with respect to the local parameters $u$ and $v$, obtaining $\partial_u f = (-\cosh v \sin u, , \cosh v \cos u, , 0)$ and $\partial_v f = (\sinh v \cos u, , \sinh v \sin u, , 1)$. Taking the inner products of these tangent vectors yields the components of the induced metric tensor, known as the first fundamental form. Explicitly, we have $g_{11} = \langle \partial_u f, \partial_u f \rangle = \cosh^2 v$, $g_{22} = \langle \partial_v f, \partial_v f \rangle = \cosh^2 v + \sinh^2 v = \cosh^2 v$, and $g_{12} = \langle \partial_u f, \partial_v f \rangle = 0$. Hence, the metric tensor is conformally equivalent to the Euclidean metric, and is concisely expressed as $g = \cosh^2 v, (du^2 + dv^2)$, with determinant $\det(g) = \cosh^4 v$.

The Christoffel symbols associated with the Levi-Civita connection can be computed from the metric derivatives. Since the metric depends solely on $v$, we have $\partial_u g_{ij} = 0$ for all $(i,j)$. The non-zero metric derivatives are $\partial_v g_{11} = \partial_v g_{22} = 2\cosh v \cdot 2 \sinh v = \sinh(2v)$. Using the Levi-Civita formula, $\Gamma^i_{jk} = \frac{1}{2}g^{i\ell}(\partial_j g_{k\ell} + \partial_k g_{j\ell} - \partial_\ell g_{jk})$, and the inverse metric components $g^{11} = g^{22} = \cosh^{-2} v$, we find the non-zero Christoffel symbols explicitly as follows: $\Gamma^u_{uv} = \Gamma^u_{vu} = \tanh v$, $\Gamma^v_{uu} = -\tanh v$, and $\Gamma^v_{vv} = \tanh v$. These Christoffel symbols reflect how tangent vectors are parallel transported across the curved geometry of the catenoid surface.

**Torus.** This space represents a classical example of a compact, orientable surface with genus one, smoothly embedded in $\mathbb{R}^3$. While the flat torus has been extensively studied in various contexts, generative modeling of the embedded, curved (non-flat) torus remains unexplored. In this study, we focus on this embedded torus as our primary model space, employing the following standard parameterization:

$$f(u,v) = ((R + r\cos v)\cos u, , (R + r\cos v)\sin u, , r\sin v), \quad \text{with} \quad (u,v) \in [0, 2\pi)^2, \tag{93}$$

where the constants $R > r > 0$ represent the major (central) and minor (tube) radii, respectively. To examine the geometry of this surface, we first compute the tangent vectors derived from the parameterization. The partial derivatives of the embedding are thus explicitly given by

$$\partial_u f = (-(R + r\cos v)\sin u, (R + r\cos v)\cos u, 0), \tag{94}$$
$$\partial_v f = (-r\sin v\cos u, -r\sin v\sin u, r\cos v). \tag{95}$$

Next, we determine the induced metric, or the first fundamental form, by taking inner products of these tangent vectors. The resulting metric tensor components become $g_{11} = \langle \partial_u f, \partial_u f \rangle = (R + r\cos v)^2$, $g_{22} = \langle \partial_v f, \partial_v f \rangle = r^2$, and $g_{12} = \langle \partial_u f, \partial_v f \rangle = 0$, yielding a diagonal metric tensor. Hence, the metric can be succinctly expressed as $g = (R + r\cos v)^2 du^2 + r^2 dv^2$, with determinant given by $\det(g) = r^2(R + r\cos v)^2$. To analyze curvature properties and parallel transport, we compute the derivatives of the metric tensor. Noting the metric's dependence solely on the parameter $v$, we find $\partial_u g_{ij} = 0$ for all $(i,j)$. Non-zero derivatives are explicitly computed as $\partial_v g_{11} = -2r(R + r\cos v)\sin v$ and $\partial_v g_{22} = 0$.

Using the inverse metric tensor components, $g^{11} = (R + r\cos v)^{-2}$ and $g^{22} = r^{-2}$, we apply the Levi-Civita connection formula, $\Gamma^i_{jk} = \frac{1}{2}g^{i\ell}(\partial_j g_{k\ell} + \partial_k g_{j\ell} - \partial_\ell g_{jk})$, to derive the non-zero

Christoffel symbols explicitly as follows:

$$\Gamma^u\_{uv} = \Gamma^u\_{vu} = -\frac{r \sin v}{R + r \cos v}, \tag{96}$$

$$\Gamma^v\_{uu} = \frac{(R + r \cos v) \sin v}{r}, \tag{97}$$

$$\Gamma^v\_{vv} = 0. \tag{98}$$

These Christoffel symbols characterize how vector fields evolve when parallel transported along curves on the torus surface.

**Dupin Cyclide.** The Dupin Cyclide is a classical example of a toroidal surface distinguished by its constant principal curvatures, and can be represented as a surface of revolution in $\mathbb{R}^3$. As with the standard torus, the Dupin torus provides a canonical model for studying nontrivial extrinsic geometry in generative modeling on curved surfaces. A standard parametrization is defined as follows:

$$f(u,v) = ((a + b \cos v) \cos u, (a + b \cos v) \sin u, b \sin v), \qquad (u,v) \in [0, 2\pi)^2,$$

where $a > b > 0$ represent the major and minor radii, respectively. This surface is a particular case of a Dupin cyclide. The tangent vectors are:

$$\partial_u f = (-(a + b \cos v) \sin u, (a + b \cos v) \cos u, 0),$$
$$\partial_v f = (-b \sin v \cos u, -b \sin v \sin u, b \cos v).$$

The first fundamental form has components:

$$g_{11} = \langle \partial_u f, \partial_u f \rangle = (a + b \cos v)^2, \qquad g_{22} = \langle \partial_v f, \partial_v f \rangle = b^2, \qquad g_{12} = \langle \partial_u f, \partial_v f \rangle = 0.$$

Thus, the induced Riemannian metric is diagonal and can be presented as follows:

$$g = (a + b \cos v)^2 du^2 + b^2 dv^2, \qquad \det g = (a + b \cos v)^2 b^2.$$

The derivatives of the metric components are:

$$\partial_v g_{11} = -2b(a + b \cos v) \sin v, \qquad \partial_v g_{22} = 0, \qquad \partial_u g_{ij} = 0.$$

The inverse metric is:

$$g^{11} = (a + b \cos v)^{-2}, \qquad g^{22} = b^{-2}.$$

Applying the Levi-Civita connection formula, the nonzero Christoffel symbols are:

$$\Gamma^u_{uv} = \Gamma^u_{vu} = -\frac{b \sin v}{a + b \cos v},$$

$$\Gamma^v_{uu} = \frac{(a + b \cos v) \sin v}{b},$$

$$\Gamma^v_{vv} = 0.$$

**Gaussian Mixture Model on Embedded Manifolds.** Let $\mathcal{M} = \Phi(U) \subset \mathbb{R}^3$ be a smooth two-dimensional embedded manifold parametrized by $\Phi : U \subset \mathbb{R}^2 \to \mathbb{R}^3$, with local coordinates $(u,v)$. A Gaussian mixture distribution defined on the intrinsic parameter domain $U$ has the form $p_U(u,v) = \sum_{k=1}^K \alpha_k \mathcal{N}((u,v); \mu_k, \Sigma_k)$, with $\alpha_k \geq 0$, $\sum_{k=1}^K \alpha_k = 1$, and $\Sigma_k \succ 0$, which induces a density on the manifold given by $p_\mathcal{M}(x) = p_U(u,v)/\sqrt{\det g(u,v)}|_{x=\Phi(u,v)}$, where $g(u,v) = J_\Phi(u,v)^\top J_\Phi(u,v)$ is the induced metric from the Jacobian $J_\Phi$ of the parametrization. Sampling from this manifold density involves drawing $(u,v) \sim p_U$ in parameter space and mapping to the manifold via $x = \Phi(u,v)$. Likelihood evaluation and inference tasks similarly involve pulling back points on the manifold to the parameter space coordinates $(u,v) = \Phi^{-1}(x)$. As an illustrative example, consider the catenoid parametrization $f(u,v) = (\cosh v \cos u, \cosh v \sin u, v)$ with $(u,v) \in [0, 2\pi) \times \mathbb{R}$. Its tangent vectors $f_u = (-\cosh v \sin u, \cosh v \cos u, 0)$ and $f_v = (\sinh v \cos u, \sinh v \sin u, 1)$ yield metric coefficients $E = \cosh^2 v$, $F = 0$, and $G = \cosh^2 v$, giving $\det g(u,v) = \cosh^4 v$. Thus, the induced density on the catenoid is explicitly $p_\mathcal{M}(x) = p_U(u,v)/\cosh^2 v|_{x=f(u,v)}$, clearly demonstrating how geometric distortion due to embedding is corrected by the induced metric.

### E.3 GEOMETRIC PROJECTION FOR EVALUATION OF EXISTING METHODS

Existing geometric generative methods cannot be directly applied to complex parametric surfaces such as Dupin cyclides or Enneper surfaces due to the lack of closed-form geodesics and tractable intrinsic geometric operators. To overcome this analytic intractability, we introduce three practically useful bijections aligning these non-canonical parametric surfaces with well-studied canonical manifolds ($e.g.$, $\mathbb{S}^n$, $\mathbb{T}^n$), which can readily support existing modeling techniques.

Our goal here is *not* to achieve isometry, but rather to establish a continuous, invertible mapping that enables transferring densities, trajectories, or fields between distinct geometric spaces. In all cases detailed below, parameters controlling scale and seam placement can be chosen for numerical convenience without affecting bijectivity. Through these geometric correspondences, standard baseline methods can first model data on canonical manifolds such as spheres or tori, after which results are projected back onto the original, more complex surfaces. By subsequently assessing the generated results in their native parametric spaces, we ensure that comparisons between methods remain geometrically meaningful, enabling fair evaluation while circumventing the complexities inherent to modeling directly on the original surfaces. We first enumerate basic notations, mappings that will be used in various conversion rules.

**Kelvin inversion on $\mathbb{R}^3$.** For a given point $x = (x_1, x_2, x_3)^\top \in \mathbb{R}^3$, center $c \in \mathbb{R}^3$, and radius $R > 0$, the *Kelvin inversion* $I_{c,R}$ is defined as

$$I_{c,R}(x) = c + \frac{R^2}{\|x - c\|^2}(x - c).$$

This mapping is a global involution on $\mathbb{R}^3 \setminus \{c\}$, satisfying $I_{c,R}^{-1} = I_{c,R}$.

**Standard torus parameterization.** Consider a standard torus in $\mathbb{R}^3$ with major radius $R_0 > 0$ and minor radius $0 < r < R_0$. The torus can be explicitly parameterized by angles $(\Theta, \Phi) \in [0, 2\pi) \times [0, 2\pi)$ as follows:

$$T(\Theta, \Phi) = \big((R_0 + r \cos \Phi) \cos \Theta, (R_0 + r \cos \Phi) \sin \Theta, r \sin \Phi\big).$$

Conversely, given a point $(x, y, z)^\top$ on this torus, its associated angular coordinates $(\Theta, \Phi)$ can be uniquely recovered through the inverse relations:

$$\Theta = \text{atan2}(y, x), \qquad \Phi = \text{atan2}\left(z, \sqrt{x^2 + y^2} - R_0\right).$$

**Stereographic projection and its inverse.** Let $N = (0, 0, 1)^\top$ be the north pole of the unit sphere $\mathbb{S}^2 \subset \mathbb{R}^3$. The stereographic projection from $\mathbb{S}^2 \setminus \{N\}$ onto the complex plane $\mathbb{C}$ is given by

$$s : \mathbb{S}^2 \setminus \{N\} \to \mathbb{C}, \quad s(y) = \frac{y_1 + iy_2}{1 - y_3}, \quad \text{for} \quad y = (y_1, y_2, y_3) \in \mathbb{S}^2 \setminus \{N\}.$$

Its inverse mapping is explicitly defined as

$$s^{-1} : \mathbb{C} \to \mathbb{S}^2 \setminus \{N\}, \quad s^{-1}(z) = \frac{1}{1 + |z|^2}\big(2 \operatorname{Re} z, 2 \operatorname{Im} z, |z|^2 - 1\big), \quad z \in \mathbb{C}.$$

This stereographic projection naturally extends to a bijection between the Riemann sphere $\mathbb{C} \cup \{\infty\}$ and the sphere $\mathbb{S}^2$ by identifying the north pole $N$ with the point at infinity:

$$s(N) = \infty, \quad s^{-1}(\infty) = N.$$

**Conversion between Catenoid and Sphere.** Given new notations and functions described above, let us consider the standard parametrization of the catenoid as described in Eq. 92. Then, we introduce an exponential compactification given by the mapping

$$\psi(u, v) = \alpha e^{v + iu} \in \mathbb{C} \cup \{\infty\}, \quad \text{with scale factor } \alpha > 0.$$

Leveraging this compactification, we construct the following explicit bijection from the cylindrical domain $[0, 2\pi) \times \mathbb{R}$ onto the sphere $\mathbb{S}^2$: $\Phi_{\text{cat} \to S^2}(u, v) = s^{-1}\big(\psi(u, v)\big)$, where $s : \mathbb{S}^2 \to \mathbb{C} \cup \{\infty\}$

denotes the stereographic projection. In particular, under this mapping, the cylindrical coordinates map naturally to spherical coordinates. Moreover, this transformation admits an explicit inverse: For any point $y \in \mathbb{S}^2$, one first computes its stereographic projection $z = s(y)$, after which the original catenoid coordinates are recovered as

$$u = \arg(z), \qquad v = \log(|z|/\alpha).$$

Consequently, any sampling procedures, density functions, or vector fields defined originally in the $(u, v)$-parametrization of the catenoid can be straightforwardly pulled forward onto $\mathbb{S}^2$ via $\Phi_{\mathrm{cat} \to S^2}$, and similarly pushed back onto the catenoid by means of the analytic inverse mapping presented above. This construction thus facilitates geometric transfer between these distinct spaces, enabling standard methods developed for canonical spherical geometries to be effectively adapted and evaluated within the analytically more complex catenoid setting.

**Conversion between Enneper Surface and Torus.** The classical Enneper surface admits a polynomial parameterization, which is known to possess intricate self-intersecting geometry, and hence is generally not globally injective. For practical applicability, it is beneficial to restrict our attention to a suitable, simply connected subdomain

$$D_R = \{(u, v) \in \mathbb{R}^2 : u^2 + v^2 < R^2\},$$

with the radius parameter $R > 0$ chosen sufficiently small to moderate, ensuring that the restriction $X_{\mathrm{enn}}|_{D_R}$ is injective over the support of interest. Such a selection of domain radius is essential to avoid self-intersections and ensure well-posedness of the ensuing geometric transformations. Next, we introduce polar coordinates within the domain $D_R$, defined explicitly by $r = \sqrt{u^2 + v^2}$ and $\theta = \operatorname{atan2}(v, u)$, providing a natural angular and radial parametrization of the Enneper surface domain. To facilitate the mapping onto the torus, we define a strictly increasing bijection $\eta : [0, \infty) \to [0, 1)$ through

$$\eta(r) = \frac{2}{\pi} \arctan(\beta r), \qquad \text{with a scaling parameter } \beta > 0,$$

which possesses the explicit inverse given by $r = \frac{1}{\beta} \tan\left(\frac{\pi}{2} \eta\right)$. This bijection is chosen specifically to achieve a smooth radial rescaling, mapping the infinite radial extent of the Enneper surface onto a finite interval suitable for embedding into the toroidal geometry. Furthermore, angles are mapped onto the standard toroidal coordinates by assigning $\Theta = \theta \bmod 2\pi$, thus ensuring periodic continuity in the angular direction, and $\Phi = 2\pi \eta(r) = 4 \arctan(\beta r)$, thereby embedding the radial direction smoothly onto the torus. Consequently, we define the explicit embedding from the Enneper surface domain to the torus as

$$\Phi_{\mathrm{enn} \to T^2}(u, v) = T(\Theta, \Phi),$$

where $T$ denotes the standard torus parametrization introduced previously. By construction, this mapping is continuous, smoothly varying, and injective when restricted to the chosen domain $D_R$, thereby establishing a well-defined bijection onto its toroidal image. Explicit recovery of the original coordinates from the toroidal embedding is straightforward: the inverse map explicitly yields $r = \frac{1}{\beta} \tan(\Phi/4)$, directly recovers the angular coordinate via $\theta = \Theta$, and subsequently produces the original planar coordinates by $(u, v) = (r \cos \theta, r \sin \theta)$. If required, the inverse chart of the original polynomial parametrization $X_{\mathrm{enn}}$ can then be applied, completing the full cycle of the geometric transformation between the Enneper surface and the torus.

**Conversion between Dupin Cyclide and Torus via Inversion.** A Dupin cyclide can be analytically represented as an inversion of a standard geometric surface such as a torus, cone, or cylinder, and thus admits a global angle-preserving bijection onto these simpler shapes through Kelvin inversion in $\mathbb{R}^3 \setminus \{c\}$. Given torus parameters $(R_0, r)$ along with an inversion defined by a center $c \in \mathbb{R}^3$ and radius $R > 0$, a Dupin cyclide can be explicitly obtained via the composition $C(\Theta, \Phi) = I_{c,R}\big(T(\Theta, \Phi)\big)$, where $T(\Theta, \Phi)$ is the standard parameterization of the torus. Conversely, given an arbitrary Dupin cyclide $C$, there exist suitable parameters $(c, R)$ that render the image $Y = I_{c,R}(C)$ into a standard torus configuration. Thus, the explicit transformation $\Phi_{\mathrm{dup} \to T^2}(x)$ defined by $(\Theta, \Phi) = \big(\operatorname{atan2}(Y_2, Y_1), \operatorname{atan2}(Y_3, \sqrt{Y_1^2 + Y_2^2} - R_0)\big)$, with $Y = I_{c,R}(x)$, provides a global bijection from the Dupin cyclide (excluding only the inversion center $c$) onto the standard torus, subject to identification along the torus seam. Its inverse is explicitly given by $x = I_{c,R}\big(T(\Theta, \Phi)\big)$, allowing full recovery of cyclide coordinates from toroidal parameters.

### E.4 REAL-WORLD DATASET

**Lunar Spherical Harmonic Dataset (LRO/LOLA SHADR).** To precisely describe the lunar surface geometry, we employ the spherical harmonic coefficient dataset released by NASA's Lunar Reconnaissance Orbiter (LRO) mission, specifically the Lunar Orbiter Laser Altimeter (LOLA) Spherical Harmonic Data Record (SHADR) Smith et al. (2010); Neumann et al. (2020). This dataset (*e.g.*, `lro_ltm05_2050_sha.tab`) represents the Moon's reference surface as a truncated expansion in real spherical harmonics:

$$r(\theta, \varphi) = r_0(\theta, \varphi) + \sum_{l=1}^{L_{\max}} \sum_{m=0}^{l} \left[ C_{l,m} \cos(m\varphi) + S_{l,m} \sin(m\varphi) \right] P_{l,m}(\cos\theta), \tag{99}$$

where $r_0$ is a reference radius, $P_{l,m}$ are the associated Legendre polynomials, and $(C_{l,m}, S_{l,m})$ are the tabulated spherical harmonic coefficients up to degree and order $L_{\max} = 2050$. This high-resolution model enables a faithful reconstruction of the lunar topography, capturing both global structure and fine-scale features. The dataset is distributed via NASA's Planetary Data System (PDS) and constitutes the standard scientific basis for lunar surface analysis and generative modeling. Empirically, we found that retaining only the leading 50 harmonic coefficients suffices to capture the essential topographic features for generative modeling. Accordingly, our experiments utilize coefficients up to degree $L_{\max} = 50$. To train the diffusion model, we sample $2^{10}$ points from the spherical harmonic parameterization of the lunar surface and employ these as data instances $x_0 \sim \rho_0^{\mathcal{M}}$ to define the score matching loss.

**Asteroid Spherical Harmonic Dataset (NEAR/Eros).** In second real-world dataset, we reconstruct and analyze the shape of asteroid 433 Eros by utilizing the spherical harmonic coefficient dataset acquired by the NEAR Shoemaker mission Miller et al. (2002). The dataset provides a tabulated set of real spherical harmonic coefficients $(C_{l,m})$ up to a specified maximum degree $L_{\max}$, as determined from laser altimeter and optical data. The surface is parameterized in the form:

$$r(\theta, \varphi) = r_0(\theta, \varphi) + \sum_{l=0}^{L_{\max}} \sum_{m=-l}^{l} C_{l,m} Y_{l,m}(\theta, \varphi), \tag{100}$$

where $Y_{l,m}$ denotes the (complex or real-valued) spherical harmonics. Each row in the dataset specifies $(l, m, C_{l,m})$, which are filtered by thresholding or truncation in downstream processing. This high-resolution expansion enables both scientific study and generative modeling of the asteroid's three-dimensional surface geometry. The dataset is publicly available as part of the NEAR A Shoemaker mission archive distributed by NASA's Planetary Data System (PDS) NASA PDS Small Bodies Node (2001). To maintain consistency with our lunar surface experiments, we adopt an analogous experimental setup for the Eros dataset. Specifically, we retain the leading 50 spherical harmonic coefficients to parameterize the asteroid's surface and uniformly sample $2^{10}$ points for use in training the diffusion model via the score matching loss.

**Shape Analysis of Human Anatomy Surfaces.** To demonstrate the applicability of our framework to biomedical shape analysis, we consider publicly available anatomical datasets distributed with the SPHARM-PDM software (Styner et al., 2006; Shen et al., 2009). Specifically, we adopt (i) the *Hippocampi dataset*, which consists of left and right hippocampus surface models from multiple subjects divided into two clinical groups, and (ii) the *Knee dataset*, which provides volumetric MRI data of the knee joint along with preprocessed surface reconstructions. Both datasets are parameterized onto the sphere using spherical harmonics, enabling direct comparison of 3D morphologies across subjects. These anatomical surfaces serve as standard benchmarks in neuroimaging and musculoskeletal research, providing a controlled experimental setting for validating geometric learning algorithms.

**Quantum Tomography.** To prepare the quantum dataset, we leverage the publicly available QDataSet Perrier et al. (2022), which provides experimentally realistic quantum-control simulations. Unlike classical shape datasets such as the Hippocampi and Knee surfaces that are preprocessed with spherical harmonics, the raw QDataSet does not directly provide Bloch vectors but rather 18-dimensional expectation values of an informationally complete operator set. To obtain usable qubit states on the manifold $\mathbb{CP}^1$, we perform a linear inversion tomography step: given measurement operators $\{V_k\}_{k=1}^{18}$ and observed expectations $E_k = \mathrm{Tr}(\rho V_k)$, we solve an overdetermined least-squares system to estimate the Bloch vector $r = (r_x, r_y, r_z)$ while enforcing the

| Block | #Linear Layers | Input → Output Dim | Notes |
|---|---|---|---|
| Input concat | – | $[E \in \mathbb{R}^d,\ \Delta x \in \mathbb{R}^d,\ \text{vec}(e^{-1}) \in \mathbb{R}^{d^2}]$ | Time embedding $\tau(t) \in \mathbb{R}^T$ injected via FiLM |
| Encoder stage $s$ | $1 + 3R$ | $C_{s-1} \to C_s$ | in-projection (1) + $R$ ResBlocks (fc1+fc2+FiLM) |
| Bottleneck | $2 \times 3$ | $C_L \to C_L$ | two ResBlocks, each with (fc1+fc2+FiLM) |
| Decoder stage $s$ | $1 + 3R$ | $(C_{s+1} \oplus C_{s+1}) \to C_s$ | merge-projection (1) + $R$ ResBlocks |
| Score head | 1 | $C_1 \to d$ | LN + Act + Linear |
| Bridge head | 1 | $C_1 \to d$ | LN + Act + Linear |
| Per-ResBlock | 3 | $C \to C$ | fc1, fc2, and FiLM$(T \to 2C)$ |

**Table 5: Layer composition of ScoreUNet.** Each encoder stage projects the features to a higher channel dimension, the bottleneck applies two residual transformations, and the decoder fuses skip connections back to lower channels. Both heads output $d$-dimensional vectors for the score and bridge predictions.

physical constraint $|r| \leq 1$. The reconstructed Bloch vectors are then mapped to density matrices $\rho = \frac{1}{2}\left(I + r_x\sigma_x + r_y\sigma_y + r_z\sigma_z\right)$, yielding a dataset supported on $\mathbb{CP}^1 \cong \mathbb{S}^2$.

### E.5   Neural Network Architecture

Our framework employs a UNet-style multilayer perceptron designed for structured vector inputs. The input feature is the concatenation of the latent state $E \in \mathbb{R}^d$, the displacement $\Delta x \in \mathbb{R}^d$, and the flattened inverse frame matrix $\text{vec}(e^{-1}) \in \mathbb{R}^{d^2}$, while the time variable $t$ is encoded by a sinusoidal embedding $\tau(t) \in \mathbb{R}^T$ and injected throughout the network via FiLM conditioning Perez et al. (2018). The encoder consists of successive stages, each performing a linear in-projection followed by $R$ FiLM-modulated residual blocks, thereby projecting features to higher channel dimensions and storing intermediate outputs as skip connections.

At the bottleneck, two residual transformations refine the representation with layer normalization, activation, and FiLM-modulated affine conditioning. The decoder mirrors the encoder by concatenating the current hidden representation with the corresponding skip features, applying a merge projection, and passing the result through $R$ residual blocks, progressively reducing the channel dimension back to the base width. Finally, two parallel output heads map the decoded representation to task-specific predictions: the *score head* outputs $d$-dimensional vectors corresponding to the learned score function $s(t, E)$, and the *bridge head* outputs $d$-dimensional vectors modeling the bridge function $v(t, \Delta x, e^{-1})$. This architecture preserves the UNet property of combining hierarchical feature extraction with precise skip connections, while FiLM conditioning ensures that temporal information influences all layers of the computation.

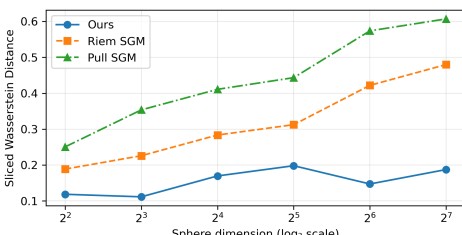

**Figure 4: Robustness to Dimensionality on High-Dimensional Spheres.** Performance of different methods as the dimension of the sphere increases. Unlike Riemannian SGM and Pull NF, our HDM maintains stable and accurate density estimation even in high-dimensional settings by leveraging Euclidean-based modeling.

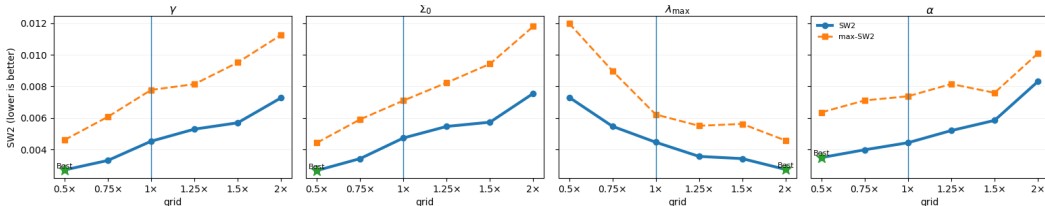

**Figure 5: Robustness Analysis on High-Dimensional Spheres.** Each panel illustrates sensitivity analysis by systematically varying a single parameter ($\gamma$, $\Sigma_0$, $\lambda_{\max}$, $\alpha$) through multiplicative perturbations around the baseline (vertical blue line at $1\times$), while keeping all other parameters constant. The performance metrics are reported as the sliced Wasserstein-2 distance and the max-sliced Wasserstein-2 distance between the target and generated samples.

### E.6 ABLATION STUDY

**Sensitivity Analysis on Toy Example.** We systematically investigate how increasing the intrinsic dimension of the sphere $\mathbb{S}^n$ affects density estimation performance. Unlike prior methods such as Riemannian SGM and Pull NF, HDM leverages the efficiency of Euclidean neural networks for modeling, resulting in substantially more robust and stable reconstruction as dimensionality grows. In particular, experiments on spheres with increasing dimension demonstrate that, while existing methods exhibit significant performance degradation, HDM maintains consistently high accuracy by capitalizing on the advantage of Euclidean-based modeling.

**Sensitivity Analysis on Hyperparameters.** We perform a comprehensive sensitivity analysis on key hyperparameters to evaluate their impact on model performance in Figure 5. Specifically, we systematically vary each of the parameters $(\gamma, \Sigma_0, \lambda_{\max}, \alpha)$ introduced in Sec E.1 individually, while holding the remaining parameters fixed at their baseline values. Empirically, the analysis demonstrates that smaller values of $\gamma$, $\Sigma_0$, and $\alpha$ lead to improved alignment between the generated and target distributions, whereas larger values of $\lambda_{\max}$ enhance performance within the tested range. The experimental setup for this analysis is standardized across all evaluations, using total integration time $T = 1$, $N = 600$ integration steps, 4096 particles, an exponential schedule for the diffusion parameter $\lambda_s$, 50 random projections, and Wasserstein exponent $p = 2$.

### E.7 ADDITIONAL EXPERIMENTAL RESULTS

**Enner Surface.** In this experiment, we consider the target distribution on the Enneper surface to be a mixture of three Gaussian components, each localized around different regions of the parameter space. Our horizontal diffusion model successfully recovers this multimodal structure: as the reverse process evolves (from right to left in Figure 6), the generated samples concentrate into three distinct clusters that match the modes of the target distribution. This demonstrates the ability of our method to accurately capture and reconstruct complex, multimodal densities defined on nontrivial manifolds.

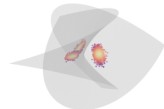 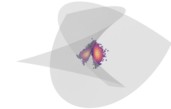 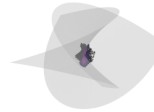 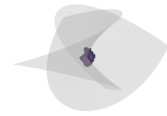

**Figure 6: Visualization of Reverse process on the Enneper surface**. The reverse process sampled from proposed method evolves from right to left, showing the density transformation as the system proceeds backward in time.

### E.8 ALGORITHM

This section presents the core algorithms for sampling and simulating horizontal diffusions on the frame bundle. We introduce methods for stationary distribution sampling, orthonormal frame construction, horizontal lift evaluation, and reverse-time diffusion simulation.

**Sampling Stationary Distribution on** $O(\mathcal{M})$**.** we first outline Algorithm 1, which samples independent points $(x, e) \in \mathcal{O}(\mathcal{M})$ from the stationary distribution given by the product measure $d\mathrm{Vol}_g \otimes d\mathrm{Haar}_{O(d)}$. The algorithm first draws a base point $x$ according to the Riemannian volume measure, followed by generating an orthonormal frame $e$ at $x$ via QR factorization of a Gaussian matrix. Thus, it ensures Haar-distributed randomness in each fiber. As the base and fiber components are sampled independently, this procedure yields unbiased samples from the stationary law of the horizontal diffusion on the frame bundle.

---

**Algorithm 1** SAMPLING STATIONARY MEASURE $d\rho_\infty^{\mathrm{Hor}} = d\mathrm{Vol}_g \otimes d\mathrm{Haar}_{O(d)}$

---

**Require:** Riemannian manifold $(\mathcal{M}, g)$. REJECTIONVOLUMESAMPLE() that returns $x \sim d\mathrm{Vol}_g$;
**Ensure:** $(x, e) \in \mathcal{O}(\mathcal{M})$ sampled from the product measure.
    **Sampling the base point**
1: $x \leftarrow$ REJECTIONVOLUMESAMPLE()
    **Sampling the orthonormal frame at** $x$
2: $Z \leftarrow \mathrm{randn}(d, d)$                                              ▷ entries i.i.d. $\mathcal{N}(0, 1)$
3: $(Q, R) \leftarrow \mathrm{QR}(Z)$                                         ▷ column-orthonormal $Q$
4: $Q \leftarrow Q\mathrm{diag}\big(\mathrm{sign}(\mathrm{diag}(R))\big)$                ▷ enforce $Q \in O(d)$ (Haar)
5: $e \leftarrow$ FRAMEFROMMATRIX$(x, Q)$           ▷ convert $Q$ to orthonormal frame in $T_x\mathcal{M}$
    **Return**
6: **return** $(x, e)$

---

Given a point $x \in \mathcal{M}$ and a Haar-random orthogonal matrix $Q \in O(d)$, Algorithm 2 constructs an orthonormal frame in $T_x\mathcal{M}$ by first orthonormalising the local coordinate basis via a metric-aware Gram–Schmidt process, and subsequently applying the columns of $Q$ as coefficients to form the final frame. This yields an isometric linear map $e : \mathbb{R}^d \to T_x\mathcal{M}$, whose orientation is distributed according to the Haar measure on the fibre over $x$.

---

**Algorithm 2** FRAMEFROMMATRIX$(x, Q)$

---

**Require:** Point $x \in \mathcal{M}$, Haar matrix $Q \in O(d)$
**Ensure:** Orthonormal frame $e = \big(e_1, \ldots, e_d\big)$ at $x$

1: $\big\{b_i = \frac{\partial}{\partial x^i}\big|_x\big\}_{i=1}^d \leftarrow$ LOCALBASIS$(x)$              ▷ chart differentials
2: $\big\{\hat{e}_i\big\}_{i=1}^d \leftarrow$ GRAMSCHMIDT$(\{b_i\}, g_x)$         ▷ produces $g$-orthonormal vectors
3: **for** $a = 1$ **to** $d$ **do**
4:     $e_a = \displaystyle\sum_{i=1}^d \hat{e}_i Q_{ia}$                             ▷ Embed Haar matrix
5: **end for**
6: **return** $e = (e_1, \ldots, e_d)$                         ▷ linear map $e : \mathbb{R}^d \to T_x\mathcal{M}$

---

In Algorithm 3, a simple rejection sampling scheme is used to generate points in local coordinates $y \in B$ with respect to the intrinsic Riemannian volume. Proposals are drawn uniformly from the coordinate patch, and accepted with probability proportional to the volume density $\mathrm{VolDens}(y)$, normalized by an upper bound $M$. This procedure yields samples exactly distributed according to the manifold's volume element, serving as an accessible alternative to more complex MCMC methods when no explicit inverse map is available.

---

**Algorithm 3** REJECTIONVOLUMESAMPLE()

---

**Require: Local Coordinate domain** $B \subset \mathcal{M}$, **Determinant oracle** $\mathrm{VolDens}(y) \equiv \sqrt{\det g_{ij}(y)}$
for $y \in B$, Constant $M \geq \sup_{y \in B} \mathrm{VolDens}(y)$

**Ensure:** Local coordinate vector $y \in B$ distributed according to the intrinsic volume measure $\mathrm{VolDens}(y)dy$

1: **repeat**
2:    $y \leftarrow \mathrm{Unif}(B)$                                      ▷ draw uniformly in coordinate region
3:    $p \leftarrow \mathrm{VolDens}(y)/M$                                ▷ $0 < p \leq 1$
4:    $u \leftarrow \mathrm{Unif}(0,1)$
5: **until** $u < p$                                                      ▷ accept with probability $p$
6: **return** $y^{\star} \leftarrow y$

---

**Sampling Reverse Horizontal Diffusion** $\mathbf{U}_s$. Algorithm 4 computes the horizontal lift matrix $H$ at a given frame bundle state $U = (x, e)$, where $x \in \mathcal{M}$ and $e \in O(d)$. The resulting $H \in \mathbb{R}^{(d+d^2) \times d}$ encodes the horizontal distribution associated with the Levi-Civita connection at $x$. Each column $H_a$ concatenates the base-point direction $e_{\cdot a}$ and the frame evolution $-\Gamma(e_{\cdot a})e$, where $\Gamma$ denotes the Christoffel symbol tensor at $x$. The upper $d$ rows of $H_a$ yield the directional derivative along $e_{\cdot a}$ in $T_x \mathcal{M}$, while the lower $d^2$ rows encode the corresponding infinitesimal frame rotation induced by parallel transport. This construction provides a local trivialization of the horizontal distribution, ensuring compatibility with the manifold's Riemannian geometry and connection.

---

**Algorithm 4** HORIZONTALLIFT($U{=}(x,e)$)

---

**Require:** State $U = (x, e)$ with *base point* $x \in \mathcal{M}$ and *frame matrix* $e = [e_{ia}] \in O(d)$
**Ensure:** Horizontal lift matrix $H = \begin{bmatrix} H_1 \mid \cdots \mid H_d \end{bmatrix}$ such that $H_a = \begin{pmatrix} e_{\cdot a}, -\Gamma(e_{\cdot a})e \end{pmatrix}$

1: $\Gamma \leftarrow \mathrm{CHRISTOFFEL}(x)$                            ▷ $\Gamma^l_{jk}$ tensor of size $d \times d \times d$
2: $H \leftarrow \mathbf{0}_{(d+d^2) \times d}$
3: **for** $a = 1$ **to** $d$ **do**
        *// base–point component*
4:    **for** $i = 1$ **to** $d$ **do**
5:        $H_{i,a} \leftarrow e_{ia}$                                    ▷ $(H_a)^x = e_{\cdot a}$
6:    **end for**
        *// vertical (frame) component*
7:    **for** $l = 1$ **to** $d$ **do**
8:        **for** $m = 1$ **to** $d$ **do**
9:            $s \leftarrow 0$
10:           **for** $j = 1$ **to** $d$ **do**
11:               **for** $k = 1$ **to** $d$ **do**
12:                   $s \leftarrow s + \Gamma^l_{jk} e_{ja} e_{km}$
13:               **end for**
14:           **end for**
15:           $r \leftarrow d + (l-1)d + m$                             ▷ row index for $(l, m)$ pair
16:           $H_{r,a} \leftarrow -s$
17:       **end for**
18:   **end for**
19: **end for**
20: **return** $H$

---

With the definition of horizontal lift of Euclidean latent vectors, Algorithm 5 describes a discretized scheme for sampling from the reverse-time dynamics of the horizontal diffusion process on the frame bundle $\mathcal{O}(\mathcal{M})$. The procedure reconstructs approximate samples from the initial data distribution $\rho_0^{\mathrm{Hor}}$ by integrating the reverse Stratonovich SDE backward from a given terminal state.

Given a terminal state $(U_T, E_T)$, the algorithm first precomputes the reverse-time grid and associated scalar schedules, including stepwise values for the bridge parameter $\lambda$, the drift $\gamma$, and the noise strength $\sigma^2$. Initialization is performed by setting the latent and geometric states to the prescribed terminal values.

---

**Algorithm 5** REVERSEHORIZONTALDIFFUSION$(U_T, E_T; T, N)$

---

**Require:** Terminal state $U_T = (x_T, e_T) \in \mathcal{O}(\mathcal{M})$, Terminal latent $E_T \in \mathbb{R}^d$, Time horizon $T > 0$,
Number of steps $N$ with step size $\Delta s = T/N$.
**Ensure:** Approximate sample $U_0$ from the data distribution $\rho_0^{\mathrm{Hor}}$
    **Pre–compute time grids and scalar schedules**
1: **for** $i \leftarrow 0$ **to** $N$ **do**
2:     $s_i \leftarrow i\Delta s, \quad \lambda_i \leftarrow \lambda(s_i), \gamma_i \leftarrow \gamma(s_i), \sigma_i^2 \leftarrow \sigma^2(s_i) = \gamma_i\Sigma(s_i)$
3: **end for**

    **Initialize reverse variables**
4: $U \leftarrow U_T, \quad E \leftarrow E_T$

5: **for** $i \leftarrow N-1$ **down to** $0$ **do**
6:     $s \leftarrow s_i, \quad \Delta s \leftarrow s_{i+1} - s_i$                         $\triangleright$ reverse time increment
    //geometry: bridge vector in frame coordinates
7:     $x \leftarrow$ INITIALPOINT$(U)$, $e \leftarrow$ INITIALFRAME$(U)$
8:     $v \leftarrow v(t, \theta)$
    //scores and Stratonovich drift
9:     $b_{\mathrm{rev}} \leftarrow -\lambda_i v + \gamma_i E + \sigma_i^2 s_\theta(E, s)$
10:    $\xi \leftarrow \sigma_i\sqrt{\Delta s} * \mathrm{randn}(d)$                        $\triangleright \xi \sim \mathcal{N}(0, \sigma_i^2\Delta sI_d)$
11:    $dE \leftarrow b_{\mathrm{rev}}\Delta s + \xi$
12:    $E \leftarrow E + dE$
    //horizontal Stratonovich update of the frame path
13:    $H \leftarrow$ HORIZONTALLIFT$(U)$                   $\triangleright H \in \mathbb{R}^{(d+d^2)\times d}$
14:    $U \leftarrow U + HdE$            $\triangleright$ Stratonovich step $dU = H(U) \circ dE$
15:    **if** $i \equiv 0 \pmod{P}$ **then**          $\triangleright$ every $P$ steps re–orthonormalise
16:        $e \leftarrow$ ORTHONORMALIZEMETRIC$(e)$             $\triangleright e^T g(x)e = I$
17:        $U \leftarrow (x, e)$
18:    **end if**
19: **end for**
20: **return** $U_0 \leftarrow U$

---

The main loop then iteratively integrates the reverse SDE from time $T$ to $0$ in $N$ discrete steps. At each iteration, the current frame and base point are extracted, and the neural networks infer the bridge vector $v = v(t, \theta)$, quantifying displacement relative to the anchor. The reverse drift combines three contributions: a deterministic pull towards $x_0$, a restoring force in the latent, and a score-driven correction based on the model's learned score SCORE$(E, s)$. Stochastic increments $\xi$ simulate Gaussian noise compatible with the instantaneous covariance.

The latent vector $E$ is updated via an Euler–Maruyama step, and the corresponding geometric state $U$ is propagated horizontally in the frame bundle according to the Stratonovich rule $dU = H(U) \circ dE$. To ensure numerical stability, we prform periodic re-orthonormalization of the frame using the manifold metric.

Upon completion, the algorithm returns the initial geometric state $U_0$, representing a sample drawn approximately from the reverse-time law of the horizontal diffusion process. This scheme thereby enables efficient simulation-based generation of samples from complex data distributions defined over the manifold and its frame bundle.

