# OpenReview forum: "Horizontal Diffusion Models: Riemannian Score-based Generative Modeling via Frame-Connection Geometry"
_ICLR.cc/2026/Conference — ICLR 2026 Conference Withdrawn Submission_

### Official Review · Reviewer_7Ecm · 2025-10-29

**Soundness:** 3
**Presentation:** 3
**Contribution:** 4
**Rating:** 6
**Confidence:** 3

**Summary:**

Previous geometric generative models typically rely on explicit geometric constructions such as geodesic distances or logarithmic maps, which are not applicable to general smooth manifolds where explicit geodesics are either unavailable or impractical to compute.
To address this, this paper proposes a diffusion model on general Riemannian manifolds using intrinsic geometry instead of explicit geodesic computations. As the Euclidean stochastic processes can be lifted into tangent spaces, the paper introduces horizontal diffusion processes that directly operate on the orthonormal frame bundle and bypass the need for geometric operations such as geodesic distance computation. Experimental results on synthetic parametric surfaces, real-world quantum qubit datasets, and celestial bodies show that the previous diffusion models relying on explicit geometry achieve limited performance, while the proposed method demonstrates superior performance.

**Strengths:**

- To the best of my knowledge, this work is the first to explore generative modeling on manifolds using intrinsic geometry. The paper identifies the limitations of previous works relying on extrinsic geometry, e.g., geodesics, and addresses this by using an intrinsic geometrical approach.

- The idea of operating directly on the orthonormal frame bundle and using horizontal lift is elegant and novel. By lifting Euclidean SDEs to the frame bundle, it can preserve geometry while overcoming the scalability issue.

- Using the horizontal parameterization trick to address the practical implementation of horizontal KL divergence for horizontal score matching is a significant contribution in this field that bridges theory and real-world application. By expressing horizontal quantities in terms of the latent Euclidean process, it bypasses the need for explicit parallel transport or projection operations.

- Although the paper relies heavily on advanced mathematics, it provides a background, for example, explaining concepts like orthonormal frame bundle, connection, and horizontal lift.

- While I couldn't fully validate that the proofs are entirely correct, the logic seems sound and rigorous.

**Weaknesses:**

- Gauge equivariance appears to be an important property used throughout the paper, including the parameterization trick and the training of the neural network. I cannot find the explanation for it, and why it is important. This should have been discussed in section 2.

- I don't understand the importance of the claim in lines 260-261, "allowing the use of standard Euclidean score networks without architectural changes". Although previous Riemannian diffusion models rely on geodesics, they still use the MLP architecture for their score network. Why is this claim important? I think architecture is not the key to learning geometry, but rather lies in the geometrical design of the diffusion process.

- How is the horizontal dynamics (as in Eq. (15)) simulated, especially the Stratonovich representation? How are the intermediate samples $
E_t\sim\rho_t$ required for computing the loss sampled? Details for the actual implementation are insufficient. Pseudo code on how the training and sampling are happening would greatly help in understanding the implementation.

- Large performance gap between previous diffusion models and the proposed model on simple manifolds, such as sphere or torus is not clear. Although Riemannian SGM or Riemannian flow matching rely on geodesics, geodesics have a closed-form in these manifolds. I don't see why these models underperform the horizontal approach.

- For more complex manifolds, such as Catenoid, Enner, or Dupin, why do the previous models and the proposed methods all show larger gaps compared to simpler manifolds like sphere or torus? I guess the computation of geodesics could affect the previous models, but why does it also happen for the intrinsic approach?

- While the paper strongly claims the scalability of lifting the Euclidean process, it lacks empirical results. Experiments on high-dimensional tori, as done in RSGM or Riemannian flow matching, would explicitly show the scalability to higher dimensions. Since the implementation-wise details are also insufficient, the scalability should be explicitly shown.

- Since the proposed method is based on a complex formulation with a new model design, I'm curious to see the time complexity comparison with previous models. I assume the intrinsic approach would be slower than the extrinsic approaches on simple manifolds like sphere or torus, but could be faster on more complex manifolds.

- (Minor) Visualization of the horizontal lift and the horizontal diffusion process would help better understand the elegant idea and the framework.

**Questions:**

Please address the questions raised in the weakness section.

---

### Official Review · Reviewer_bmvr · 2025-11-01

**Soundness:** 2
**Presentation:** 2
**Contribution:** 2
**Rating:** 2
**Confidence:** 2

**Summary:**

This paper introduces Horizontal Diffusion Models, a framework for generative modeling on Riemannian manifolds that works on the orthonormal frame bundle and enforces horizontality through the Levi Civita connection. The central technical move is to lift a Euclidean score network into a horizontal vector field and to show that a geometry aware Kullback Leibler on the bundle reduces to a Euclidean L two score matching loss. Training then uses a lifted Ornstein Uhlenbeck bridge, and under design choices for the drift and noise the stationary law approaches the Sasaki volume, see equation 14. Empirically, the paper reports density modeling on parametric surfaces and on real surfaces parameterized by spherical harmonics, with evaluation based on sliced Wasserstein and maximal sliced Wasserstein distances.

**Strengths:**

### **Strengths**

- Conceptual simplicity with a potentially wide reach. Lifting Euclidean scores to the frame bundle to obtain geometry consistent flows avoids geodesic computations and give rise to manifold specific architectures, and the built in gauge equivariance is appealing, see Proposition 2 point 4 and Corollary 2 point 5.
- A concrete training recipe. Proposition 2 point 3 and equation 11 make the objective look exactly like standard Euclidean score matching, which could make adoption straightforward for practitioners. Algorithmic details for sampling and orthonormalization are clear and useful.
- Initial evidence on several geometries. The paper compares against Riemannian score models and continuous flow methods on the torus, catenoid, Enneper, Dupin cyclide, quantum qubit states, and real planetary or anatomical surfaces, with consistently lower sliced Wasserstein numbers in the tables on pages viii and ix.

**Weaknesses:**

### **Weaknesses**

- The empirical evaluation is too narrow for ICLR standards. Results rely mainly on sliced Wasserstein and maximal sliced Wasserstein metrics on a small set of manifolds, with several baselines requiring projection of complex surfaces to canonical ones, which may bias comparisons. The paper would be much stronger with broader and more intrinsic evaluations, specifically to more truly real-world data.
- The theory is presented in a highly abstract style, which makes it difficult for non-experts to follow. A concise intuitive explanation of what the frame bundle contributes, how horizontality works in practice, and why the training objective becomes Euclidean before formal propositions would significantly improve clarity (see Proposition 2 points 2–3 and Theorem 2 point 6). It would make the work not only more accessible, but it would also be easier to point out strength of the work.
- The approach depends on geometric quantities like Christoffel symbols and the second order expansion of the logarithmic map (equation 16), yet computational cost, stability, and approximation error are not systematically analyzed. Ablations showing how these approximations and curvature effects influence performance would help assess robustness and scalability.

**Questions:**

### **Questions**

- What are the most promising real-world applications for Horizontal Diffusion Models, and could you demonstrate one beyond synthetic manifolds, for instance in molecular or geometric data?
- What assumptions make the reduction of the horizontal Kullback Leibler to the Euclidean L two loss in Equation 11 valid, and is this equality stable under discretization?
- Small things I was wondering about: Under what conditions on curvature and noise schedules does the stationary law in Equation 14 exist and remain unique? A brief proof or reference would clarify this. Also, how significant is the error from the second order logarithmic expansion in Equation 16, and would higher order or numerical maps improve stability on highly curved manifolds?

---

### Official Review · Reviewer_yvDV · 2025-11-08

**Soundness:** 3
**Presentation:** 2
**Contribution:** 3
**Rating:** 6
**Confidence:** 3

**Summary:**

This paper presents Horizontal Diffusion Models (HDM), a new framework for score-based generative modeling on Riemannian manifolds built on orthonormal frame bundles. The main idea is to avoid costly geometric operations such as geodesics and exponential maps by lifting Euclidean diffusion processes to the frame bundle using horizontal connections. The method achieves strong empirical results on complex surfaces while remaining theoretically well-grounded.

**Strengths:**

- The work proposes a novel, interesting, and mathematically sound solution to some critical limitations of geometric generative models.
- The mathematical framework is well-developed, supporting the method rigorously. The generalization bound provides valuable theoretical guarantees.
- The paper provides extensive algorithmic details and hyperparameter specifications, which are useful for reproducibility.

**Weaknesses:**

While the introduction effectively presents the previous literature, background challenges, motivations, and contributions, the remainder of the paper is highly technical and quite inaccessible to readers without deep mathematical expertise in differential geometry. The technical complexity is not adequately tailored for a broader audience, and some figures and tables lack sufficient clarity to facilitate
 understanding.

**Questions:**

**Paper structure**
- Section 2 occupies the majority of the paper, introducing the method and proving its properties. However, it is highly technical and not very fluid to read. I would recommend reducing the amount of space devoted to technical details and expanding the sections that provide intuitive explanations for readers unfamiliar with these concepts. This could be achieved through diagrams, visual representations, or a clear algorithmic summary outlining how training and generation work, making it easier to compare this approach with standard diffusion models. Additionally, a brief introduction to other geometry-aware generative methods and a comparison with them would be helpful.

**General concerns**
- You frequently mention scalability as a core property of your method, but this is not demonstrated in the experiments. What exactly do you mean by scalability, and how do you justify this claim? How would the model perform on higher-order geometric structures?
- What is the computational cost of repeatedly computing the Christoffel symbols, and more generally, what are the computational requirements of your method? You mention “offering computational compatibility with Euclidean learning” - does this also hold for the lifting operations?
- Is this lifting more efficient than, for instance, performing geodesic simulations when closed-form geodesics are unavailable in other Riemannian methods that rely on geodesic computation?

**Writing/figures**
- Figure 3: What is the ground truth for the figure on the left? It is also unclear how the last two figures relate to the second one. Overall, this figure is not very informative, as there is no ground truth provided for comparison with the generated samples.
- Table 1: What do the triangles represent, and what does the “Closed-form?” row indicate?

---

### Official Review · Reviewer_kjP2 · 2025-11-09

**Soundness:** 3
**Presentation:** 1
**Contribution:** 2
**Rating:** 2
**Confidence:** 2

**Summary:**

The authors propose a technique for score-based diffusion when the data lie on Riemannian manifolds. Under certain geometric choices, the proposed approach does not rely on manifold-specific neural architectures or geometric operations that are difficult to compute in general. Instead, it resembles Euclidean diffusion models, relying only mildly on geometric quantities. The authors provide theoretical analysis for components of the method, as well as experiments supporting their claims.

**Strengths:**

- The approach is very interesting, as it combines differential geometric principles in a clever way to simplify diffusion on manifolds.
- The theoretical analysis is provided and appears sound, but I have not checked all proofs in detail.
- The experiments seem to support the claim that the proposed approach is more effective than related methods.

**Weaknesses:**

- The accessibility of the paper is rather limited. It combines two advanced topics, differential geometry and diffusion models, resulting in a dense manuscript that is difficult to follow. I acknowledge that the analysis must be rigorous and that some concepts cannot be simplified further; however, a small illustrative example could have helped clarify the concepts and break down the method’s steps explicitly.
- Overall, the actual functionality of the method and the motivation behind the approach remain somewhat unclear.
- As far as I understand, the method requires access to the parametrization of the manifold to compute the necessary geometric quantities, even though the paper claims this is not necessary (line 177). It seems that some geometric quantities must indeed be known to ensure proper diffusion on the manifold.

**Questions:**

Q1. My main question is about the intuitive functionality of the approach, which I find somewhat unclear. I would appreciate it if the authors could explain the method in concrete, simplified steps.

Q2. How is the noisy process defined in the intrinsic space, and why is the horizontal lift necessary in the approach?

Q3. It is unclear how parallel transport can be avoided when comparing tangent vectors at different points (line 216 and following).

Q4. I suppose that the logarithmic map (Eq. 16) may work only for infinitesimally close points, is this correct?

---

### Note · Authors · 2025-11-17

**Comment:**

Thank you for the constructive review. We have incorporated your valuable feedback into the revised manuscript, and we plan to submit the improved version to an upcoming conference.

**Withdrawal Confirmation:**

I have read and agree with the venue's withdrawal policy on behalf of myself and my co-authors.